# Modelling integrated soil fertility management for maize production in Kenya using a Bayesian calibration of the DayCent model.

Moritz Laub[1], Magdalena Necpalova[1,2], Marijn Van de Broek[1], Marc Corbeels[3,4], Samuel Mathu Ndungu[4], Monicah Wanjiku Mucheru-Muna[5], Daniel Mugendi[6], Rebecca Yegon[6], Wycliffe Waswa[4], Bernard Vanlauwe[4], and Johan Six[1]

[1]Department of Environmental Systems Science, ETH Zurich, 8092 Zürich, Switzerland
[2]University College Dublin, School of Agriculture and Food Science, Ireland
[3]AIDA, University of Montpellier, CIRAD, Avenue d'Agropolis, F-34398 Montpellier, France
[4]International Institute of Tropical Agriculture (IITA), c/o ICIPE Compound, P. O. Box 30772-00100, Nairobi, Kenya
[5]Department of Environmental Sciences and Education, Kenyatta University, P.O. Box 43844-00100, Nairobi, Kenya
[6]Department of Water and Agricultural Resource Management, University of Embu, P.O. Box 6-60100, Embu, Kenya

**Correspondence:** Moritz Laub (moritz.laub@usys.ethz.ch)

**Abstract.** Sustainable intensification schemes such as integrated soil fertility management (ISFM) are a proposed strategy to close yield gaps, increase soil fertility, and achieve food security in sub-Saharan Africa. Biogeochemical models such as DayCent can assess their potential at larger scales, but these models need to be calibrated to new environments and rigorously tested for accuracy. Here, we present a Bayesian calibration of DayCent, using data from four long-term field experiments in Kenya in a leave-one-site-out cross-validation approach. The experimental treatments consisted of the addition of low- to high-quality organic resources, with and without mineral N fertilizer. We assessed the potential of DayCent to accurately simulate the key elements of sustainable intensification, including 1) yield, 2) the changes in soil organic carbon (SOC), and 3) the greenhouse gas (GHG) balance of $CO_2$ and $N_2O$ combined.

Compared to the initial parameters, the cross-validation showed improved DayCent simulations of maize grain yield (with the Nash–Sutcliffe model efficiency (EF) increasing from 0.36 to 0.50) and of SOC stock changes (with EF increasing from 0.36 to 0.55). The simulations of maize yield and those of SOC stock changes also improved by site (with site-specific EF ranging between 0.15 and 0.38 for maize yield and between -0.9 and 0.58 for SOC stock changes). The four cross-validation-derived posterior parameter distributions (leaving out one site each) were similar in all but one parameter. Together with the model performance for the different sites in cross-validation, this indicated the robustness of the DayCent model parameterization and its reliability for the conditions in Kenya. While DayCent poorly reproduced daily $N_2O$ emissions (with EF ranging between -0.44 and -0.03 by site), cumulative seasonal $N_2O$ emissions were simulated more accurately (EF ranging between 0.06 and 0.69 by site). The simulated yield-scaled GHG balance was highest in control treatments without N addition (between 0.8 and 1.8 kg $CO_2$ equivalent per kg grain yield across sites) and was about 30 to 40% lower in the treatment that combined the application of mineral N and of manure at a rate of 1.2t C ha$^{-1}$ yr$^{-1}$. In conclusion, our results indicate that DayCent is well-suited for estimating the impact of ISFM on maize yield and SOC changes. They also indicate that the trade-off between maize yield and GHG balance is stronger in low-fertility sites, and that controlling SOC losses, while difficult to achieve through the addition of external organic resources, is a priority for the sustainable intensification of maize production in Kenya.

# 1   Introduction

In Kenya, as in many other Sub-Saharan Africa (SSA) countries, maize yields have remained low, on average 1.7 t ha$^{-1}$ compared to the global average of 5.6 t ha$^{-1}$ over the last decade (2011-2021; FAO, 2023). This contributes to the low self-sufficiency of food production, with around 20% of the Kenyan population facing severe food insecurity (World-Bank, 2021b). If yields are not improved, increased population growth will further deteriorate food self-sufficiency and food security in general in the coming decades (Zhai et al., 2021), especially considering expected yield declines resulting from more frequent extreme weather events (Lobell et al., 2011). One of the key limitations to sustainable maize production in SSA is the insufficient use of mineral fertilizer and organic inputs (Vanlauwe et al., 2010). Integrated soil fertility management (ISFM) is a sustainable intensification practice that can alleviate these limitations by combining the use of mineral fertilizers with organic inputs (Vanlauwe et al., 2010). Several studies have reported that ISFM has the potential to more than double maize yields in Kenya, especially on infertile soils, due to its positive impact on soil fertility, including soil organic matter (SOM) content (Chivenge et al., 2009, 2011). Furthermore, increasing SOM can help mitigate adverse effects of climate change, offering considerable potential in carbon-depleted soils across SSA (Corbeels et al., 2019). However, the effectiveness of ISFM in increasing yields strongly depends on local site conditions, such as soil and climate (Chivenge et al., 2011).

To close yield gaps in a resource-efficient way and to assess the climate change mitigation potential of ISFM, we need to understand the long-term effects of ISFM practices at a larger scale. Ideally, this would be facilitated by implementing a large number of long-term experiments across a representative range of soil and climatic conditions. However, the significant costs, labor, and time required to maintain long-term experiments limit the number of sites for evaluating the variable effects of ISFM practices under site-specific conditions. In addition, relying on statistical predictive techniques to upscale results from a limited number of sites may lead to low predictive power and large errors, because it is unlikely that the effects of soils and climate on yield and SOM would be fully captured in the statistical models.

Biogeochemical process-based ecosystem models, such as DayCent (Parton et al., 1998; Del Grosso et al., 2001), simulate the effects of important driving variables on crop yield and SOM formation using semi-mechanistic functions developed through decades of agronomic and soil research. Because they (partly) embed our current understanding of the complex ecosystem processes, they are more robust for scaling up the yield potential (Saito et al., 2021) and the SOM building capacity (Lee et al., 2020), compared to statistical predictive techniques. However, to avoid bias in model output, it is best practice that models are calibrated and evaluated for local conditions (Necpálová et al., 2015), especially when applied in novel contexts such as different climate zones with different soils.

Although DayCent has been used to estimate SOM stock changes in Kenya on a national scale (Kamoni et al., 2007), and recently to assess the impact of conservation agriculture on SOM in Ethiopia (Lemma et al., 2021), its modules of SOM and maize crop growth have never been rigorously calibrated and evaluated for tropical agroecosystems in SSA. A recent evaluation of DayCent in Kenyan maize systems showed that SOM turnover is underpredicted by the model (Nyawira et al., 2021). Because SOM is coupled to nitrogen (N) mineralization in biogeochemical models, there is the potential that this translates into biased crop responses to N addition and biased crop productivity predictions in any upscaling exercise. A potential solution

to this issue is the simultaneous calibration of soil and crop parameters in DayCent using data from local long-term experiments. Ideally, this calibration would include the uncertainty in the model parameters and model outputs (Clifford et al., 2014), so an estimation of uncertainties is possible in upscaling exercises (Stella et al., 2019). This is especially relevant given a recent study showing considerable uncertainty in DayCent's SOM turnover rates, even when calibrated using a range of long-term experiments (Gurung et al., 2020).

In order to use DayCent to assess the potential of ISFM to reduce yield gaps while minimizing environmental impact in Kenya and other SSA countries, this study used a Bayesian calibration to derive robust DayCent parameters of SOM cycling and maize growth in Kenya. With robust, we mean that the model evaluation statistics are representative of applying the model to new sites with the same climate and soils. We used the experimental data of four long-term ISFM experiments conducted in Kenya over nearly two decades (Laub et al., 2023a, b). Of these, two sites were in humid western Kenya and two in subhumid to semi-arid central Kenya.

The first objective of our study was to evaluate to what extent DayCent can reproduce the differences in yields and SOM development in response to the addition of different qualities and rates of organic resources combined with different rates of N fertilizer for a number of contrasting sites. The second objective was to evaluate the greenhouse gas (GHG) balance of different addition rates of organic material in ISFM to find the optimal balance between limiting GHG emissions from the soil and optimizing crop yield (that is, sustainable intensification). ISFM can be a source of $N_2O$ to the atmosphere (Leitner et al., 2020) but compared to standard practices, it reduces SOC losses or even increases SOC (Laub et al., 2023a), thereby mitigating $CO_2$ emissions.

The specific steps to reach the objectives of this study were (i) to test the capability of an uncalibrated version of DayCent to simulate yield and SOC development of the different ISFM practices, (ii) to calibrate DayCent to represent ISFM under Kenyan conditions using experimental data from four long-term experiments, displaying the uncertainty of model parameters by Bayesian calibration, and (iii) to use the calibrated model to gain understanding of the GHG balance of the different ISFM treatments.

## 2   Material and Methods

### 2.1   The experimental sites

The present study used data from four long-term field experiments in Kenya, in which the effect of the addition of different organic resources at different rates was tested, either alone or in combination with the application of mineral N fertilizer, in the context of ISFM. The sites are located in agriculturally important areas in central and western Kenya (Fig. A1). The Embu and Machanga sites are both located in Embu County, in the central part of Kenya. The Aludeka site is situated in Busia County in western Kenya, while Sidada is located in the adjacent Siaya County, south of Busia (Table A1). The experiments at Embu and Machanga began in early 2002, while those at Aludeka and Sidada began in early 2005. Therefore, 19 years of data were available in central Kenya and 16 years in western Kenya (2 sites x 16 years + 2 sites x 19 years = 70 site*years = 140 site*seasons). The sites cover a range of altitudes, temperatures, and precipitations. Embu, with a mean annual temperature

(MAT) of 20 °C and an annual precipitation of 1200 mm, is the coolest site, while Machanga has a MAT of 24°C and the lowest annual precipitation (800 mm). Sidada (23°C, 1700 mm) and Aludeka (24°C, 1700 mm) have a high MAT and receive significantly more precipitation than the sites in central Kenya. There are two rainy seasons at each site, corresponding to two maize growing seasons per year. The long rainy season occurs from March to August or September, while the short rainy season occurs from October until January or February. In terms of soil texture, Machanga and Aludeka have low clay content

(13% clay at both sites), while Sidada and Embu are rich in clay (56 and 60%, respectively).

All experiments were set up as a split plot design with three replicates, with different qualities and quantities of organic resources as main plots and the presence or absence of N fertilizer as subplots. Maize was grown continuously in all experiments, with two crops per year, one in the long rainy season and one in the short rainy season. The experimental design was identical at all four sites and has been described in detail in earlier publications (Chivenge et al., 2009; Gentile et al., 2011; Laub

et al., 2023a, b). Organic resource treatments consisted of high quality *Tithonia diversifolia* (TD) green manure, high quality *Calliandra calothyrsus* (CC) prunings, low quality stover of *Zea mays* (MS), low quality sawdust from *Grevillea robusta* trees (SD), locally available farmyard manure (FYM), and a control treatment (CT) without organic resource additions. Organic resources differed in quality by the contents of N, lignin and polyphenols (Table A2). Each organic resource was applied once a year at two rates, 1.2 and 4 t C ha$^{-1}$ yr$^{-1}$, while mineral N fertilizer was applied at a fixed rate of 120 kg N ha$^{-1}$ (CaNH$_4$NO$_3$)

in each of the two growing seasons. Of this, 40 kg N ha$^{-1}$ were applied at planting, and the remaining 80 kg N ha$^{-1}$ about six weeks later. Organic resources were applied only once a year, prior to planting in the long rainy season, i.e., in January or February. They were incorporated to a depth of 15 cm with hand hoes. Furthermore, a blanket application of 60 kg P ha$^{-1}$ as triple superphosphate and of 60 kg K ha$^{-1}$ as muriate potash at planting was provided to all plots each season. The plots were kept weed free by hand weeding, two to three times per season, and selective application of pesticides was used when

necessary to control armyworm, stemborer, and/or termites.

### 2.2 The DayCent model

DayCent (2017 version of DD_EVI) is a terrestrial ecosystem model of intermediate complexity (Del Grosso et al., 2001). It simulates daily C and N fluxes within the soil-plant-atmosphere continuum and has been parameterized for several crops and ecosystems (Necpalova et al., 2018). It has submodules to simulate plant growth, organic resource and soil organic matter

(SOM) decomposition including mineralization of N, soil water and temperature, N gas fluxes, and CH$_4$ oxidation. The net primary productivity of plants is a function of their genetic potential, a simplified phenology, solar radiation, temperature, and stresses, such as reduced water or N availability. Here, we used the non-growing degree day version of the DayCent crop module, that does not simulate phenology but has a seedling stage with reduced growth until a certain biomass (full canopy) is reached. SOC and soil N in the topsoil are represented by an active, slow, and passive SOM pool, while litter and organic

resources are represented by a structural and metabolic litter pool (Parton et al., 1987). All SOM pools are conceptual and have no measurable counterparts, whereas the litter pools are semi-quantitative. Their division is based on the measurable ratio of lignin to N in the organic resources and plant litter. DayCent can adequately simulate crop yields, SOC and soil N dynamics,

and $N_2O$ emissions in temperate conditions (Del Grosso et al., 2005; Necpálová et al., 2015; Necpalova et al., 2018; Gurung et al., 2020, 2021) but a recent paper showed inadequate performance for tropical conditions (Nyawira et al., 2021).

## 2.3 Data used for the DayCent model calibration and evaluation

To provide an overall assessment of the performance of DayCent for its use in Kenya a leave-one-site-out cross-validation approach was applied to evaluate the model performance. Specifically, this involved using a data sub-set from three of the four sites for model calibration, with evaluation performed using the data from the fourth site. This process was repeated four times, every time with another site serving as the evaluation site. Different data, were used for this: Maize grain yield and the aboveground biomass, both on a dry matter basis, were available for each cropping season between 2002 and 2020 (further details in Laub et al., 2023b). All this data was used with one exception - the short rainy season of 2019 at Sidada, which had unrealistically high maize grain yields of up to 16 t ha$^{-1}$. In addition, plot-scale SOC and total N contents in the top 15 cm soil layer were available at several time points, and in 2021 for the 0-30 cm soil depth. At Embu and Machanga, soil samples were taken every two to three years since the start of the experiment in 2002 until 2021, while at Sidada and Aludeka, soil sampling occurred only in 2005, 2018, 2019 and 2021 (further details in Laub et al., 2023a). Because soil bulk density data was not available for most time points and there was no significant difference in topsoil bulk density between treatments at any site in 2021, the mean soil bulk density per site was used to calculate SOC stocks of the top 15 cm of soil depth. We used a DayCent parameterization that was developed to simulate SOC stocks of the IPCC-recommended 0-30 cm topsoil layer (Gurung et al., 2020) (further details in section 2.3.2). Thus, the 0-15 cm SOC stocks were adjusted to 0-30 cm depth. This was done by adding the site-specific SOC stocks from the 15-30 cm layer (specifically, the 15-30 cm equivalent-soil-mass-based ones (Wendt and Hauser, 2013; Lee et al., 2009)) to the treatment-specific SOC stocks from 0-15 cm. Due to limited data availability for the 15-30 cm soil depth (only 2021), this approach was considered the most conservative and robust; subsoil carbon usually changes very slowly, and a statistical test revealed no differences in the equivalent soil mass based SOC stocks of the 15-30 cm layer (2.5-4.7 t soil ha$^{-1}$) between treatments at the same site in 2021 (with only one single exception at Aludeka; Fig. A2).

Data on $N_2O$ emissions were used in the model evaluation phase, but not for model calibration, due to their scarcity and high uncertainty. The $N_2O$ measurements were conducted after N fertilization in 2005 (weekly measurements form March to June at Embu and Machanga and daily measurements at Machanga in November), in 2013 and 2018 (weekly measurements form March to beginning of May at Sidada and Aludeka), and in 2021 (weekly measurements form mid-March to mid-May at Sidada). The measurements applied the static chamber method (Hutchinson and Mosier, 1981) with two measuring frames per plot permanently installed for a whole rainy season (one within, one between maize rows). The sampling chambers (0.27 × 0.375 × 0.11 m) had a vent tube and fan for to homogenize the gas sample before extraction with a 60 mL polypropylene syringe through a septum-sealed sampling port. Four gas samples were collected at 0, 15, 30 and 45 min of chamber closure. Gas samples from within and between maize rows were combined per time point in the same syringe (Arias-Navarro et al., 2017). All analyses were conducted using a SRI 8610C gas chromatography (456-GC, Scion Instruments, Livingston, United Kingdom) equipped with an electron capture detector for $N_2O$ analysis. Fluxes per surface area were determined using the linear slope of gas concentration over time (Pelster et al., 2017; Barthel et al., 2022). Simulated $N_2O$ emissions were evaluated

against measured daily and cumulative N$_2$O emissions. To determine the cumulative emissions at plot scale, we used the trapezoid method (Levy et al., 2017), specifically, the trapz function of R (Tuszynski, 2021). Treatment-scale means and variances of the daily and cumulative N$_2$O emissions were then computed in a similar way as for the other measurements.

Finally, continuous soil moisture measurements were conducted using sensors placed in each replicate at 10 cm soil depth (EC-5 Soil Moisture Sensor, Meter, Germany) in the control and the 1.2 t C plots of the *Calliandra*, farmyard manure and maize stover treatments at the Sidada and Aludeka sites (March 2018 to December 2020). These soil moisture data were used to initially determine the optimal pedotransfer functions for soil hydraulic conductivity but not used in the model calibration.

### 2.3.1 Model driving variables and model assumptions

The site-specific crop management data was obtained from season- and site-specific records of field management operations. These included dates of organic resource application, manual plowing before planting, maize planting, split application of mineral N, weeding and harvest. Dates of pesticide applications and gap filling or maize thinning were also available, but these operations are not part of standard DayCent management and were therefore not included in modelling. Therefore, our model runs assumed no occurrence of pests or diseases, and an optimal plant density at emergence, which, in practice, was ensured
by manual thinning and gap filling.

Recorded weather data existed for all sites, but filling in data gaps was necessary due to unavailability and loss of recorded data. At Embu and Machanga, manual recordings of daily minimum and maximum temperature and precipitation were available from 2002 until the end of 2007, but from 2008 until 2017, only measured precipitation was available. After 2017, newly installed TAHMO stations (https://tahmo.org/climate-data/) were available for these two sites, providing daily values for tem-
175 perature and precipitation. At Aludeka and Sidada, manual recordings of daily minimum and maximum temperature and precipitation were available for all years from 2005 to 2017. Thereafter, weather stations (Meter climate station, Meter Environment, Munich, Germany) were installed and provided the data. Data gaps were filled by using the NASA POWER product (https://power.larc.nasa.gov/docs/methodology/). A bias correction for the minimum and maximum temperature of NASA POWER data was performed, using a linear regression with measured data as dependent variable (y) and NASA POWER data
as independent variable (x). Specifically, the slope and intercept of the regression equation $y = mx + b$, were used to produce a corrected estimate of these data. In our specific case, the slopes were not significantly different from 1, but intercepts ($b$) were significantly different from 0. The specific intercepts for maximum temperature were -0.3°C, -0.4°C, +3°C and +6°C for Embu, Machanga, Sidada and Aludeka, respectively. The intercepts for the minimum temperature were -0.25°C, -0.5°C, -3°C and +1°C for Embu, Machanga, Sidada, and Aludeka, respectively. For precipitation, no bias correction was done.

The data on the soil hydraulic properties needed in DayCent (volumetric soil water content at field capacity, wilting point, and saturated hydraulic conductivity K$_S$) were calculated based on the soil texture measured at each site. The pedotransfer functions of Hodnett and Tomasella (2002) was used, because it was specifically designed for tropical soils. Its' soil hydraulic properties also showed better agreement between the measured and simulated soil moisture contents than when soil hydraulic properties of Saxton and Rawls (2006) were used. Because the Hodnett and Tomasella (2002) equation does not provide a
method to estimate K$_S$, K$_S$ was calculated using the Saxton and Rawls (2006) equation, with values of the water retention

curve, $\alpha$ and $n$ (van Genuchten, 1982), calculated with the equation from Hodnett and Tomasella (2002). The equations can be found in the supplementary material (A1).

### 2.3.2   Initial model parameterization and selection of potentially sensitive parameters for calibration

It was assumed that the organic resource inputs had the same properties across all sites (i.e., mean values of lignin contents and C/N ratios per organic resource were assumed; Table A2). This approach was used because measurements were not available for all sites and years, and was justified as an analysis of variance of data from the years 2002, 2003, 2004, 2005 and 2006 at Embu and Machanga, and from 2018 at all sites, did not indicate any significant differences in lignin contents and C/N ratios between the sites or years. The C content of maize grain was assumed to be 42.5% throughout the simulation period. This was the mean value of measured grain C content across sites (standard deviation 1.8%) in the short rainy season 2018 and long rainy season 2019 (data not shown). Given the strong correlation between maize grain yield and aboveground biomass in the measured data (r = 0.87), the aboveground biomass data was transformed to harvest index data for the model calibration process, because harvest index had a lower correlation with yield (r = 0.59) than aboveground biomass.

The DayCent simulations were conducted at the treatment scale using average values across all three replicate plots for soil parameters (i.e., soil texture, bulk density, pH), SOC and soil N stocks, maize grain yield and aboveground biomass/harvest index. This aggregation was done to reduce the computation time of the simulations and because initial tests showed similar model performance as compared to applying the model to each experimental replicate individually. The site-specific standard deviation for each type of measurement was used as a measure of uncertainty of the measured data (computed from the three replicates at each time point for each treatment at each site). This choice was based on the statistical models of Laub et al. (2023a, b), showing variance heterogeneity between sites but not between treatments.

The standard parameter values of the DayCent 2020 version were taken as initial model parameters, with three exceptions. First, we used the adjusted decomposition parameter values of the SOM pools from Gurung et al. (2020) to allow the use of DayCent for simulating SOC stocks of the 0-30 cm soil depth layer instead of the standard 0-20 cm layer. Second, we modified the parameter value representing the fraction lost as $CO_2$ upon structural litter and lignin turnover (ps1co(1&2)&rsplig). The default value for this parameter is 0.5 assigning a carbon use efficiency (CUE) value of 50% to structural litter, based on outdated theories that lignin-rich materials form stable SOC most efficiently (Frimmel and Christman, 1988). Newer studies have, however, clearly shown that minimal structural litter is conserved in the long term, while metabolic litter forms SOC more efficiently (Cotrufo et al., 2013; Denef et al., 2009; Puttaso et al., 2013; Kallenbach et al., 2016). Thus, we opted for a more realistic prior value of 0.85 for ps1co(1&2)&rsplig, corresponding to a more plausible CUE value of 15% for structural litter (Mueller et al., 1997). Third, for the parameters determining the minimum and maximum proportion of nitrified N lost as $N_2O$, we used values that fell between the DayCent default values and recent values from Gurung et al. (2021). This choice was motivated by the fact that the DayCent default parameter values led to excessively high emissions, while the Gurung et al. (2021) parameter values resulted in emissions that were too low. Finally, we assumed that the maize growth parameters of the second highest production level (C5 in DayCent) represent best the production levels observed in the experiment.

To identify which model parameters to include in the global sensitivity analysis (see section 2.4) and model calibration, we reviewed literature for recently conducted sensitivity analyzes of the DayCent model (Necpálová et al., 2015; Gurung et al., 2020). Additionally, we consulted the DayCent manual to identify and add further parameters of potential importance for the processes considered in our study (i.e., plant productivity and soil C and N cycling). This resulted in a selection of 66 parameters (Table 1 and Table A3). Some of these parameters belong to the same category, but can be individually calibrated in DayCent. For example, the "tillage multiplier" of SOM turnover can have different values for different SOM pools but is usually the same for all SOM pools in the standard DayCent parameterization. Thus, we decided to have the same tillage multiplier value for all SOM and litter pools. Some parameters can have different values between the surface and soil SOM pools (e.g., C/N ratios and turnover rates). For simplicity, we assigned the same C/N ratios and a constant ratio to the turnover rates of surface and soil SOM pools (i.e., decX(2)/decX(1)). This simplified parameter sensitivity analysis and calibration with regard to surface and soil SOM pools. Finally, the parameters governing the minimum and maximum values were reformulated. Instead of calibrating them as a maximum and a minimum value, we considered the maximum value and the difference between the minimum and maximum values (i.e., N2Oadjust_(max-min) and aneref(1)-anaref(2)). This ensured that the minimum value was smaller than the maximum value, thereby avoiding numerical problems(initial N2Oadjust_max was set to 0.015; N2Oadjust_(max-min) to 0.003).

### 2.3.3 Soil organic matter pools initialization based on measured data

Instead of relying on spin-up simulation based on uncertain historical land use and management of the simulated sites, we used measured mineral associated organic carbon (MAOC) fractions as a proxy for the initialization of the passive SOM pool (Zimmermann et al., 2007). Replacing SOM initialization assumptions with measured proxies can enhance model performance (Laub et al., 2020; Wang et al., 2023), and, more importantly, is less sensitive to user assumptions. It also aligns with the DayCent concepts on SOM; the manual (Hartman et al., 2020) denotes that particulate organic carbon (POC) and MAOC are related to the slow and the passive SOM pool, respectively. MAOC data for samples from the 0-30 cm soil layer was available from the year 2021 (specifically for the control -N, control +N and the farmyard manure -N and *Tithonia diversifolio* -N treatments at 4 t C ha$^{-1}$ yr$^{-1}$ at all sites). It was derived by density fractionation using sodium polytungstate solution (1.6 g cm$^{-3}$ for Aludeka and 1.7 g cm$^{-3}$ for the other sites). Aggregates were dispersed with ultrasonication at 400 J ml$^{-1}$ (217 s at 200-240W), after which samples were centrifuged for 2h at 4700 rpm to separate the heavy and the light fraction, which were then separated, washed with deionised water, dried at 60°C for 24h and analyzed for weight and C content. A statistical analysis revealed the absence of treatments differences within the same site, so the site-specific MAOC values for the 0-30 cm soil depth across treatments (0.91, 0.88, 0.85, 0.86 g MAOC g$^{-1}$ SOC for Aludeka, Embu, Machanga, and Sidada in 0-30 cm, respectively) were used to initialize the SOC in the passive SOM pool in DayCent simulations. Further, 3% of initial SOC was assigned to the active SOM pool (mean value recommended in the DayCent manual) and the remainder of SOC was assigned to the slow SOM pool.

The DayCent manual further states that, although the slow SOM pool is closely related to the POC fraction, it tends to be larger (Hartman et al., 2020). Consequently, the passive SOM pool must be smaller than the MAOC fraction. Additionally, the

fractionation data was from 2021, when the experiments were already 19 and 16 years old. To address these issues, two new parameters were introduced in the simulations: 1) an intercept ($IC_{MAOC}$) to account for the passive SOM pool being smaller than the MAOC fraction, and 2) a slope for the time since the start of the experiment ($SL_t$) to account for SOM changes (mostly losses) since the start of the experiments, with the passive SOM pool typically changing at the slowest rate. Given that all sites were converted to agriculture only a few decades ago (Laub et al., 2023a), the percentage of total C in the passive SOM pool at the start of the experiment should be higher than the 30-40 %, that are common at steady state of SOM pools (Hartman et al., 2020). Considering this, it was assumed that the intercepts initial value was -0.1 g MAOC g$^{-1}$ SOC and the slopes initial value value was -0.005 g MAOC g$^{-1}$ SOC yr$^{-1}$ since the start of the experiment, giving both terms approximately the same weight. Thus, the fraction of SOC in the passive SOM pool at the start of the experiment was

$$SOC_p(g\ g^{-1}) = MAOC_{2021} + IC_{MAOC} + SL_t * t_{dif} \tag{1}$$

Here, SOC$_p$ represents the fraction of SOC in the passive SOM pool at the start of the experiment, MAOC$_{2021}$ the MAOC fraction in 2021 (g MAOC g$^{-1}$ SOC), IC$_{MAOC}$ the intercept, and SL$_t$ the slope value that is multiplied by the time difference between the measurement and the start of the experiment in years (t$_{dif}$). With the selected standard values for IC$_{MAOC}$ and SL$_t$, between 66% (Machanga) and 73% (Aludeka) of SOC were assumed to be in the passive SOM pool at the start of the experiment. The uncertainty related to this initialization approach was accounted for in the model calibration by allowing large ranges for these parameters. Finally, to initialize the soil N pools, C/N ratios of the active, slow, and passive SOM pools were set to 10, 17.5, and 8.5, respectively, which are the best estimates provided by the manual (Hartman et al., 2020).

## 2.4 Global sensitivity analysis

To reduce the number of optimised parameters during the calibration, we performed a parameter screening (van Oijen, 2020). For this purpose, a global sensitivity analysis was conducted to quantify the relative importance of different model parameters to the relevant model outputs regarding our study's focus on maize yield and the greenhouse gas mitigation potential of ISFM. The aim was to identify and fix less influential model parameters to their initial values, reducing the computational cost for performing the consecutive Bayesian model calibration (see section 2.5). The global sensitivity analysis was performed using the Sobol method (Saltelli, 2002a, b), which allows for the estimation of the proportion of variance in the model outputs that is explained by each model parameter, while considering the interaction terms of first-order and higher-order (Gurung et al., 2020). The "sensitivity" package (function sobolSalt; Iooss et al., 2021) of R version 4.0 (R Core Team, 2020) was applied. This function implements a simultaneous Monte Carlo estimation of first-order and total-effect Sobol indices. The computational cost is $N(p+2)$ model runs, $N$ being the dimension of the two matrices to construct the Sobol sequence, $p$ being the number of parameters (66 in our case). Our tests indicated similar results for $N$ = 500/1000, so we chose a dimension of 1000. The preselected model parameters to include are described above and in Tables 1 and A3. The ranges used for the global sensitivity analysis were centered around the initial parameter value obtained as described above (section 2.3.2). The upper and lower parameter boundaries were based on previous sensitivity analyses (e.g. Necpálová et al., 2015; Gurung et al., 2020), plausible

ranges reported in the DayCent manual and variations observed in different maize parameterizations in the literature. The parameters were then grouped according to the magnitude of their ranges. Parameters with very small, small and moderate ranges were varied by ±10, 25 and 50% from the initial parameter value, respectively. For parameters with large and very large ranges, the upper boundaries were the initial parameter values multiplied by 3 and 10, respectively, the lower boundaries were the initial parameter values divided by 3 and 10, respectively. The parameter sensitivity was independently determined for the

mean maize grain yield and aboveground biomass, averaged over all seasons at all sites, as well as for the SOC and soil N stocks at the end of the simulation period (equations presented in the supplement A2).

**Table 1.** DayCent model parameters and the coefficient of variation used in the calibration. Displayed are parameters considered for calibration due to total sensitivity index > 2.5% (top) and with a total sensitivity index > 1% (bottom). The remainder of parameters (<1%) are not included in this table and can be found in the supplementary (Table A3). The presented calibrated parameter values correspond to the single parameter set with the highest likelihood, which was derived by using the data from all four sites combined. The posterior was also derived by using the data from all four sites combined. Abbreviations: CV, coefficient of variation; SD, standard deviation, 95% CI, 95% credibility interval; Ly, langley.

| Parameter | Description | Possible ranges of values | Units | Initial value | CV | Calibrated value | Posterior mean | SD | 95% CI |
|---|---|---|---|---|---|---|---|---|---|
| **Included in calibration (total sensitivity >2.5%)** | | | | | | | | | |
| himax | Maximum harvest index for maize | moderate | g g$^{-1}$ (C) | 0.40 | 0.23 | 0.43 | 0.46 | 0.06 | 0.36-0.59 |
| ppdf(1) | Optimum temperature for growth of maize | very small | °C | 30.00 | 0.08 | 28.63 | 28.5 | 1.67 | 25.44-31.57 |
| ppdf(2) | Maximum temperature for growth of maize | very small | °C | 45.00 | 0.08 | 47.1 | 46.48 | 3.03 | 40.01-52.19 |
| prdx(1) | Potential aboveground production of maize | large | g C m$^{-2}$ Ly$^{-1}$ | 2.25 | 0.38 | 2.62 | 2.45 | 0.39 | 1.86-3.37 |
| clteff(1,2&4) | Tillage multiplier for SOM turnover | large | unitless | 10.00 | 0.38 | 4.93 | 9.02 | 3.35 | 2.91-15.78 |
| aneref(3) | Min. impact of soil anaerobiosis on SOM turnover | large | unitless | 0.95 | 0.38 | 0.79 | 0.82 | 0.13 | 0.55-0.99 |
| dec4 | Max. turnover rate of passive SOM pool | very large | g g$^{-1}$ yr$^{-1}$ | 0.0035 | 0.45 | 0.0060 | 0.004 | 0.001 | 0.001-0.007 |
| dec5(2) | Max. turnover rate of slow SOM pool | large | g g$^{-1}$ yr$^{-1}$ | 0.10 | 0.38 | 0.13 | 0.12 | 0.03 | 0.06-0.17 |
| fwloss(4) | Scaling factor potential evapotranspiration | moderate | unitless | 0.75 | 0.23 | 0.94 | 0.9 | 0.05 | 0.81-0.99 |
| pmco2(1&2) | C lost as $CO_2$ with metabolic litter turnover[*] | large | g g$^{-1}$ (C) | 0.54 | 0.38 | 0.91 | 0.71 | 0.11 | 0.48-0.91 |
| ps1co2(1&2)&rsplig | C lost as $CO_2$ with structural litter turnover[*] | large | g g$^{-1}$ (C) | 0.85 | 0.38 | 0.77 | 0.86 | 0.11 | 0.61-0.99 |
| IC$_{MAOC}$ | Intercept for fraction of MAOC in slow pool | very large | g g$^{-1}$ (C) | -0.1 | 1 | -0.02 | -0.1 | 0.08 | -0.25-0.06 |
| SL$_t$ | Slope for time difference of MAOC measurement | large | g g$^{-1}$ yr$^{-1}$ (C) | -0.005 | 0.38 | -0.006 | -0.005 | 0.002 | -(0.001-0.008) |
| **Not included in calibration (total sensitivity <2.5% & > 1% )** | | | | | | | | | |
| frtc(1) | C allocated to roots at planting, without stress | small | fraction of NPP | 0.50 | 0.15 | - | - | - | - |
| frtc(3) | Time after planting at which maturity is reached | small | number of days | 90.00 | 0.15 | - | - | - | - |
| pramn(1,2) | Min. aboveground C/N ratio at maturity | small | C/N ratio | 62.50 | 0.15 | - | - | - | - |
| hiwsf | Max. harvest index reduction with water stress | moderate | g g$^{-1}$ (C) | 0.60 | 0.23 | - | - | - | - |
| teff(1) | Temperature inflection point (SOM turnover) | moderate | unitless | 17.05 | 0.23 | - | - | - | - |
| varat21&22(2,1) | Min. C/N ratio for material entering slow SOM pool | small | C/N | 12.00 | 0.15 | - | - | - | - |
| basef | Soil water of bottom layer lost via base flow | moderate | fraction $H_2O$ | 0.30 | 0.23 | - | - | - | - |
| N2Oadjust_max | Proportion of nitrified N that is lost as $N_2O$ | large | g g$^{-1}$ (N) | 0.015 | 0.38 | - | - | - | - |
| MaxNitAmt | Maximum daily nitrification amount | large | g N m$^{-2}$ | 0.40 | 0.38 | - | - | - | - |

[*](1 - microbial carbon use efficiency)

## 2.5 Combined Bayesian calibration of plant and soil model parameters

Bayesian calibration is a probabilistic inverse modeling or data assimilation technique, which is used to estimate the joint posterior distribution of model parameters ($\theta$) given the measured data ($D$) and the model structure ($M$), expressed as $p(\theta|D, M)$. It uses the proportionality form of Bayes' theorem, where $p(\theta|D, M)$ is proportional to the prior belief about model parameters, $p(\theta)$ times the likelihood function of the data given the model and the parameters, $p(D|M,\theta)$:

$$p(\theta|D, M) \propto p(\theta) * p(D|M,\theta) \tag{2}$$

While the prior, $p(\theta)$, is chosen based on previous knowledge of the model parameters, the likelihood function, $p(D|M,\theta)$, measures how well the model and the data match. In practice it is derived for a given set of parameters sampled from the prior, by running and evaluating the model using the measured data, the simulated counterpart and the variance-covariance matrix of the model residuals. Following Gurung et al. (2020), we applied the R software (R Core Team, 2020) to create a mixed model with restricted maximum likelihood estimation with the lme4 package (Bates et al., 2015), which automatically constructed the variance covariance matrix based on the nested design of observations to account for autocorrelation of residuals. The likelihood was a function of the following form:

$$p(D|M,\theta_z) = \frac{1}{\sqrt{2\pi\Sigma}} \exp\left(-\frac{1}{2}(M(\theta_z) - D)^T \Sigma^{-1}(M(\theta_z) - D)\right) \tag{3}$$

Here, $\Sigma$ is the variance covariance matrix, $M(\theta_z)$ is the vector of simulated values using the z-th parameter set $\theta_z$ and $D$ the vector of observed data. In the R software, this can be constructed by setting the residual (modelled value - measured) as the dependent variable of a zero intercept model with nested random effects (i.e., sampling date within site), and assigning the inverse of the median standard deviation (of each type of measurement at each site) as weight. By using the inverse of the standard deviation of each type of measurement as weight of the zero-intercept model, it is possible to include different types of measurements into the same likelihood function. This is similar to what is done in weighted analyses commonly performed in meta-analyses (Möhring and Piepho, 2009). The logLik() function is then used to extract the log-likelihood, which is transformed to the likelihood by raising $e$ to the power of the log-likelihood.

The sampling importance resampling method, which was used in this study, is a direct form of Bayesian calibration, which has recently been used by Gurung et al. (2020), to calibrate the parameters of the SOM module of DayCent using a large collection of temperate long-term experiments. It samples the prior by running the model for a large sample of parameter sets of size *I* from the prior, computing the likelihood for each sample, and filtering the prior based on importance weights $w(\theta_z)$

$$w(\theta_z) = \frac{p(D|M,\theta_z)}{\sum_{i=1}^{I} p(D|M,\theta_z)} \tag{4}$$

where $p(D|M,\theta_z)$ is the likelihood function of the *z*th parameter set and $w(\theta_z)$ is the corresponding importance weight. It is consistent with the proportionality form of Bayes' theorem in that it uses the importance weights $w(\theta_z)$ as probabilities for sampling from the prior, without replacement, to derive the posterior.

Combined Bayesian calibration of the sensitive DayCent parameters was performed using all available data on maize grain yield, harvest index (calculated from aboveground biomass), and SOC stocks. A notable exception was that SOC stocks from the Machanga site were not used in the calibration process, because this site was severely affected by soil erosion (Laub et al., 2023a) that is not represented by DayCent. The main reason for only using grain yield, harvest index and SOC data was that the yields, SOC stocks, and their trade-offs were the focus of this study. Technical constraints also influenced the decision; the creation and readout of daily simulation outputs to match simulated and measured soil moisture content, mineral N content and $N_2O$ fluxes would slow down the whole Bayesian calibration process by a factor of 5. The Bayesian calibration would have taken more than three months on the virtual machine with 64 cores. Following Gurung et al. (2020), model parameters that had a total sensitivity index of at least 2.5% for either yield, aboveground biomass, or SOC were considered influential and thus were subjected to calibration (11 parameters). Additionally, the new parameters associated with the initialization, $IC_{MAOC}$ and $SL_t$ had to be calibrated, resulting in a total of 13 parameters for calibration (Table 1).

Overall, a total of 200000 simulations were performed, from which 0.1% (200) of the parameter sets were sampled to derive the posterior distribution through resampling (Gurung et al., 2020). It was assured that this number of simulations was sufficient by splitting the simulations into two halves and visually assessing the similarity of derived posteriors for these subsets. In our experience, the sampling importance resampling algorithm is highly suitable for DayCent, that is prone to crashing with inappropriate parameter combinations. Unlike chain-dependent methods such as Markov Chain Monte Carlo, this method relies on model runs that are independent of each other, ensuring that an erroneous run does not stop the algorithm. In addition, this method allows for an efficient cross-validation of the posterior parameter set, such as the leave-one-site-out cross-validation employed in this study. Notably, the sampling importance resampling algorithm's advantage lies in its ability to store model results for each parameter set by site, allowing for straightforward cross-validation by site, without the need for rerunning the model for each iteration. The posterior parameter distributions of this study are displayed for both the leave-one-site-out cross-validation and the combined dataset from all four sites (Fig. 2). The former shows the importance of individual sites in the calibration process, while the latter provides the most representative posterior distribution for model upscaling, making efficient use of all available data.

To ensure computational efficiency, we used informed Gaussian priors that were centered around the standard parameter values of DayCent, with different coefficients of variation based on different observed ranges in previous studies. To make optimal use of existing knowledge about the parameters, the selected coefficients of variation per range were initially based on previous studies that had performed Bayesian calibration of the DayCent model. The coefficients of variation were chosen in a way that the prior from our study covered the whole range of the posterior from previous studies and then multiplied by a factor of 1.5 to account for the additional uncertainty that arose from applying DayCent at tropical sites. The studies of Gurung et al. (2020) and Mathers et al. (2023) were the basis to derive the coefficient of variation for the parameters dec4, dec5(2), clteff(1,2,&4), ps1co2(1&2) & rsplig, and pmco2(1&2). The study of Yang et al. (2021) was the basis for the parameters ppdf(1) and ppdf(2), and the study of (Necpálová et al., 2015, though not being Bayesian) was the basis for the parameters aneref(3) and fwloss(4). For himax and prdx(1), we looked into the default parameters of annual crops in DayCent, to assure that the whole range of values (0.30-0.55, and 1.1-3.5: 1.7-2.5, respectively) was covered by the prior. The final coefficients

of variation were 0.08, 0.15, 0.23, 0.38 and 0.45 for parameters with very small, small, moderate, large and very large ranges (Table 1). For the newly introduced parameters, we used large coefficients of variation, namely 0.38 for $SL_t$ and 1 for $IC_{MAOC}$, the reason for the latter being an initial test, in which $IC_{MAOC}$ was set to -0.3 instead of -0.1, which proved to be too low, but the uncertainty range with a standard deviation of 0.1 proved to be reasonable. Additionally, all parameters were constrained to remain within their physically sensible limits (i.e., not <0 for all and not >1 for those representing fractions).

## 2.6 Model evaluation

We used the following standard model evaluation statistics (Loague and Green, 1991):

$$MSE_y = \frac{1}{n}\sum_{z=1}^{n}(O_{yz} - P_{yz})^2 \tag{5}$$

$$RMSE_y = \sqrt{MSE_y} \tag{6}$$

$$EF_y = 1 - \frac{\sum_{z=1}^{n}(O_{yz} - P_{yz})^2}{\sum_{z=1}^{n}(O_{yz} - \bar{O}_y)^2} \tag{7}$$

The $MSE_y$ is the mean-squared-error and $RMSE$ is its root. $EF_y$ is the Nash-Sutcliffe model efficiency. We further divided $MSE_y$ into squared bias ($SB$), nonunity slope ($NU$) and lack of correlation ($LC$), as suggested by Gauch et al. (2003). We expressed them as a percentage of the $MSE_y$:

$$SB_y(\%) = \frac{(\bar{O}_y - \bar{P}_y)^2}{MSE_y} * 100 \tag{8}$$

$$NU_y(\%) = \frac{(1 - b_y)^2 * \left(\frac{\sum_{z=1}^{n}(O_{yz}^2)}{n}\right)}{MSE_y} * 100 \tag{9}$$

$$LC_y(\%) = \frac{(1 - r_y)^2 * \left(\frac{\sum_{z=1}^{n}(P_{yz}^2)}{n}\right)}{MSE_y} * 100 \tag{10}$$

Here, $O_{yz}$ is the measured value of the *z-th* measurement of the *y-th* type of measurement, $\bar{O}_y$ the mean of the *y-th* type of measurement and $P_{yz}$ the simulated value corresponding to $O_{yz}$. $\bar{P}_y$ is the mean predicted value of the *y-th* measurement type, $b$ the slope of the regression of *P* on *O* and *r* the correlation coefficient between *O* and *P*. The indicators *LC*, *SB* and *NU* show the nature of model errors, that is, a high *LC* shows that it is mostly random, a high *SB* a systematic bias, while a high *NU* shows issues of model sensitivity.

## 2.7 Greenhouse gas balance

To compare different ISFM treatments in terms of their greenhouse gas (GHG) emissions, their net GHG balance was computed on a yearly basis (kg $CO_2$eq ha$^{-1}$ yr$^{-1}$) over the whole simulation period. This calculation was based changes in the SOC content and cumulative emissions of $N_2O$ using a 100-year time horizon of global warming potentials (Necpalova et al., 2018):

$$GHG\ balance = \frac{44}{12} * \Delta SOC + 265 * N_2O \tag{11}$$

Here, $\Delta SOC$ is the change in SOC content (kg C ha$^{-1}$ yr$^{-1}$), $N_2O$ the cumulative $N_2O$ flux (kg $N_2O$ ha$^{-1}$ yr$^{-1}$). The $CH_4$ oxidation capacity was not considered, because it usually makes a very limited contribution to GHG balance in rainfed cropping systems (Lee et al., 2020) and we did not have data to evaluate the reliability of this simulated flux. In addition to the net annual GHG balance (in t $CO_2$eq ha$^{-1}$ yr$^{-1}$), we calculated the yield-scaled GHG balance (in $CO_2$eq kg$^{-1}$ maize grain yield) by dividing the cumulative GHG balance over the entire simulation period by cumulative simulated yields (dry matter base).

## 3 Results

### 3.1 Most sensitive DayCent parameters

The results of the global sensitivity analysis showed that of the 66 model parameters analyzed, only 20 parameters had a Sobol total sensitivity index >1% for either maize grain yield, aboveground biomass, SOC or soil N stocks (Fig. 1). Of these, only 11 parameters had a Sobol total sensitivity index >2.5%, a threshold that captures the most influential parameters and represents a suitable selection of parameters for model calibration (Gurung et al., 2020). The parameters that turned out to be the most sensitive, with a Sobol total sensitivity index >10% for at least one type of measurement, were radiation use efficiency (prdx(1); for all measurement types); the optimal and maximum temperature for maize growth (ppdf(1) and ppdf(2), respectively; only for grain yield and aboveground biomass), and maximum harvest index (himax; only for grain yield). Further, the turnover rate of the slow and passive SOM pools (dec5(2) and dec4, respectively; only for SOC and soil N), the decomposition multiplier for soil tillage (clteff(1,2&4); only for SOC and soil N) and the fraction lost as $CO_2$ of the metabolic litter pool (pmco2(1&2), i.e., 1 - CUE); only for SOC and soil N) belonged to the most sensitive model parameters. The parameters of further importance, with a Sobol total sensitivity index >2.5% and <10%, were the minimum value for the impact factor of anaerobic soil conditions (aneref(3); only for SOC and soil N), the scaling factor for potential evapotranspiration (fwloss(4); only for maize grain yield), and the fraction lost as $CO_2$ of the structural litter and lignin pools (ps1co(1&2)&rsplig, i.e., 1-CUE; only for SOC and soil N). The fact that the Sobol 1st order and total sensitivity indexes were similar for most parameters suggested only a limited number of interactions between the parameters identified by the global sensitivity analysis.

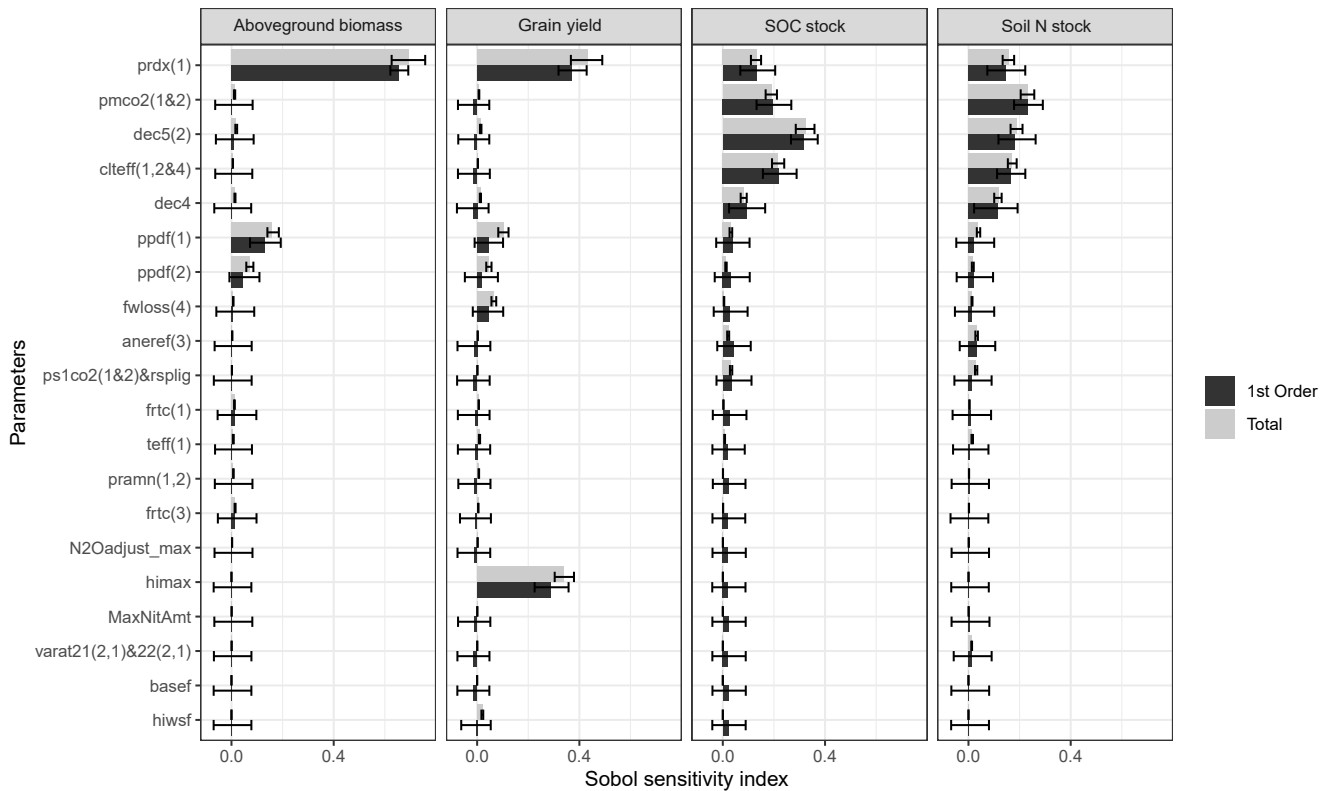

**Figure 1.** Results of the global sensitivity analysis of the most relevant DayCent model parameters. Parameter sensitivity was independently determined for the mean maize aboveground biomass, grain yield, and SOC and soil N stocks at the end of the simulation period. Only parameters with a Sobol sensitivity index >1% are displayed.

## 3.2 Posterior parameter distributions from the Bayesian model calibration

Following the global sensitivity analysis, 13 selected model parameters were calibrated using Gaussian priors which were centered around the initial parameter value, with standard deviations according to the uncertainty ranges (Table 1). It should be noted that the presented calibrated parameter values in Table 1 correspond to the single best parameter set for all four sites combined (i.e., the parameter set that had the highest likelihood in the case of no cross-validation).

Compared to the range of the prior parameter sets, the ranges of the posterior parameter sets calibrated with data from all four sites changed significantly for the parameters fwloss(4) and pmco2(1&2), had a similar mean value but a more narrow distribution for the parameters $IC_{MAOC}$, prdx(1), and ps1co2(1&2)&rsplig, and changed slightly for the parameters dec4, dec5(2), ppdf(1), ppdf(2), and himax (Fig. 2). The posterior parameter sets of the leave-one-site-out cross-validations were largely in agreement with each other and with the posterior parameter sets calibrated with data from all four sites. The exception was the

parameter pmco2(1&2), which was centered around 0.55 for the case that the Aludeka site was left out and around 0.70 for all other cases (Fig. 2).

    The parameter that changed most strongly in the parameter sets calibrated with data from all four sites was the scaling factor for potential evapotranspiration (fwloss(4); from 0.75 to 0.94) thereby not including the initial value in the 95% posterior credibility interval (0.81 to 0.99; Table 1). Also the CUE of metabolic litter was reduced (by an increase of pmco2(1&2) from

0.54 to 0.91 g g$^{-1}$) but the initial value was still within the 95% posterior credibility interval (0.48 to 0.91 g g$^{-1}$). The turnover rates increased for both the slow SOM pool (dec5(2); from 0.10 to 0.13 g g$^{-1}$ yr$^{-1}$) and the passive SOM pool (dec4; from 0.0035 to 0.0060 g g$^{-1}$ yr$^{-1}$), which was however counterbalanced by a reduction of the effect of tillage on decomposition (clteff(1,2,&4); from 10 to 5) and all three of these parameters contained their initial values in the 95% posterior credibility intervals. The maximum harvest index slightly increased (himax; from 0.40 to 0.43 g g$^{-1}$) and so did the potential production

of maize per unit of light interception (prdx(1); from 2.25 to 2.62 g C m$^{-2}$ langley$^{-1}$). Finally, the optimum temperature for maize growth decreased (ppdf(1); from 30 to 28.6 °C), while the maximum temperature for maize growth increased (ppdf(2); from 45 to 47.1 °C). Of the two parameters that translated measured MAOC into SOC in the passive SOM pool, only IC$_{MAOC}$ was altered (from -0.1 to -0.02 g g$^{-1}$) but the initial value was still in the 95% posterior credibility intervals(-0.25 to 0.06 g g$^{-1}$). Overall, the parameter correlations in the posterior parameter set across the four sites were low for soil carbon related

parameters (around 0.4 at maximum), but stronger correlations existed between plant productivity-related parameters (e.g., -0.7 between himax and prdx(1) and 0.58 between ppdf(1) and ppdf(2); Fig. A3).

### 3.3   Simulation of maize grain yields and aboveground biomass at harvest

While the overall variation of maize grain yields across sites and treatments could be captured to some extent with the initial model parameter set, for two sites a negative model efficiency was obtained (Fig. 3). With the leave-one-site-out cross-

validation approach, the model efficiency for maize grain yields at the left-out site improved ubiquitously (i.e., from 0.32 to 0.38 at Aludeka; from -0.04 to 0.15 at Embu; from 0.32 to 0.38 at Machanga, from -0.16 to 0.31 at Sidada, and from 0.36 to 0.50 across all sites) and so did RSME and bias. The same was true for the simulation of aboveground biomass (e.g., from 0.03 to 0.23 across sites; Fig. 4), with the exception of Machanga. Overall, biases in simulated grain yields were mostly eliminated through the model calibration, and biases in simulated aboveground biomass were eliminated at Sidada, reduced at Embu, but

increased at Machanga.

    While DayCent could not capture the full season-to-season variability of grain yields and aboveground biomass, the mean yields and aboveground biomass throughout the simulation period were simulated well for most treatments without the addition of mineral N (Fig. A4). The exception to this was the Embu site, where there was a systematic underestimation of yields in the +N treatments. Nonetheless, DayCent was able to acceptably simulate the variability of grain yields across sites by organic

resource and mineral N treatment (model efficiencies between 0.30 and 0.54; with values for control -N (0.08) and sawdust -N (0.18) being the exception; Fig A7). Interestingly, DayCent poorly distinguished the mean yields and aboveground biomass of treatments with high compared to very high rates of N inputs (i.e., the differences between the different organic resources and the control within the +N treatment). An additional test of the model sensitivity of mean yields to different levels of mineral

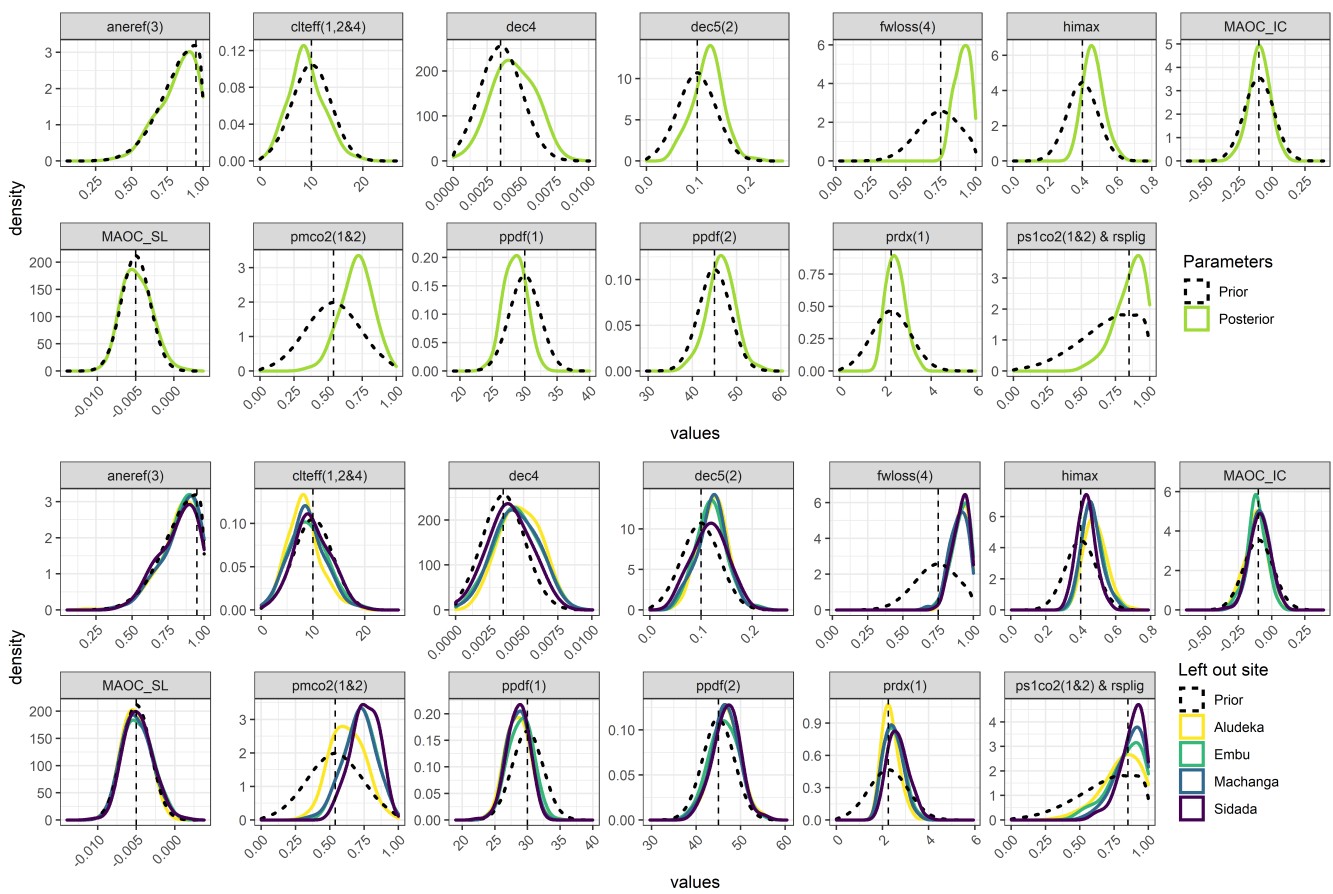

**Figure 2.** Prior compared to the posterior model parameter distribution resulting from the Bayesian model calibration of DayCent using data from all sites combined (top) and the leave-one-site-out cross-validation (bottom). The uncertainty ranges of the priors were based on the range of parameter values found in the literature and increased by a factor of 1.5, because DayCent was applied to tropical site, while historically, it was mostly calibrated based on temperate sites. Dashed vertical lines represent the values of the initially selected parameter set. The posterior distributions are based on all four study sites combined. For the description of the parameters see Table 1.

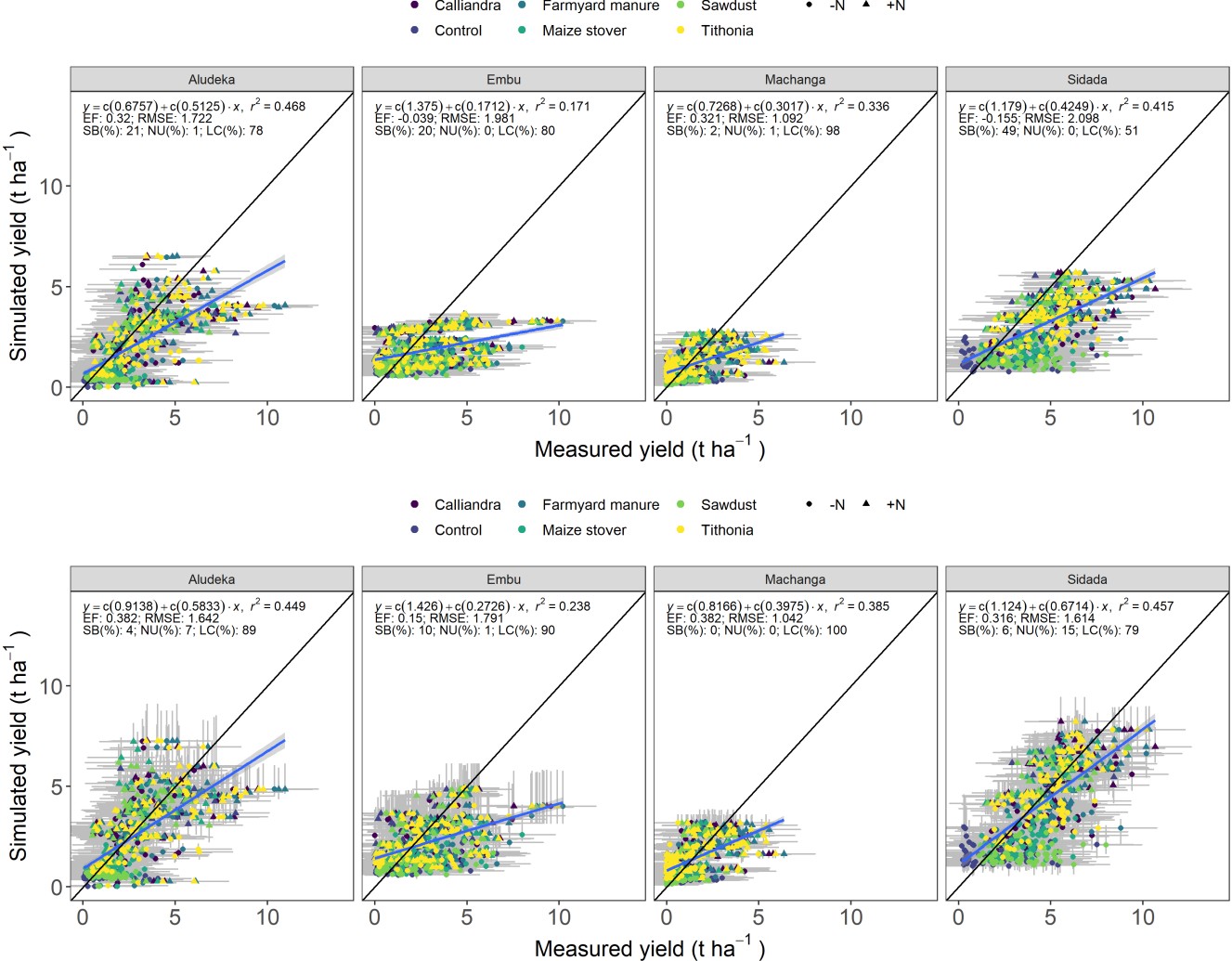

**Figure 3.** Simulated compared to measured maize grain yields at the four study sites for the initial DayCent parameter set (top) versus the calibrated parameter set by leave-one-site-out cross-validation (bottom). The 2985 data points correspond to the observations from the experimental treatments over 32 to 38 seasons, depending on the site. Symbols represent the different organic resource and chemical nitrogen fertilizer treatments. Grey bands show the 95% confidence intervals of measured (horizontal) values and the 95% credibility intervals of posterior distribution (vertical). Abbreviations: EF, Nash-Sutcliffe model efficiency; RMSE, root mean squared error; SB, squared bias; NU, non-unity slope; LC, lack of correlation. Across all sites model statistics: EF, 0.358; RSME, 1.757 t ha[-1]; SB, 21%; NU, 1%; LC, 77% before and EF, 0.503; RSME, 1.545 t ha[-1]; SB, 4%; NU, 2%; LC, 94% after calibration, with 27% of measurements being in the 95% credibility interval of the posterior.

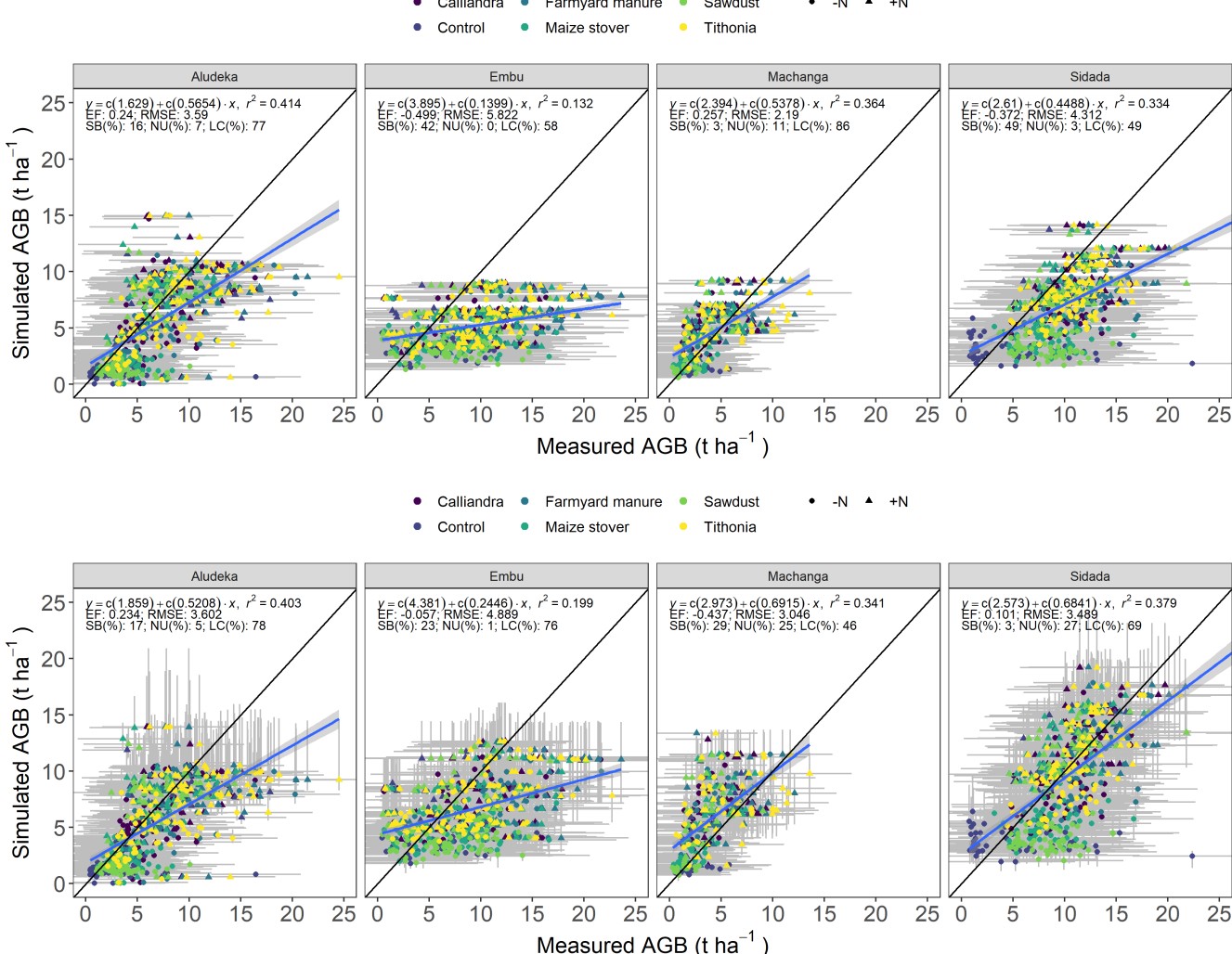

**Figure 4.** Simulated compared to measured maize aboveground biomass (AGB) at the four study sites for the initial DayCent parameter set (top) versus the calibrated parameter set by leave-one-site-out cross-validation (bottom). The 2985 data points correspond to the observations from the experimental treatments over 32 to 38 seasons, depending on the site. Symbols represent the different organic resource and chemical nitrogen fertilizer treatments. Grey bands show the 95% confidence intervals of measured (horizontal) values and the 95% credibility intervals of posterior distribution (vertical). Abbreviations: EF, Nash-Sutcliffe model efficiency; RMSE, root mean squared error; SB, squared bias; NU, non-unity slope; LC, lack of correlation. Across all sites model statistics: EF, 0.033; RSME, 4.392 t ha⁻¹; SB, 27%; NU, 1%; LC, 72% before and EF, 0.231; RSME, 3.915 t ha⁻¹; SB, 7%; NU, 8%; LC, 85% after calibration, with 36% of measurements being in the 95% credibility interval of the posterior.

N fertilizer in the control provided further insights into this (Fig. A5). In this test, the yields plateaued at mineral N rates that were lower than the maximum N rates provided in the organic resource +N treatments by mineral N and organic resources combined (up to >500 kg N per year or > 250kg N per growing season). At Machanga and Embu, simulated mean yields stopped increasing at around 100 kg N ha$^{-1}$ per growing season, which is less the 120 kg N ha$^{-1}$ per growing season in the control +N. At Aludeka and Sidada, simulated mean yields stopped increasing at 200 to 250 kg N ha$^{-1}$ per growing season, but most of the response to N was below 120 kg N ha$^{-1}$ per growing season (Fig. A5). Although the mean yields in -N treatments with the high-quality inputs were well simulated, some of the low-quality input treatments at Aludeka and Sidada, namely maize stover and sawdust at 1.2 and 4 t C ha$^{-1}$ yr$^{-1}$, had lower simulated than observed mean yields in their -N treatments (Fig. 5). The same was true for the control -N at Aludeka, Embu, and Machanga.

## 3.4 Simulated SOC stocks in response to integrated soil fertility management

Similar to the simulation of maize grain yields, the simulations of changes in SOC stocks following the application of organic resources at different rates (1.2 and 4 t ha$^{-1}$ yr$^{-1}$) generally improved across sites by the leave-one-site-out cross-validation approach compared to using the initial model parameter set (Fig. 6). The improvement was even stronger when compared to DayCent simulations with the default CUE value for the structural pool (these had a negative model efficiency at all four sites; Fig. A6). While Aludeka experienced an improved but negative model efficiency for simulated changes in SOC stocks with the leave-one-site-out cross-validation (from -4.17 to -0.90), the model efficiencies at Embu and Sidada were positive in the initial parameter set and improved with calibration (from 0.53 to 0.54 at Embu and from 0.47 to 0.58 at Sidada). Also across sites, the model efficiency (computed without Machanga) improved considerably from 0.36 to 0.55 following calibration. As expected, Machanga, for which the SOC stock data had been removed from the calibration dataset due to soil erosion at this site, still exhibited a poor model efficiency after calibration (-1.9 compared to -4.8 before calibration). DayCent performed well in simulating the variability of the changes in SOC stocks across sites, evaluated by organic resource and mineral N treatment (also computed without Machanga). With the exception of the treatments farmyard manure ±N, maize stover ±N, and sawdust -N model efficiencies were between 0.51 and 0.79 (with RSME between 3.2 and 4.3 t ha$^{-1}$; Fig A8) with the highest performance for the control +N (0.79) and control -N (0.69) treatments. The other treatments still had positive model efficiencies (0.15 to 0.42), but the SOC losses of the farmyard manure treatments were overestimated (EF of 0.15 for -N, 0.29 for +N, RSME of 5.1 and 5.3).

While SOC changes were well captured in the control treatments across all sites, it should be noted that the increases in SOC stocks in the treatment receiving 4 t C ha$^{-1}$ yr$^{-1}$ were overestimated at Aludeka (Fig. A9). As a result, the posterior credibility intervals of simulated SOC stocks matched well with measured SOC stocks of Embu and Sidada, but not with those of Aludeka (Fig. 7). The difference between the 4 t C ha$^{-1}$ yr$^{-1}$ input and the control treatments were generally well simulated, but the fact that Machanga, the other sandy site, could not be used in the calibration due to erosion, likely contributed to the poorer performance of the sandy site in Aludeka.

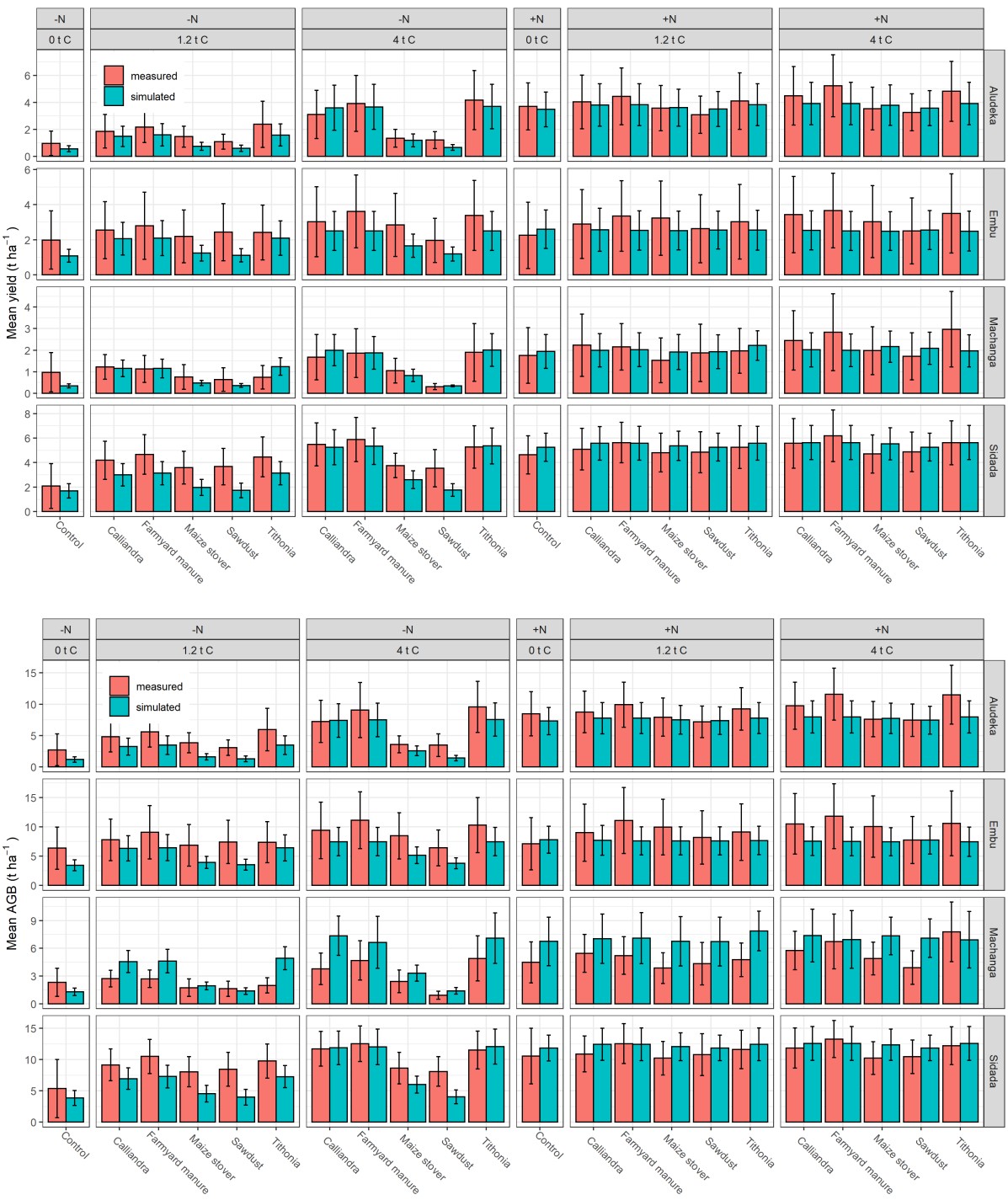

**Figure 5.** Barplots of mean simulated and mean measured maize grain yield and aboveground biomass (AGB) from cross-validation. Error bars represent standard deviation.

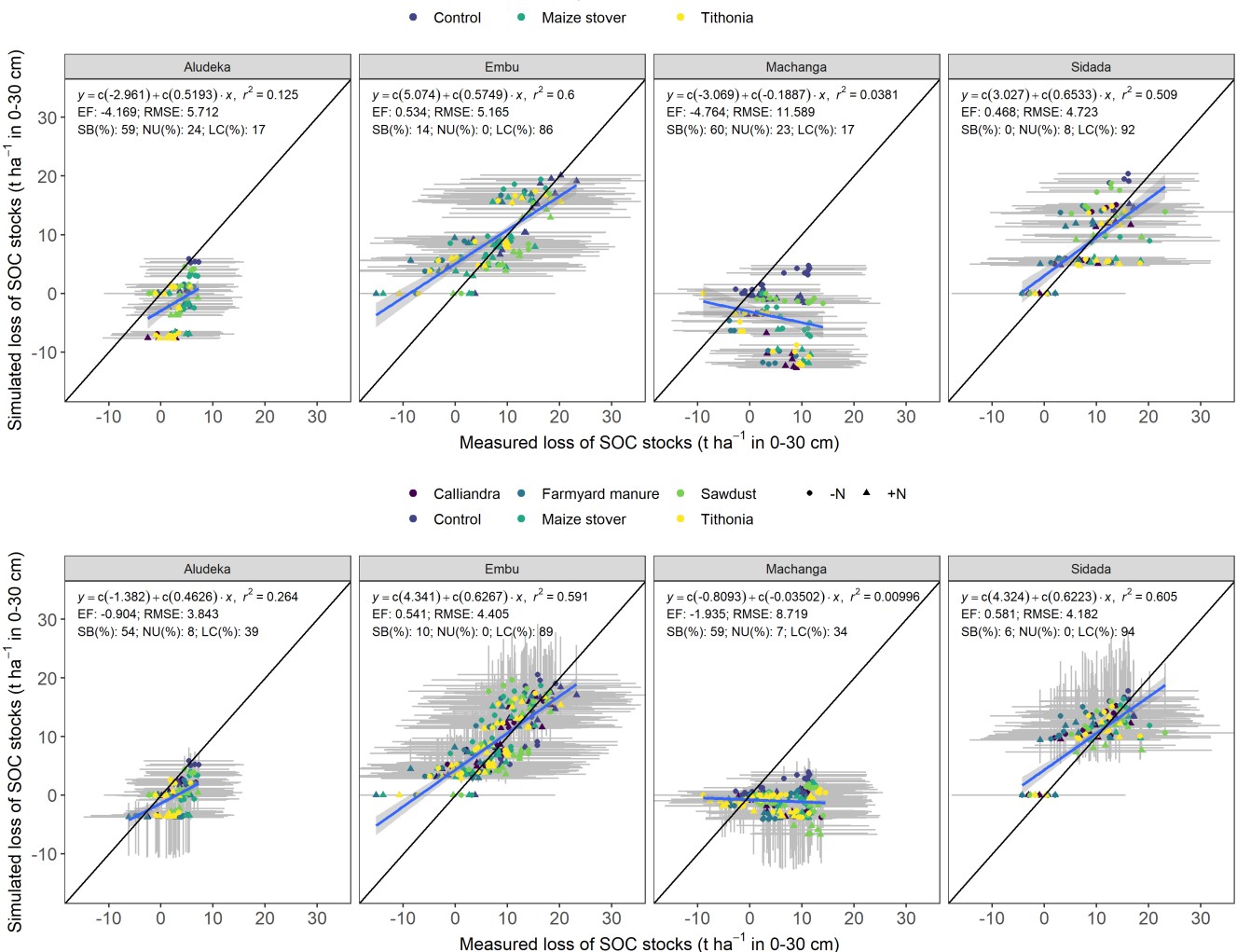

**Figure 6.** Simulated compared to measured changes in SOC stocks since the start of the experiment at the four study sites for the initial Day-Cent parameter set (top) versus the calibrated parameter set by leave-one-site-out cross-validation (bottom). The 724 data points correspond to the observations from the experimental treatments over 32 to 38 seasons, depending on the site. Symbols represent the different organic resource and chemical nitrogen fertilizer treatments. Grey bands show the 95% confidence intervals of measured (horizontal) values and the 95% credibility intervals of posterior distribution (vertical). Abbreviations: EF, Nash-Sutcliffe model efficiency; RMSE, root mean squared error; SB, squared bias; NU, non-unity slope; LC, lack of correlation. Across all sites model statistics without Machanga (from which SOC data was excluded in the calibration process due to strong erosion): EF, 0.364; RSME, 5.199 t ha$^{-1}$; SB, 1%; NU, 22%; LC, 77% before and EF, 0.548; RSME, 4.22 t C ha$^{-1}$; SB, 1%; NU, 11%; LC, 89% after calibration, with 45% of measurements being in the 95% credibility interval of the posterior.

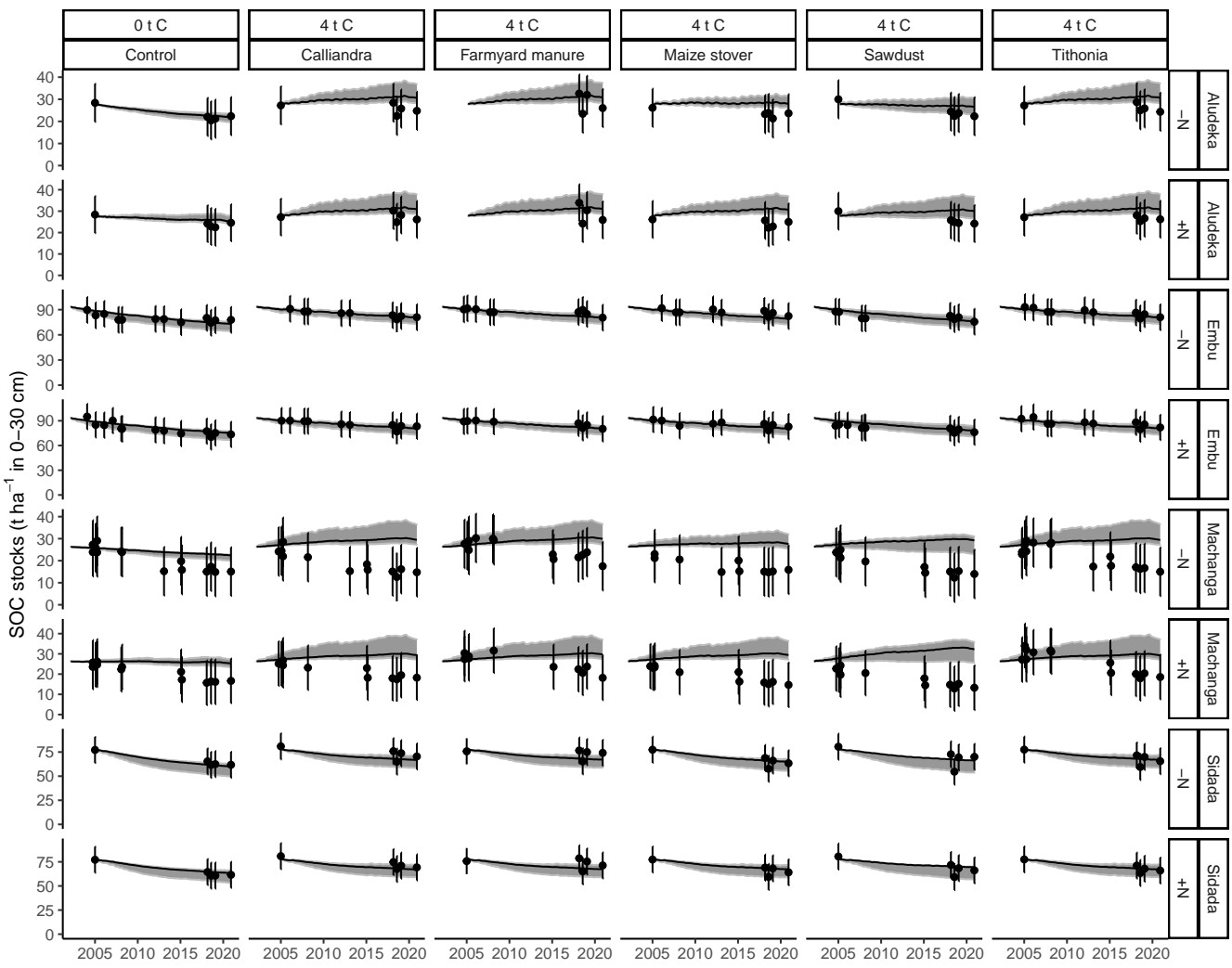

**Figure 7.** Measured (dots) versus simulated SOC stocks over time at the four study sites for the different organic resource and chemical nitrogen fertilizer treatments. Error bars represent 95% confidence intervals for measured data, the black solid line the simulation by the best parameter set. Grey bands represent the 95% credibility intervals of the model posterior simulations, calibrated by leave-one-site-out cross-validation. Note that due to intense soil erosion, data from Machanga was not used in the calibration process.

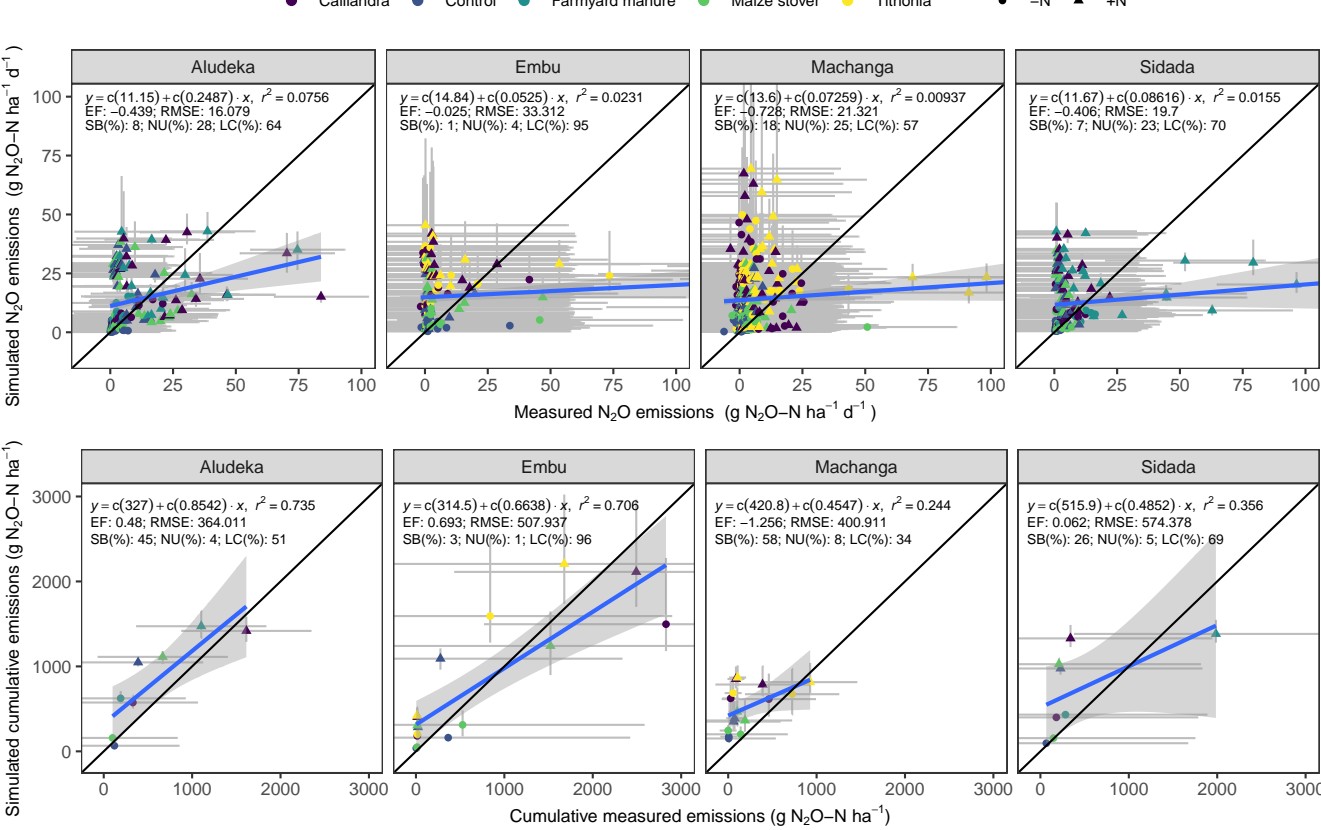

**Figure 8.** Simulated compared to measured N₂O emissions at the four study sites for the different organic resource and chemical nitrogen fertilizer treatments, based on the calibrated parameter set using leave-one-site-out cross-validation. Displayed are the measured versus modelled per treatment for the days where measurements were conducted (top) and for the mean of cumulative flux measurements per season using the trapeziod method (bottom). The 808 data points (top) correspond to the daily measurements from the experimental treatments over one to two seasons, depending on the site. Symbols represent the different organic resource and chemical nitrogen fertilizer treatments. Error bars represent 95% confidence intervals (measurements) and credibility intervals (simulations). Note that the credibility intervals are only informed by yield, SOC and harvest index data and therefore do not represent the full uncertainty of N₂O emissions. Abbreviations: EF, Nash-Sutcliffe model efficiency; RMSE, root mean squared error; SB, squared bias; NU, non-unity slope; LC, lack of correlation.

## 3.5 Simulated N₂O emissions and GHG balance

The negative model efficiencies and the absence of correlation between observed and simulated daily N₂O values indicated that model performance for daily N₂O emissions was poor (Fig. 8). While treatments with higher N loads had both higher simulated and measured N₂O fluxes compared to those with lower loads, the peaks of N₂O emissions were often simulated on different dates than the measurements. This was most noticeable in +N treatments (Fig. A10). Conversely, the simulated cumulative N₂O

emissions per season were in a better agreement with the measured values. All sites, except Machanga, showed positive model efficiencies (highest at Embu, 0.69; lowest at Sidada, 0.06), but generally underestimated the uncertainty around cumulative $N_2O$ emissions (Fig. 8). Additionally, the correlation between simulated and measured $N_2O$ emissions was notably higher for the cumulative emission fluxes than for daily fluxes ($R^2$ of 0.74 for Aludeka, 0.7 for Embu, and 0.36 for Sidada, compared to $R^2$ close to 0 for daily fluxes). Furthermore, despite some bias at Aludeka and Sidada, most of the error in seasonal $N_2O$ emissions was not systematic (i.e., LC of 51 - 96%).

The simulated changes in SOC and seasonal $N_2O$ emissions revealed a positive GHG balance for all treatments at all sites (Fig. 9 a). Yet, the magnitude of emissions, as well as the relative contributions of $N_2O$ and $CO_2$, differed strongly between sites and treatments. For instance, in the control -N treatment, emissions ranged from 1.5 t $CO_2$ equivalent ha$^{-1}$ yr$^{-1}$ at Aludeka to 5 t $CO_2$ equivalent ha$^{-1}$ yr$^{-1}$ at Sidada. The relative contribution of $N_2O$ also differed strongly by site. At Aludeka, for example, all positive GHG balance values in the 4 t C ha$^{-1}$ yr$^{-1}$ treatments receiving farmyard manure, *Tithonia*, and *Calliandra* came from $N_2O$, while SOC acted as a sink of GHG. In contrast, at Sidada and Embu, most treatments had between a third and half of their GHG balance associated with $N_2O$ emissions, with the remainder attributed to SOC losses. Compared to the control -N treatment, all organic resource treatments in the -N treatments were simulated to have lower emissions at inputs of 1.2 t C ha$^{-1}$ yr$^{-1}$ (Fig. 9 b). Yet, when including the 4 t C ha$^{-1}$ yr$^{-1}$ and the +N treatments, the changes ranged from an increase of around 1.5 t $CO_2$ equivalent ha$^{-1}$ yr$^{-1}$ to a reduction of 2 t $CO_2$ equivalent ha$^{-1}$ yr$^{-1}$. Embu was the site where the addition of mineral N (+N treatment) led to the strongest increase in simulated GHG balance compared to the control -N treatment.

Finally, there were site- and treatment-specific differences in the yield-scaled GHG balance. The treatments control -N, maize stover -N and sawdust -N had the highest simulated emissions per kg of maize grain yield across sites (0.8 to 1.8 kg $CO_2$ equivalent per kg of yield). In contrast, the farmyard manure, *Callianda* and *Tithonia* treatments at inputs of 1.2 t C ha$^{-1}$ yr$^{-1}$ in the +N treatment and at 4 t C ha$^{-1}$ yr$^{-1}$ in both -N and +N treatments tended to have the lowest simulated emissions at all sites (around 0.3, 1 and 0.6 kg $CO_2$ equivalent per kg of yield at Aludeka, Embu, and Sidada, respectively).

## 4    Discussion

### 4.1    Robustness of the Bayesian calibration shown by cross-validation

As shown by the leave-one-site-out cross-validation (Figs. 3 and 4), the Bayesian calibration considerably improved the predictive capability of DayCent for maize grain yield, aboveground biomass and changes in SOC stocks across sites. The model evaluation statistics from this calibration were comparable to those reported in recent publications that also combined the predictions of crop yield and SOC (Necpalova et al., 2018; Levavasseur et al., 2021; Nyawira et al., 2021). However, while these studies generally showed a better simulation of crop yield than SOC, our study diverged. We found that while better yield simulations compared to SOC simulations were evident at the Aludeka and Machanga sites with soils of low clay content, the results were different at the Embu and Sidada sites with clay-rich soils. Here, SOC stock changes were more accurately simulated than maize grain yield. This, together with the fact that the simulation of aboveground biomass worsened at two sites as a result of the calibration (Fig. **??**), suggests that no single best parameter set exists for the current version of DayCent to

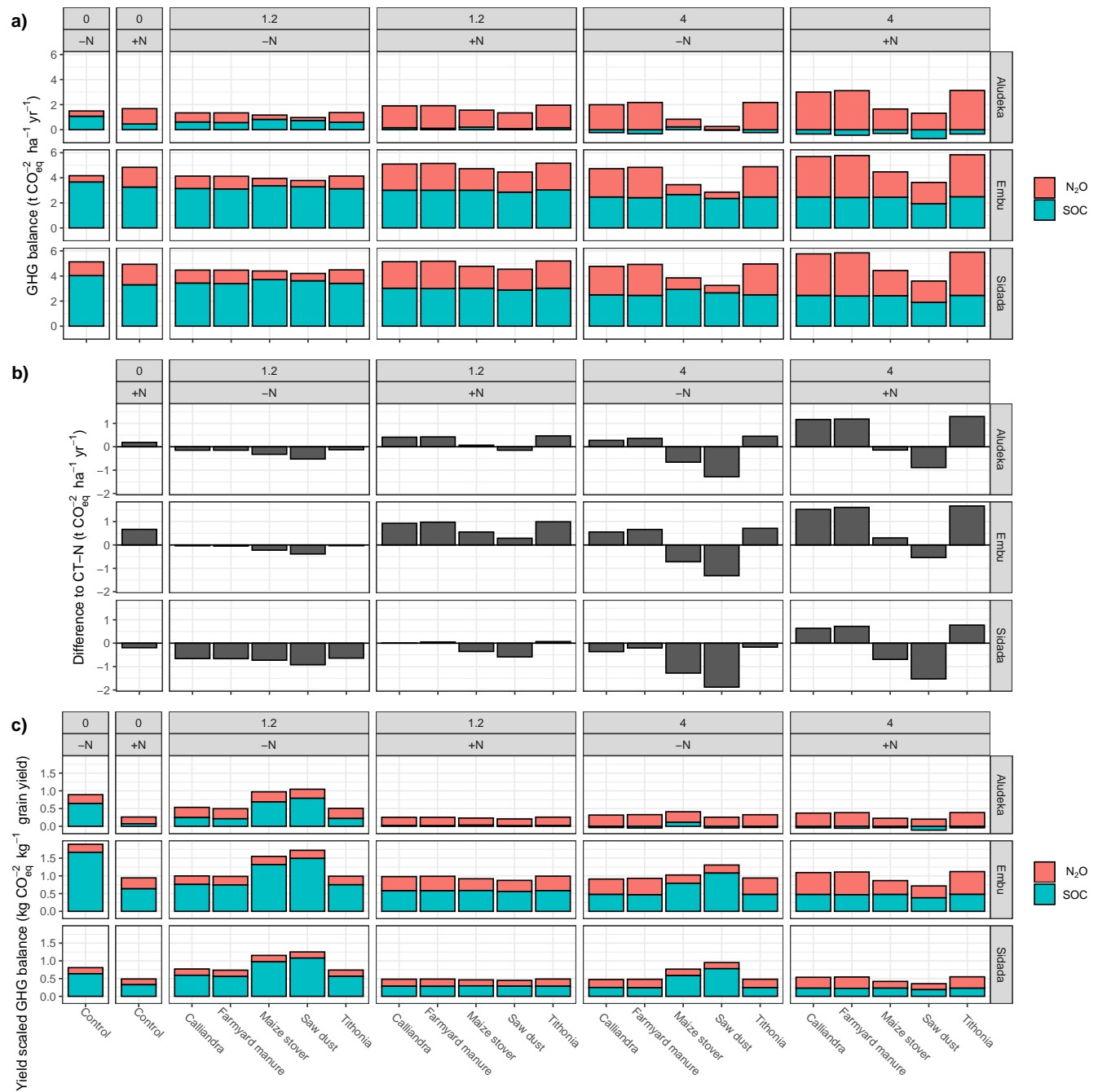

**Figure 9.** Cumulative simulated greenhouse gas (GHG) balance of $N_2O$ emissions and $CO_2$ emissions due to loss of SOC at the four study sites for different organic resource and chemical nitrogen fertilizer treatments combined throughout the simulated period (16 years for Aludeka/Sidada; 19 years for Embu/Machanga). Displayed are the GHG balance a) per area of land and year, b) the difference of GHG balance per area of land and year to a no-input treatment, and c) the yield-scaled GHG balance. The GHG balance is expressed in $CO_2$ equivalent over a 100-year horizon.

accurately represent the conditions at all four sites. In that regard, the discrepancy between the sites with clay-rich and clay-poor soils could indicate that DayCent insufficiently includes soil textures effects on nutrient availability and SOC formation. Yet, drawing definitive conclusions from just four sites is probably not warranted. In the absence of data from more sites, it is preferable to apply the full range of possible parameter sets that are supported by the available data (Mathers et al., 2023), rather than using only the single best parameter set.

Because our calibration shows a good model fit with observed mean yields and changes in SOC stocks across sites, with no overall major bias (positive EF and errors mostly consisting of LC), the parameter set, especially the full posterior, appears suitable for upscaling of model simulations. Specifically, the yields of the ISFM treatments applying farmyard manure, *Calliandra*, and *Tithonia* were simulated well, both with and without the addition of mineral N fertilizer (Fig A7). The changes in SOC stocks for the control, *Calliandra*, and *Tithonia* treatments were also simulated well across sites, while DayCent underestimates the SOC buildup from farmyard manure treatments (Fig A8). However, on should keep in mind that the season-to-season yield variability is captured less accurately than the mean yields (lower RMSE) and that changes in SOC are better represented at sites with clay-rich soils than those with clay-poor soils. Because the model calibration and evaluation were performed at sites with diverse characteristics, it is reasonable to assume that DayCent, when applied to sites with similar climate and soil conditions, will provide satisfactory results with similar model uncertainties and errors. In that respect, while the leave-one-site-out cross-validation made efficient use of data for model evaluation, further model upscaling should apply the full posterior model parameter set including all sites (Fig. 2) should be used. In that case, a computationally inexpensive exercise would use only the single best parameter set (Table 1), while the full posterior parameter set should be used to get estimates of the posterior credibility intervals for changes in SOC stocks.

## 4.2 Bayesian calibration shows uncertainty of model parameters

To estimate the potential yield and long-term sustainability of cropping systems without major bias using biogeochemical models, region-specific model calibrations are needed (Rattalino Edreira et al., 2021; Yang et al., 2021). Therefore, while previous studies have simulated crop productivity under ISFM and similar practices with the default parameter values (e.g. Nezomba et al., 2018; Nyawira et al., 2021), the results of our study underscore the importance of a local calibration, especially when simulations are done with a single parameter set. On the one hand, the similar ranges of the prior and posterior model parameter sets for SOC-decomposition-related parameters (i.e., clteff(1,2&4), dec4, dec5(2)) indicate that the included prior knowledge about DayCent parameters from recent Bayesian calibration studies (Gurung et al., 2020; Yang et al., 2021; Mathers et al., 2023) represent good parameter estimates for a tropical setting. On the other hand, the CUE values of both metabolic litter (centered around 25% with pmco2(1&2) around 0.75) and structural litter (centered around 10% with ps1co2(1&2)&rsplig around 0.9) are low compared to the default values. This indicates the difficulty in stabilizing the organic resource additions into SOM at the tropical soils of these four long-term experiments. Both parameters (pmco2(1&2) and ps1co2(1&2)&rsplig) are even higher than in a recent study by Mathers et al. (2023). However, because these values had reached their predefined upper boundary limit in the study by Mathers et al. (2023), our CUE values might even be representative for temperate and not just for the tropical conditions of Kenya. In general, including prior knowledge about model parameter values from similar studies

substantially improves model performance compared to using default parameter values (e.g., see the poor model performance without including prior knowledge on ps1co(1&2)&rsplig; Fig. A6). In fact, the aligning turnover rates of the slow and passive SOM pools with those derived for temperate conditions (Gurung et al., 2020), indicate that the DayCent temperature function is well suited to handle the faster SOM turnover under tropical conditions.

It is important to note that our sites were under natural vegetation (i.e. forest) or fallow until relatively shortly before the establishment of the experiments (Laub et al., 2023a). Consequently, upon the start of cultivation, erosion and potentially accelerated decomposition (due to soil disturbance) occurred, and SOC has likely not yet reached a new equilibrium with C inputs from maize cultivation. Therefore, C loss is the dominant process occurring at the sites. The good simulations of the strong SOC changes in the control treatments when using MAOC initialized SOM pools, a method not commonly used with DayCent, further supports suggestions to move away from purely conceptual SOM pools (Abramoff et al., 2018; Laub et al., 2024). Such conceptual pools require many assumptions about the initial vegetation and soil conditions (e.g., in the spin-up modelling or estimation of SOM pool distribution). In fact, the high uncertainty about initial vegetation, and time and management since site conversion, was a major reason to move away from the model spin-up and site history run usually typically done with DayCent. Thus, our study provides additional support to modify DayCent, incorporating measurable SOM pools (e.g. Dangal et al., 2022).

Nevertheless, soil property maps are also subject to uncertainty. For example, differences between different SOC maps used in model initialization propagate into differences in the changes in SOC stocks (Zhou et al., 2023). It was shown that uncertainty of the simulated effect of a soil management practice on the difference of SOC stocks compared to a counterfactual is lower than the uncertainty of the simulated temporal development of SOC stocks (Zhou et al., 2023). Therefore, it may be best practice to work with a baseline and an improved scenario. Both spinup and SOC map initialization have their shortcomings and in the end the model user must make an informed decision on which initialization method they consider subject to less uncertainty, based on which data is locally available.

The similarity of our DayCent model calibration with that of Gurung et al. (2020) and earlier studies, despite using different model initialization approaches, indicates the broad applicability of DayCent. It suggests that the SOM turnover and maize traits in DayCent are representative for temperate to tropical conditions. The adjustments made to the values of optimal and maximum temperature for maize growth (ppdf(1) and ppdf(2)) could be attributed to the local maize varieties that are adapted to the higher temperatures in Kenya. For example, Yang et al. (2021) conducted a region-specific Bayesian model calibration of the DayCent maize growing parameters and found ppdf(1) to vary between 26 and 32 °C, a range similar to our posterior. However, the differences in model performance by site shows that the broad representativeness of DayCent comes at the cost of model simplification and site-specific model performance. A main reason for this may be that DayCent model formalisms do not include the latest mechanistic understandings of the role of microbes in SOM decomposition (Laub et al., 2024), and the sorption kinetics of carbon to minerals for SOM protection (Abramoff et al., 2018; Ahrens et al., 2020). Additionally, Daycent does not fully consider that a lot of stabilized SOC is formed by microbes from metabolic and not structural litter (Cotrufo et al., 2013; Kallenbach et al., 2016). For example, it was recently demonstrated that the Millenial model, which includes measurable SOM pools and improved kinetics of carbon sorption better predicts SOC stocks at the global scale than

the CENTURY model, which has conceptual SOM pools (Abramoff et al., 2022). While model calibration can compensate for deficiencies in mechanistic accuracy at a single site (Laub et al., 2024), this is likely not possible across sites with different conditions.

An interesting observation is that while the model bias for the mean maize yield was treatment specific (i.e., the mean yields of +N treatments of farmyard manure at 4 t C ha$^{-1}$ yr$^{-1}$ at all sites and of *Tithonia* at the same rate in all but Sidada, were underpredicted by DayCent), the bias for SOC stocks was mostly site specific (i.e., SOC formation at Aludeka at 4 t C ha$^{-1}$ yr$^{-1}$ was overpredicted). A potential explanation for this site-specific bias for SOC is the fact that DayCent was developed under the paradigm of SOM formation occurring mainly from recalcitrant humic compounds in the soil. Alternatively, it might indicate that soil texture alone is insufficient to explain the mineralogy-driven storage potential of SOC (e.g. Reichenbach et al., 2021; Mainka et al., 2022) or that, because Machanga SOC stocks were not used in the calibration due to erosion, SOC changes at clay-rich sites cannot inform SOC changes at sandy sites like Aludeka. Finally, our model sensitivity test to mineral N inputs suggests that the maize yield bias at high N is due to DayCent's inability to capture yield increases above 100-150 kg N per ha and season at the four sites (Fig. A5); the +N treatments of *Tithonia*, *Calliandra* and farmyard manure at 4 t C ha$^{-1}$ yr$^{-1}$ supplied on average >250 kg N per ha and season. Here, it should be noted that DayCent does not include other potential beneficial effects of organic resource treatments, such as increased pH from farmyard manure application (Xiao et al., 2021; Mtangadura et al., 2017), or improved water infiltration of treatments that maintain SOC stocks compared to those that reduce them.

### 4.3   N$_2$O emissions and GHG balance

In general, the poor match between observed and measured daily N$_2$O emissions (Fig. A10) illustrates the difficulty of simulating the timing of microbial processes, through which nitrate (NO$_3^-$) is converted to N$_2$ and N$_2$O gasses, with models of intermediate complexity such as DayCent. One reason is the poor representation of soil moisture dynamics by the 'tipping bucket' soil water balance approach and that soil gas diffusivity is not explicitly simulated (Zhang and Yu, 2021; Wang et al., 2020). However, the fact that cumulative N$_2$O emissions were better simulated than daily emissions, there was no systematic under- or over-prediction of cumulative N$_2$O emissions, and simulated N$_2$O emissions were within the uncertainty range of measured N$_2$O emissions, demonstrates the suitability of DayCent to represent average N$_2$O emissions with the current calibration. Nonetheless, the fact that the uncertainty around predicted cumulative N$_2$O emissions was lower than the uncertainties of the measurements indicates that the posterior, which was only calibrated with yield, SOC, and harvest index data, underestimates the uncertainty around N$_2$O emission predictions. Thus, although DayCent's simulations of N$_2$O emissions are superior to using emission factor approaches (dos Reis Martins et al., 2022), simulating N$_2$O emissions remains challenging and highly uncertain due to the complexity of the processes involved and their high temporal and spatial variability. Given the limited bias in simulating SOC changes and cumulative N$_2$O emissions showed, the DayCent simulations provide a reasonable first estimate of the GHG balance. Nevertheless, the contributions of N$_2$O emissions to the GHG balance of up to 100% (at Aludeka) and between 10 to 60% (at the other sites; Fig. 9), are subject to high uncertainty, as evident from the measurements.

Despite this unresolved uncertainty, our modeling results show that all ISFM options in a maize monocropping system have a net positive GHG balance, aligning with the prevalent trend of SOC losses in recently established (< 50 years) maize systems

in SSA (Sommer et al., 2018; Laub et al., 2023a). The findings also support the postulate that closing yield gaps in SSA will increase $N_2O$ emissions per area of land (Leitner et al., 2020). However, the large differences in the yield-scaled GHG balance between treatments, such as the 30 to 60% lower yield-scaled GHG balance in the FYM 1.2+N treatment compared to the control-N treatment across the sites, indicate that ISFM has the potential to produce crops with relatively lower GHG emissions than no- or low-input input systems, and that the calculation of emissions per unit of food is preferable to the calculation per land area (Clark and Tilman, 2017). Specifically, the ISFM treatments with low-emissions and high yields, such as FYM 1.2+N, that produces between 2 and 4 t of yield per season at emissions of between 0.3 and 1 kg $CO_2$ equivalent per kg of yield, are a suitable mitigation practice compared to the control treatment with little or no inputs of organic and/or chemical fertilizer. Consequently, sustainable intensification and mitigation of greenhouse gases can go hand in hand.

### 4.4 DayCent is suitable to upscale simulations of "real" ISFM, but has a limited sensitivity to high N inputs

Because mean maize yields across sites were reasonably well represented by the calibrated version of DayCent, it can be used for upscaling to predict the potential impact of ISFM in lowering yield gaps at national levels. However, the plateauing of mean yields at high N loads (Fig. A5) indicates that DayCent may not be suitable for estimating maximum achievable yields (e.g., Ittersum et al., 2016), and should thus be restricted to yield predictions for medium N input levels. Given that the historical rates of N fertilizer application in Kenya are less than 50 kg of N ha$^{-1}$ (World-Bank, 2021a), the model seems suitable to simulate the effect of implementing 'realistic' ISFM practices, which target maximum N use efficiency (Vanlauwe et al., 2010), with N input rates considerably below the maximum N rates used in the field experiments of this study (e.g., 80 kg N per season; Mutuku et al., 2020). At all sites, the prediction of mean maize yields was reasonably well for *Calliandra*, *Tithonia*, and farmyard manure treatments at 1.2 and 4 t C ha$^{-1}$ in the -N treatment, as well as for CT+N, i.e., all treatments that supply N at the desired rate for ISFM. Also, the variability of yields and SOC stock changes per treatment across sites was simulated well for *Calliandra*, *Tithonia*, and farmyard manure in both +N and -N treatments (with changes in SOC being simulated a bit worse in farmyard manure treatments; Figs. A7 and A8). While this shows the general capability of DayCent to simulate differences in yields and SOC changes between sites as a function of organic resource composition, it also shows that DayCent cannot capture the better performance of farmyard manure compared to *Calliandra* and *Tithonia* treatments when only considering C, N and lignin contents. Overall, simulated mean maize yields at medium N-levels are likely representative of the achievable yield through ISFM. In summary, the model calibration seems suitable for assessing the long-term effects of relevant ISFM practices on soil fertility, maize yield, and GHG emissions as well as their trade-offs, given the good representation of mean yield potential and SOC changes by the model. Nevertheless, since year-to-year yield variations were not captured well by DayCent, it remains uncertain how effectively the current model calibration can simulate scenarios of climate change, where temperature and precipitation patterns will become more erratic. In the absence of major pests (which in the experiments were controlled), the variations in seasonal precipitation and temperature are responsible for these differences, and if these are not well represented, the applicability of DayCent beyond the climatic range that it was calibrated for is questionable.

## 5  Conclusions

In this study, we demonstrated the effectiveness of simultaneously calibrating the SOM and plant modules of DayCent to simulate maize productivity and changes in SOC stocks under integrated soil fertility management (ISFM) in Kenya. Using a Bayesian calibration approach, our study shows the importance of using a local calibration and of choosing correct prior values for model parameters. Although the initial DayCent parameterization represented the tropical conditions in Kenya acceptably, the overall model performance for maize grain yield, aboveground biomass, and SOC stock changes was improved after cali-

bration using local data. Furthermore, while parameters related to SOM turnover were comparable to previous studies, a lower carbon use efficiency of applied organic resources (higher values of $CO_2$-loss related parameters) compared to previous studies highlighted the difficulty in building new SOC stocks in the studied tropical soils. Our leave-one-site out cross-validation showed that the calibration-derived parameter set is robust for upscaling the model simulations to larger areas in Kenya, particularly when applying the full posterior parameter set. At the same time, while mean maize grain yields were well simulated,

the year-to-year yield variability raised concerns about the model's ability to capture the short-term effects of climate change adequately. Finally, while no ISFM treatment was predicted to act as a net sink of greenhouse gases, treatments with high and intermediate yields exhibited the lowest yield-scaled emissions.

*Code availability.*  To get the latest version of DayCent, we suggest to contact the developers directly, who in our case kindly provided the latest DayCent version.

*Data availability.*  The data sets used for the calibration of this study are available under the IITA data repository. For SOC: https://doi.org/10.25502/wdh5-6c13/d. For yields and biomass: https://doi.org/10.25502/be9y-xh75/d.

*Author contributions.*  JS, MN and ML designed the modeling exercise. ML summarized the data, conducted the modeling exercise and prepared the original draft. MWMM, DM, RY, SMN and WW managed and maintained the long-term experiments. ML, SMN, MN, WW, MvdB, MC and JS were involved in the various sampling campaigns. MC, MN, BV and JS acquired funding for the long-term experiments.

All co-authors contributed in writing and editing of the final submitted article.

*Competing interests.*  All authors declare that they have no conflict of interest.

*Acknowledgements.*  We want to thank Silas Kiragu, who is responsible for maintaining the experiments at Embu and Machanga site, and John Mukalama, who implemented and maintained the experiments at Aludeka and Sidada. Also, we want to thank John Waruingi for helping

with sample processing over the years, and Dr. Moses Thuita for coordinating the experiments for many years, and Britta Jahn-Humphrey for organizing and overseeing the measurement of most SOC and TN in recent years. Further, we thank Matti Barthel and Maryam Cissé, who measured $N_2O$ emissions, and Fiona Stewart Smith, who conducted the measurements of MAOC fractions. This study was supported by funds from the European Union's Horizon2020 framework (LANDMARC; Grant agreement ID 869367) the Swiss National Science Foundation (SNSF; grant number 172940) and by the DSCATT project "Agricultural Intensification and Dynamics of Soil Carbon Sequestration in Tropical and Temperate Farming Systems" (N° AF 1802-001, N° FT C002181), supported by the Agropolis Foundation ("Programme d'Investissement d'Avenir" Labex Agro, ANR-10-LABX-0001-01) and by the TOTAL Foundation within a patronage agreement. We further acknowledge funding and technical support from the Tropical Soil Biology and Fertility Institute of CIAT (TSBF-CIAT), the International Institute of Tropical Agriculture (ITTA), and ETH Zurich in maintaining the experiments throughout many years. The AI language model "Writefull for Overleaf" has been used to improve the grammar of the manuscript.

## Appendix A: Appendix

 ## A1 Pedotranfer functions to derive the hydraulic parameters

The equations used to calculate the soil hydraulic properties were based on the pedotransfer functions of Hodnett and Tomasella (2002):

$$\theta_r = 0.22733 - 0.00164 \times Sa + 0.00235 \times CEC - 0.00831 \times pH + 1.8 \times 10^{-5} \times Cl^2 + 2.6 \times 10^{-5} \times Sa \times Cl \tag{A1}$$

$$\theta_s = 0.81799 + 9.9 \times 10^{-4} \times Cl - 0.3142 \times BD + 1.8 \times 10^{-4} \times CEC + 0.00451 \times pH - 5 \times 10^{-6} \times Sa \times Cl \tag{A2}$$

 $$ln(\alpha) = -0.02294 - 0.03526 \times Si + 0.024 \times SOC - 7.6 \times 10^{-3} \times CEC - 0.11331 \times pH \tag{A3}$$

$$ln(n) = 0.62986 - 0.00833 \times Cl - 0.00529 \times SOC + 0.00593 \times pH + 7 \times 10^{-5} \times Cl^2 - 1.4 \times 10^{-4} \times Sa \times Si \tag{A4}$$

Here, $\theta_r$, $\theta_s$, $\alpha$, and $n$ are the soil water retention parameters of van Genuchten (1982), *Sa*, *Si* and *Cl* are Sand, Silt, and Clay content (in %), *BD* is the bulk density (t m$^{-3}$) CEC is the cation exchange capacity (cmol kg$^{-1}$), *pH* is the soil pH measured in H$_2$O, and *SOC* is the SOC content (g kg$^{-1}$).

 The wilting point (WP) and field capacity (FC) values were then calculated as

$$WP = \theta_r + \frac{(\theta_s - \theta_r)}{(1 + (\alpha \times |-15000|)^n)^{1-\frac{1}{n}}} \tag{A5}$$

$$FC = \theta_r + \frac{(\theta_s - \theta_r)}{(1 + (\alpha \times |-330|)^n)^{1-\frac{1}{n}}} \tag{A6}$$

K$_S$ was calculated using the Saxton and Rawls (2006) equation, with values of the water retention curve, $\alpha$ and $n$ (van Genuchten, 1982), calculated with the equation from Hodnett and Tomasella (2002):

 $$\lambda = \frac{\ln(FC) - \ln(WP)}{\ln(1500) - \ln(33)} \tag{A7}$$

$$K_S = \frac{1930 \times (\theta_s - FC)^{(3-\lambda)}}{10 \times 60 \times 60} \tag{A8}$$

Here, $\lambda$ is the slope of logarithmic tension-moisture curve and $K_S$ is the saturated water conductivity (cm s$^{-1}$).

## A2 Equations for the global sensitivity analysis

The means across all sites, which were used in the GSA were calculated as follows:

$$Mean = \frac{1}{n}\sum_{j=1}^{n} \frac{\sum_{i=1}^{N} Mod_{ij}}{N} \tag{A9}$$

Here $n$ is the number of sites (4), $N$ is the number of modelled values per site, and $Mod_{ij}$ are the individually modelled values. For aboveground biomass and grain yield, $N$ corresponded to the total number of modelled yields and biomass at all treatments and seasons. For SOC and soil N stock $N$ corresponded to the total number of treatments per site. The reason is that because changes in SOC and soil N stocks are expected to be stronger the longer a simulation lasts, only the stocks from the end of the simulation were used.

## A3 Site and organic resource characteristics

**Table A1.** Locations, soil properties and climatic conditions of the study sites. Soil properties are given for the 0 - 15 cm depth layer. Coordinates are given in the WGS 84 reference system. The table is adopted from Laub et al. (2022) under the creative common license 4: http://creativecommons.org/licenses/by/4.0/.

| Soil characteristics | Embu | Machanga | Sidada | Aludeka |
|---|---|---|---|---|
| Latitude | -0.517 | -0.793 | 0.143 | 0.574 |
| Longitude | 37.459 | 37.664 | 34.422 | 34.191 |
| Initial soil C (%) | 3.1 | 0.8 | 2.6 | 0.7 |
| Initial N (%) | 0.3 | 0.05 | 0.21 | 0.06 |
| Initail bulk density (g cm$^{-3}$) | 1.26 | 1.51 | 1.3 | 1.45 |
| pH (H$_2$O) | 5.43 | 5.27 | 5.4 | 5.49 |
| Sand (%) | 0 | 31.1 | 0.1 | 31 |
| Clay (%) | 59.8 | 13.2 | 55.7 | 13.4 |
| Soil type (FAO, 1998) | Humic Nitisol | Ferric Alisol | Humic Ferralsol | Acrisol |
| Altitude (m)[*] | 1380 | 1022 | 1420 | 1180 |
| Annual rainfall (mm)[*] | 1175 | 795 | 1730 | 1660 |
| Mean annual temperature (°C) | 20.1 | 23.7 | 22.6 | 24.4 |
| Months of long rainy season | 3 - 8 | 3 - 8 | 3 - 9 | 3 - 9 |
| Months of short rainy season | 10 - 01 | 10 - 01 | 10 - 01 | 10 - 01 |

[*]Means calculated based on measured data from 2005 to 2020

**Table A2.** Mean measured chemical characteristics (and 95% confidence intervals) of organic resources applied at all sites. Measurements were available from Embu and Machanga from 2002 to 2004, all sites from 2005 to 2007 and in 2018. Significant differences in residue properties were found between the different organic resources, but not between sites and years. Mean values in a row not sharing any lowercase letter are significantly different from each other (p < 0.05). Abbreviations: n.c. = not classified * according to Palm et al. (2001). The table is adopted from Laub et al. (2023a) under the creative common license 4: http://creativecommons.org/licenses/by/4.0/.

| Measured property | *Tithonia* | *Calliandra* | Maize stover | Sawdust | Farmyard manure |
|---|---|---|---|---|---|
| C (g kg$^{-1}$) | 345$^b$ (333-357) | 396$^c$ (383-409) | 397$^c$ (386-408) | 433$^d$ (416-449) | 234$^a$ (213-255) |
| N (g kg$^{-1}$) | 33.2$^d$ (28.9-38.2) | 32.5$^d$ (28.3-37.3) | 7.2$^b$ (6.5-8) | 2.5$^a$ (2.1-2.8) | 18.1$^c$ (15-21.8) |
| C/N ratio | 12.4$^a$ (10.8-14.1) | 13.6$^a$ (11.9-15.5) | 58.7$^b$ (52.8-65.2) | 199.1$^c$ (174.1-227.7) | 12.3$^a$ (9.9-15.4) |
| P (g kg$^{-1}$) | 2.3$^d$ (1.8-2.9) | 1.1$^c$ (0.8-1.5) | 0.4$^b$ (0.3-0.6) | 0.1$^a$ (0-0.2) | 3.1$^d$ (2.3-3.9) |
| K (g kg$^{-1}$) | 37.2$^c$ (21.2-65.2) | 8.7$^b$ (5-15.3) | 9$^b$ (6-13.5) | 2.8$^a$ (1.6-4.9) | 19.4$^{bc}$ (7.8-48.6) |
| Lignin (g kg$^{-1}$) | 90$^{ab}$ (62-117) | 105$^b$ (77-133) | 48$^a$ (37-60) | 172$^c$ (144-199) | 198$^c$ (154-242) |
| Polyphenols (g kg$^{-1}$) | 19$^c$ (14.9-24.3) | 108.7$^d$ (85.3-138.6) | 11.3$^b$ (9.5-13.6) | 4.9$^a$ (3.8-6.2) | 7.8$^{ab}$ (5.2-11.5) |
| Lignin/N ratio | 2.6$^a$ (1.8-3.7) | 3.1$^{ab}$ (2.2-4.3) | 6.2$^c$ (4.8-8) | 58.3$^d$ (41.1-82.8) | 6.9$^{bc}$ (3.9-12.3) |
| Quality / turnover rate* | High / fast | High / slow | Low / fast | Low / slow | n.c. |
| Class* | 1 | 2 | 3 | 4 | n.c. |
| kg N in 4.0 t C ha$^{-1}$ yr$^{-1}$, -N [+N] | 323 [563] | 295 [535] | 68 [308] | 20 [260] | 324 [564] |
| kg N in 1.2 t C ha$^{-1}$ yr$^{-1}$, -N [+N] | 97 [337] | 88 [328] | 20 [260] | 6 [246] | 97 [337] |

**Table A3.** DayCent model parameters (and feasible ranges) of parameters which were not included in the Bayesian model calibration due to a Sobol total sensitivity index < 1%.

| Parameter | Description | Range width | Units | Initial value | Coefficient of variation | Model file |
|---|---|---|---|---|---|---|
| frtc(2) | C allocated to roots at time frtc(3) without stress | small | fraction of NPP | 0.20 | 0.15 | crop.100 |
| frtc(4) | Max. increase in C going to roots under stress | small | fraction of NPP | 0.10 | 0.15 | crop.100 |
| frtc(5) | Max. increase in C going to roots under stress (maturity) | small | fraction of NPP | 0.10 | 0.15 | crop.100 |
| biomax | AGB at which min. and max. C/E ratios of plant increases | small | g biomass m$^{-2}$ | 700.00 | 0.15 | crop.100 |
| pramx(1,2) | Max. aboveground C/N ratio with biomass > biomax | small | C/N ratio | 125.00 | 0.15 | crop.100 |
| prbmn(1,1) | For computing min. C/N ratio for belowground matter | small | C/N ratio | 45.00 | 0.15 | crop.100 |
| efrgrn(1) | Fraction of above ground N which goes to grain. | small | fraction | 0.75 | 0.15 | crop.100 |
| flig(1,1) | Intercept for annual rainfall effect on lignin content | small | fraction of lignin | 0.12 | 0.15 | crop.100 |
| ppdf(3) | Right curve shape for temperature effect on growth curve | very small | unitless | 1.00 | 0.08 | crop.100 |
| ppdf(4) | Right curve shape for temperature effect on growth curve | very small | unitless | 2.50 | 0.08 | crop.100 |
| favail(1) | Fraction of N available per day to plants | moderate | fraction of N | 0.15 | 0.23 | crop.100 |
| (aneref(1)-aneref(2)) | Rain/ET ratio below which, no effect of anaerobiosis | small | unitless | 1.00 | 0.15 | fix.100 |
| aneref(2) | Rain/ET ratio with max. anaerobiosis effect | moderate | unitless | 3.00 | 0.23 | fix.100 |
| damr(1,1)&(2,1) | Fraction of surface N and soil N absorbed by residue | large | fraction of N | 0.02 | 0.38 | fix.100 |
| damrmn(1) | Min. C/N ratio allowed in residue after direct absorption | moderate | C/N | 15.00 | 0.23 | fix.100 |
| dec1(2) | Max. structural litter turnover | small | g g$^{-1}$ yr$^{-1}$ | 4.90 | 0.15 | fix.100 |
| dec2(2) | Max. metabolic litter turnover | small | g g$^{-1}$ yr$^{-1}$ | 18.50 | 0.15 | fix.100 |
| dec3(2) | Max. active pool turnover | small | g g$^{-1}$ yr$^{-1}$ | 7.30 | 0.15 | fix.100 |
| (decX(2)/decX(1)) | Ratio soil to surface turnover (newly defined parameter) | small | unitless | 1.25 | 0.15 | fix.100 |
| fwloss(1) | Scaling factor; interception & evaporation by biomass | moderate | unitless | 1.00 | 0.23 | fix.100 |
| fwloss(2) | Scaling factor; bare soil precipitation evaporation | moderate | unitless | 1.00 | 0.23 | fix.100 |
| fwloss(3) | Scaling factor; transpiration water loss | moderate | unitless | 1.00 | 0.23 | fix.100 |
| pabres | Residue amount which results in max. direct N absorption | moderate | g C m$^{-2}$ | 100.00 | 0.23 | fix.100 |
| teff(2) | Y location of temperature inflection point (decomposition) | large | unitless | 11.75 | 0.38 | fix.100 |
| teff(3) | Step size of temperature effect on decomposition | moderate | unitless | 29.70 | 0.23 | fix.100 |
| teff(4) | Inflection point slope of temperature effect (decomposition) | very large | unitless | 0.25 | 0.45 | fix.100 |
| varat11&12(1,1) | Max. C/N ratio for material entering active pool | small | C/N | 20.00 | 0.15 | fix.100 |
| varat11&12(2,1) | Min. C/N ratio for material entering active pool | small | C/N | 3.00 | 0.15 | fix.100 |
| varat21&22(1,1) | Max. C/N ratio for material entering slow pool | small | C/N | 20.00 | 0.15 | fix.100 |
| varat3(1,1) | Max. C/N ratio for material entering passive pool | small | C/N | 13.00 | 0.15 | fix.100 |
| varat3(2,1) | Min. C/N ratio for material entering passive pool | small | C/N | 6.00 | 0.15 | fix.100 |
| drain | Fraction of excess water lost by drainage | moderate | fraction of H$_2$O | 0.80 | 0.23 | site.100 |
| dmp_st | Damping factor for calculating soil temperature by layer | large | unitless | 0.01 | 0.38 | sitepar.in |
| N2Oadjust_(max-min) | Proportion of nitrified N that is lost as N$_2$O (difference) | large | fraction of N | 0.003 | 0.38 | sitepar.in |
| Ncoeff | Min water/temperature limitation coefficient (nitrification) | large | unitless | 0.03 | 0.38 | sitepar.in |
| dmpflux | The damping factor for soil water flux | large | unitless | 0.00 | 0.38 | sitepar.in |
| astlig_TD | lignin fraction content of organic matter | small | g g$^{-1}$ biomass | 0.09 | 0.15 | omad.100 |
| astrec(1)_TD | C/N ratio of added organic matter | very small | C/N ratio | 12.40 | 0.08 | omad.100 |
| astlig_CC | lignin fraction content of organic matter | small | g g$^{-1}$ biomass | 0.10 | 0.15 | omad.100 |
| astrec(1)_CC | C/N ratio of added organic matter | very small | C/N ratio | 13.60 | 0.08 | omad.100 |
| astlig_MS | lignin fraction content of organic matter | small | g g$^{-1}$ biomass | 0.05 | 0.15 | omad.100 |
| astrec(1)_MS | C/N ratio of added organic matter | very small | C/N ratio | 58.70 | 0.08 | omad.100 |
| astlig_SD | lignin fraction content of organic matter | small | g g$^{-1}$ biomass | 0.17 | 0.15 | omad.100 |
| astrec(1)_SD | C/N ratio of added organic matter | very small | C/N ratio | 199.10 | 0.08 | omad.100 |
| astlig_FYM | lignin fraction content of organic matter | small | g g$^{-1}$ biomass | 0.20 | 0.15 | omad.100 |
| astrec(1)_FYM | C/N ratio of added organic matter | small | C/N ratio | 12.30 | 0.15 | omad.100 |

## A1   Map of the four study sites

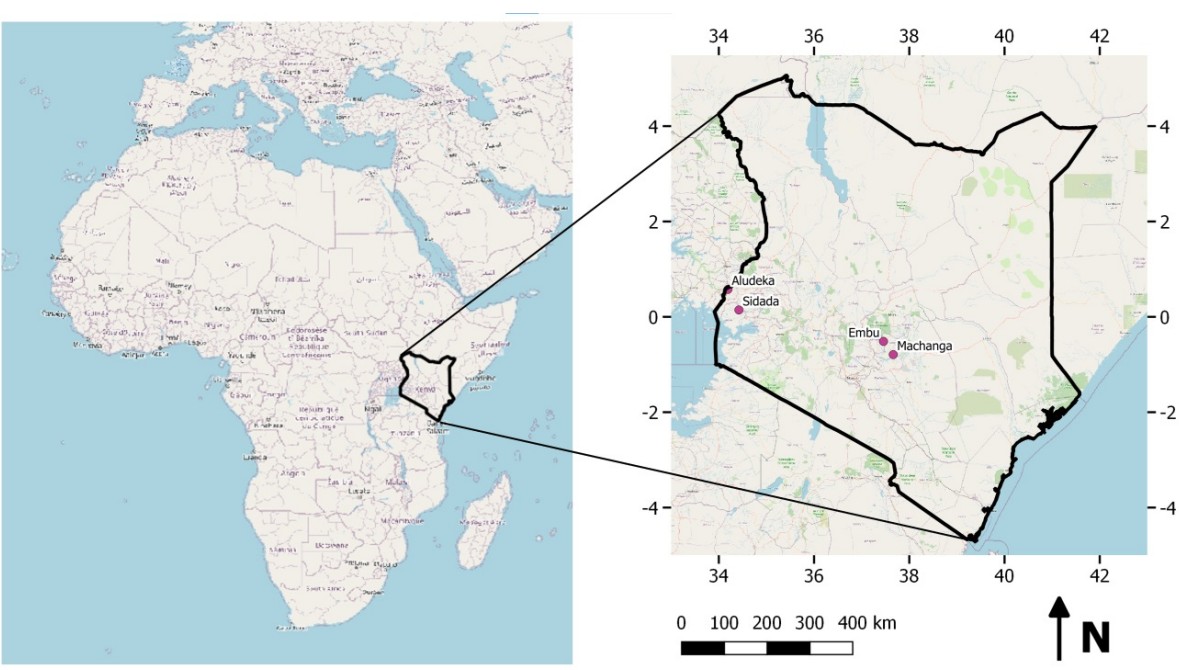

**Figure A1.** Map displaying the location of the four study sites. The background map is based on OpenStreetMap under the Open Database License.

## A2 Subsoil SOC stocks for scaling SOC

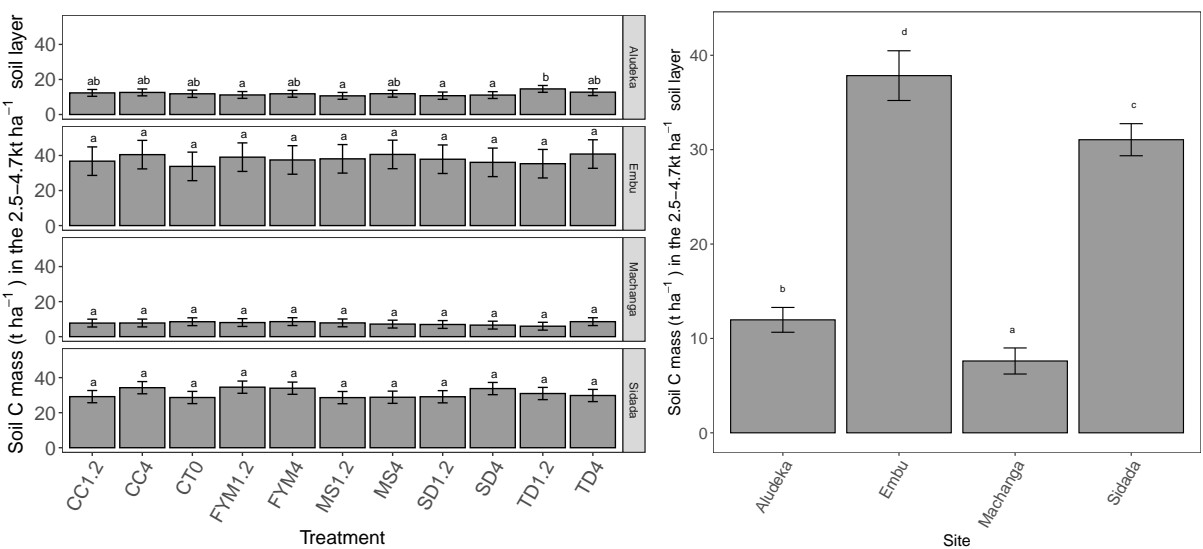

**Figure A2.** Subsoil SOC stocks for the 2.5-4.7 kt ha[-1] equivalent soil mass layer, corresponding to an approximate soil depth of 15-30 cm. Displayed are the least square means estimated by the linear mixed model described in (Laub et al., 2023a) for planted plots by treatment (left) and site (right). Error bars display the 95% confidence intervals. Mean values at each site not sharing any lowercase letter are significantly different from each other (left figure). In the right figure, mean values per site not sharing any lowercase letter are significantly different from each other (all $p < 0.05$). Abbreviations: CC, *Calliandra*; CT, control; FYM, farmyard manure; MS, maize stover; SD, sawdust; TD, *Tithonia Diversifolia*. 0, 1.2 and 4 correspond to C additions of 0, 1.2 and 4 t C ha[-1] yr[-1].

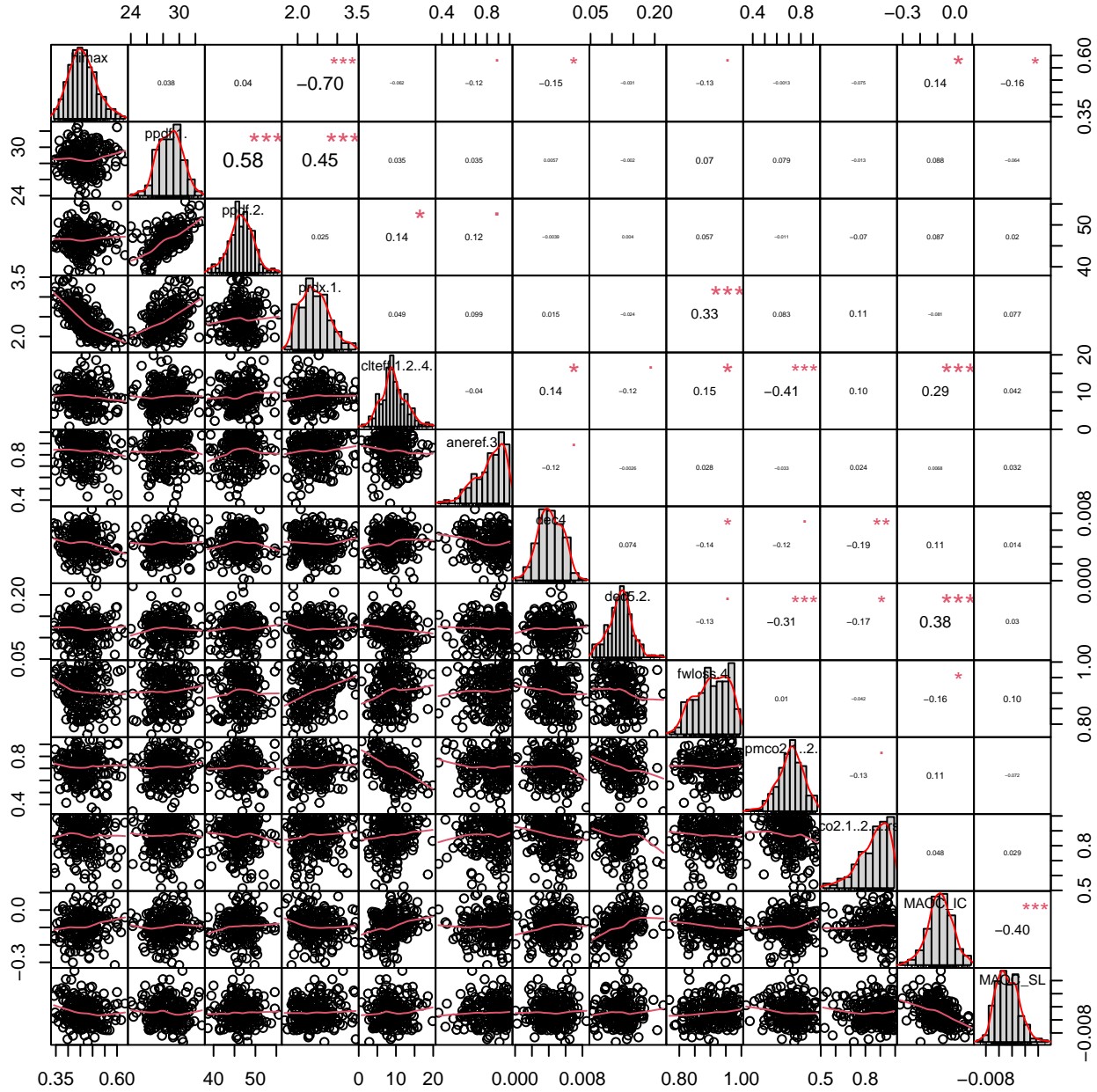

**Figure A3.** Correlation of parameters from the posterior parameter sets. The posterior distributions are based on all four sites combined.

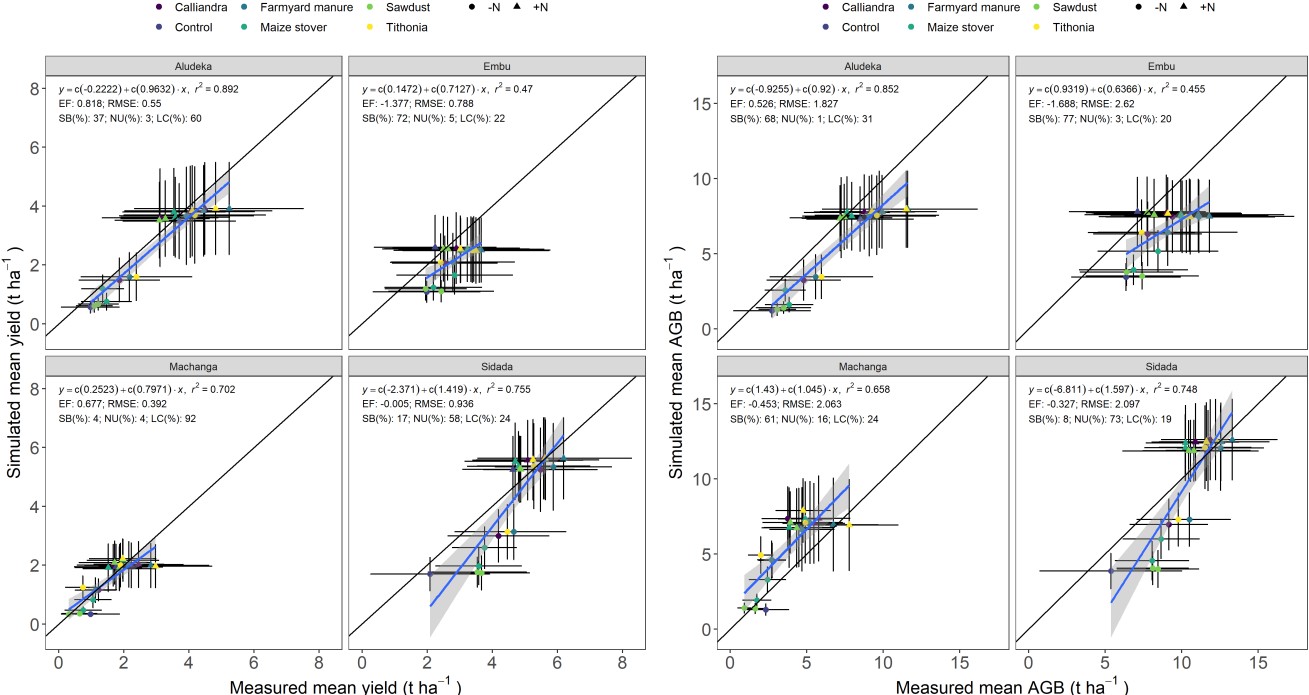

**Figure A4.** Mean simulated versus measured yield and aboveground biomass (AGB) from the leave-one-site-out cross-validation. Error bars represent the standard deviation of measured and simulated values over all years. Abbreviations: EF, Nash-Sutcliffe model efficiency; RMSE, root mean squared error; SB, squared bias; NU, non-unity slope; LC, lack of correlation. Across all sites model statistics: EF, 0.760; RSME, 0.699 t ha$^{-1}$; SB, 28%; NU, 8%; LC, 64% for yield; EF, 0.513; RSME, 2.17 t ha$^{-1}$; SB, 10%; NU, 9%; LC, 81% for AGB.

## A3    Comparing measured and simulated mean yield

 **A4    Site specific sensitivities of yield to N fertilizer**

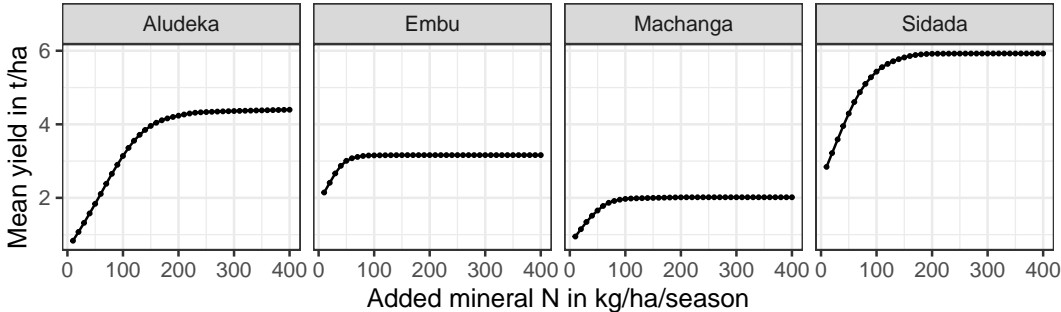

**Figure A5.** Yield response curve of DayCent to varying levels of mineral N application (control + N treatment, without organic resources) using the calibrated DayCent parameters. Displayed are the simulated mean yields across all simulated seasons (32 at Sidada and Aludeka, 38 seasons at Embu and Machanga). The amount of mineral N applied per season in the simulations was evenly split between the actual application dates of mineral N in each season at each site.

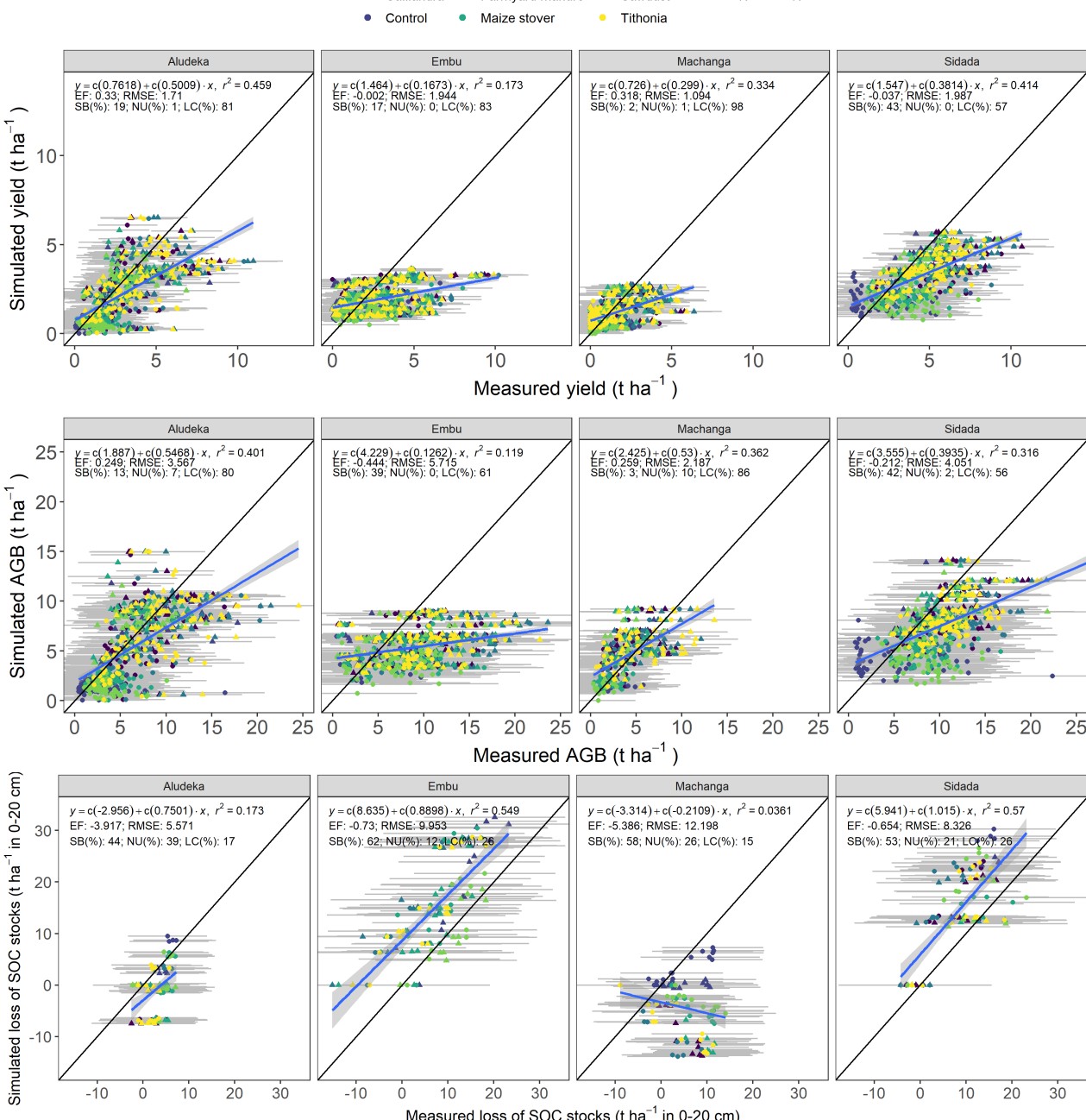

**Figure A6.** Simulated compared to measured maize grain yields, abovoground biomass and change in SOC stocks at the four study sites for the default DayCent parameter set before adjusting ps1co(1&2)&rsplig from 0.5 to 0.85. Grey bands show the 95% confidence intervals of measured (horizontal) values and the 95% credibility intervals of posterior distribution (vertical). Abbreviations: EF, Nash-Sutcliffe model efficiency; RMSE, root mean squared error; SB, squared bias; NU, non-unity slope; LC, lack of correlation.

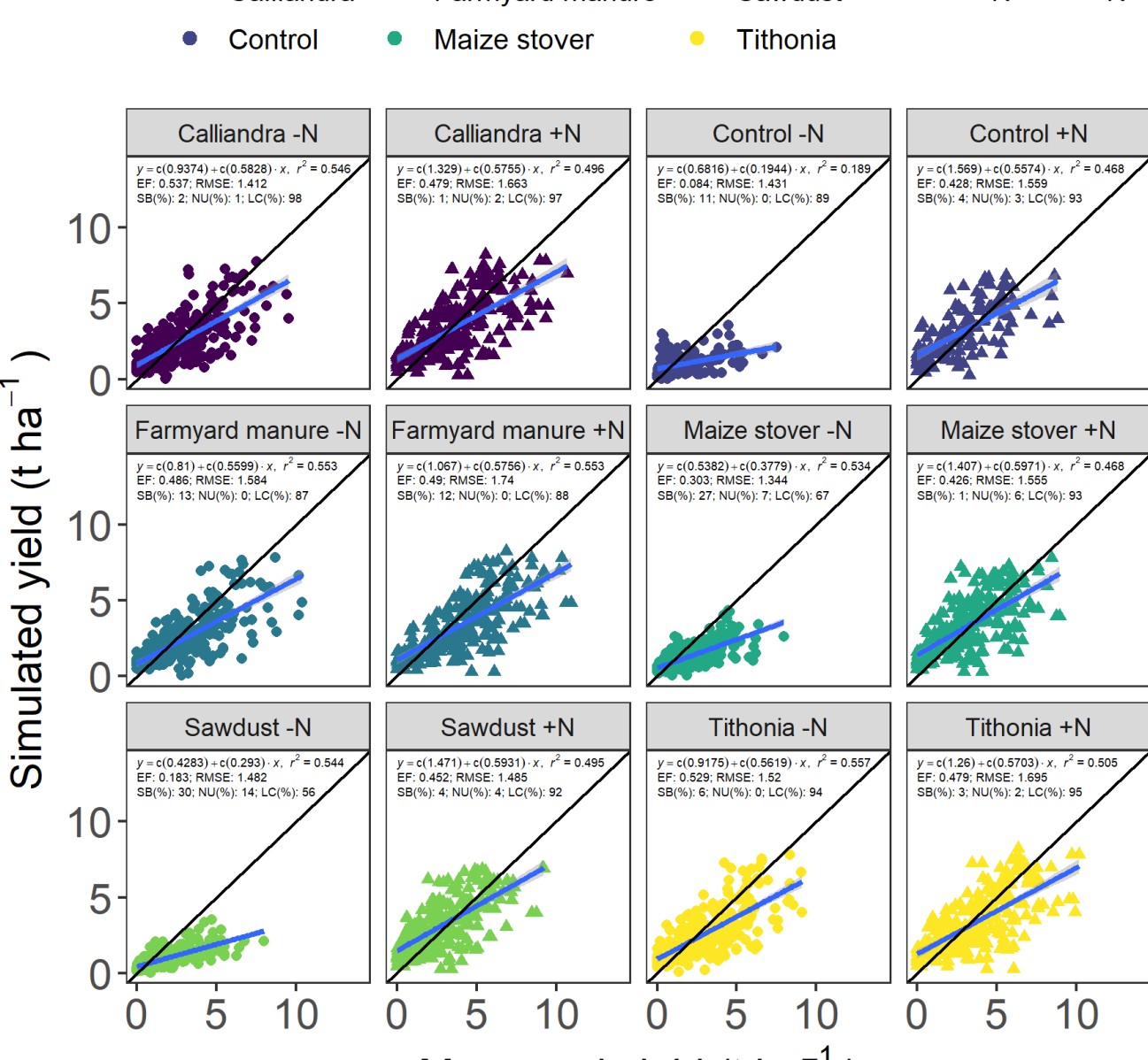

**Figure A7.** Treatment-specific simulated compared to measured maize grain yields at the four study sites for the calibrated parameter set by leave-one-site-out cross-validation. Abbreviations: EF, Nash-Sutcliffe model efficiency; RMSE, root mean squared error; SB, squared bias; NU, non-unity slope; LC, lack of correlation.

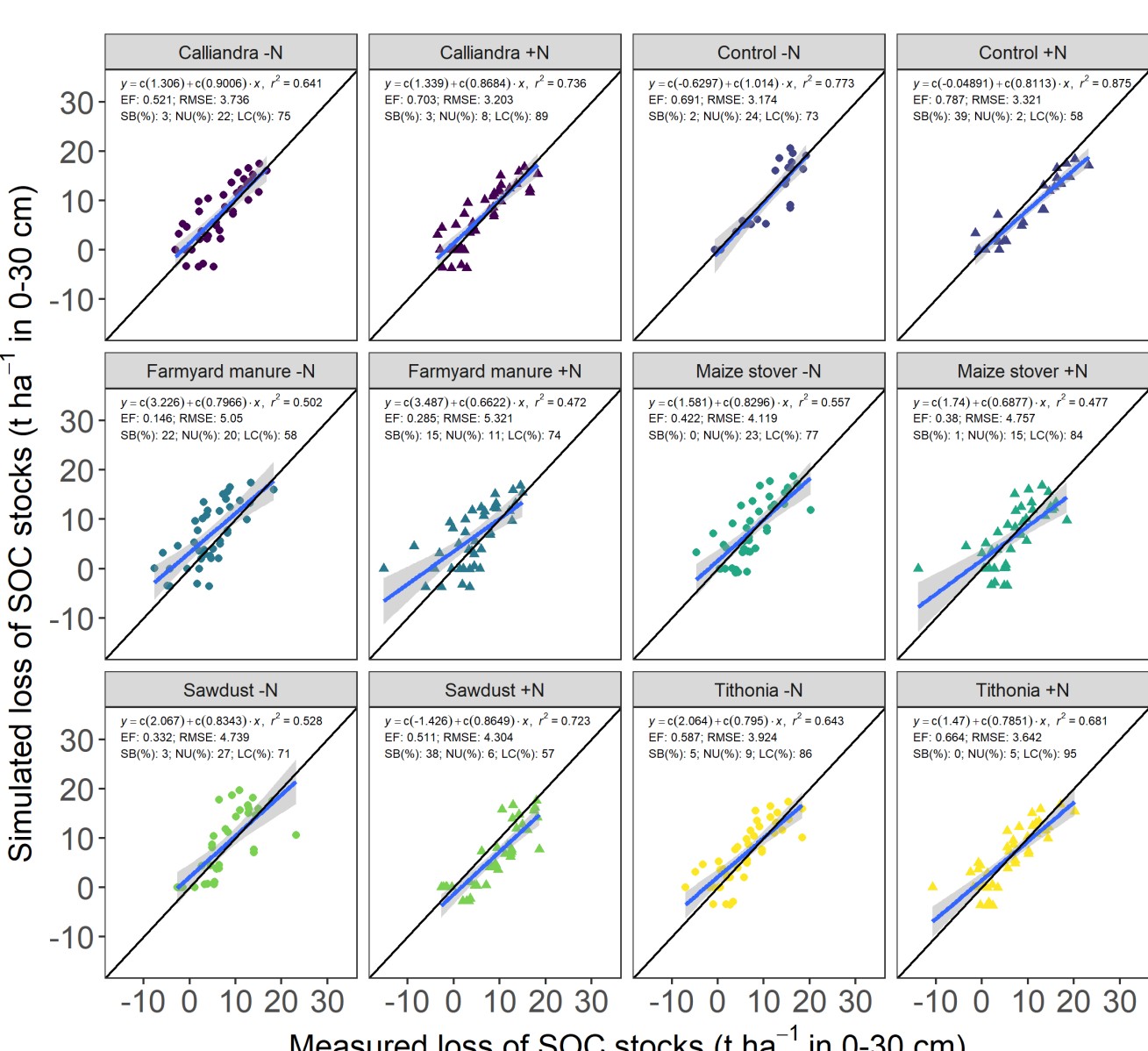

**Figure A8.** Treatment-specific simulated compared to measured changes in SOC stocks (without the Machanga site) since the start of the experiment at the four study sites for the calibrated parameter set by leave-one-site-out cross-validation. Abbreviations: EF, Nash-Sutcliffe model efficiency; RMSE, root mean squared error; SB, squared bias; NU, non-unity slope; LC, lack of correlation.

## A5 Barplots of SOC

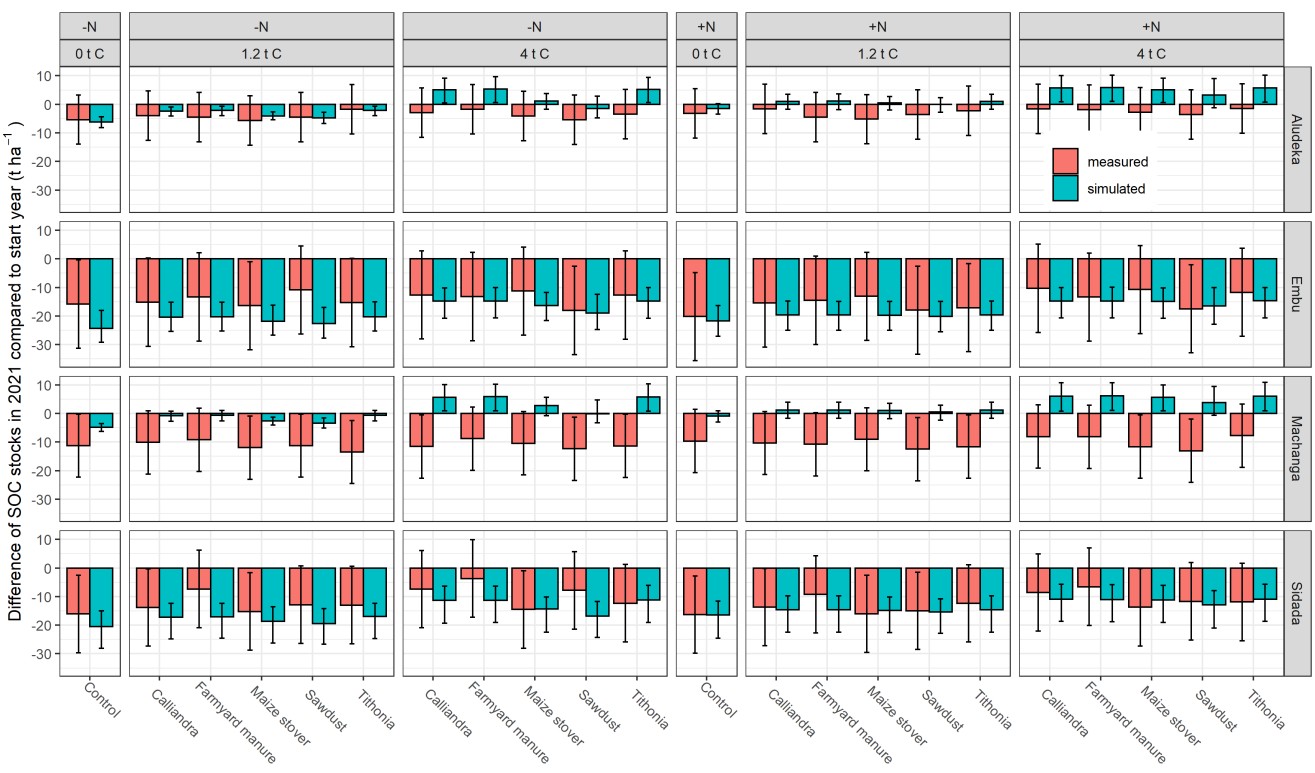

**Figure A9.** Barplots of simulated and measured change of SOC stocks (0-30 cm depth) until 2021 from cross-validation, at the four study sites for the different organic resource and chemical nitrogen fertilizer treatments. Error bars represent 95% confidence intervals based on BC (simulations) and variance (measurements).

## A6  N₂O emissions

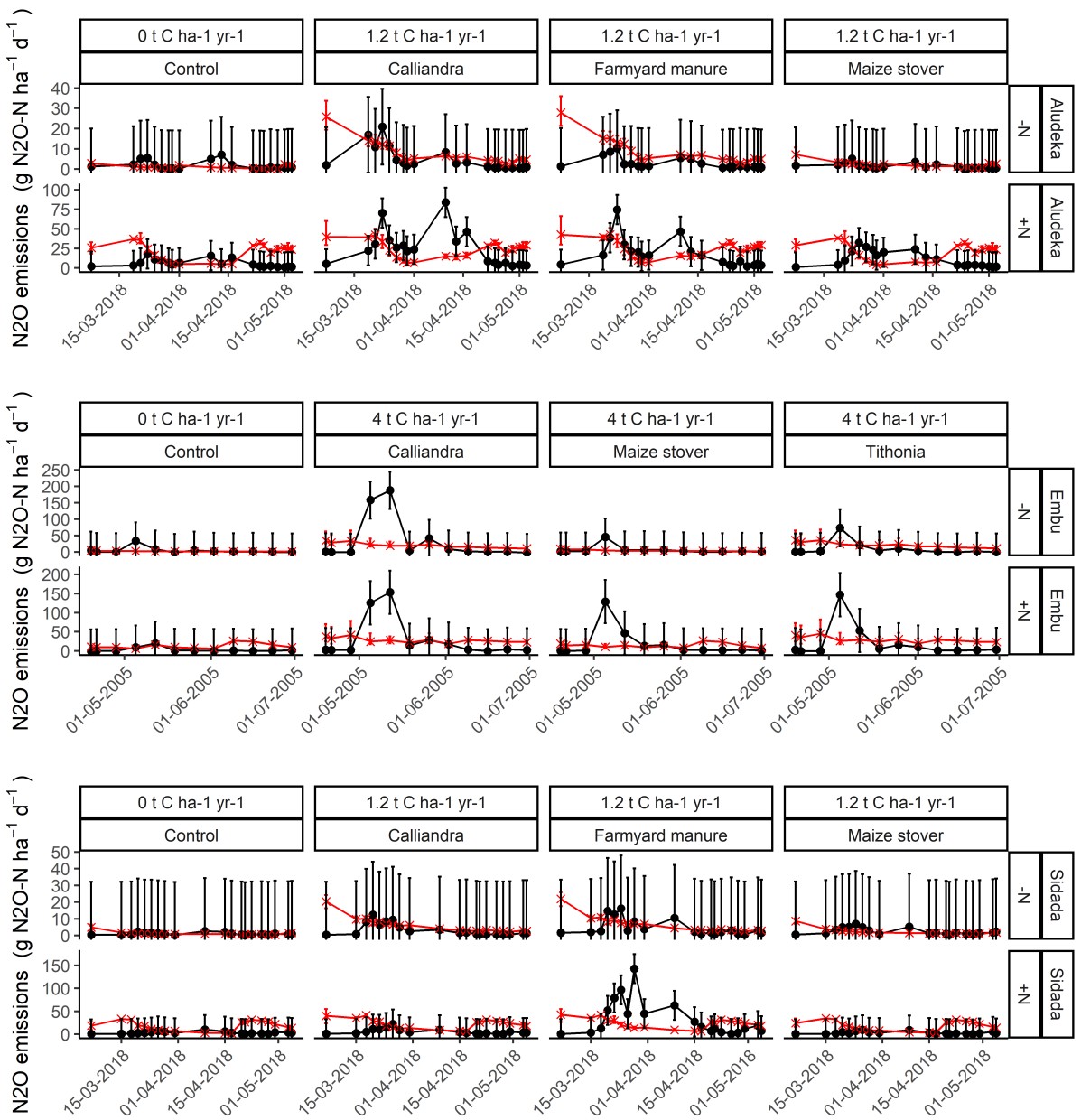

**Figure A10.** Example of the temporal development of measured (black) vs simulated (red) N₂O emissions by site. The black error bars represent the 95% confidence intervals due to spatial replication error, the red error bars represent the 95% credibility intervals of simulated N₂O emissions resulting from parameter distribution of the posterior parameter set.

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
