# Peer review of "A robust DayCent model calibration to assess the potential impact of integrated soil fertility management on maize yields, soil carbon stocks and greenhouse gas emissions in Kenya."

_EGUsphere, 2023_

## Referee Comment (RC1)

The paper describes the capability of DayCent model to simulate yield and SOC development of the different ISFM practices in SSA and its improvement after cal-val. So as presented, the paper is quite long and verbose, resulting quite hard to follow. The figures do not follow a chronological order and are often hard to interpret (see fig. A5). While authors report in M&M a wide description of parameters selection and initialization values which is appropriate and detailed, results are not very clear, often reporting average data which do not highlight the model's ability to reproduce the different selected managements. Also, the mismatch in N2O simulations make hard accounting the GWP here reported. Based on these premises, I recommend a major revision before to be acceptable for publication.

Comments:

L118: …CH oxidation4. Typo.

L241-243: As authors state, DayCent needs to initialize the SOM pools to equilibrium using the typical input of biomass of the native vegetation. However, simulating native vegetation in SSA is not plausible since it is characterized by tropical evergreen forest, dry savanna and humid savanna that, with the only exception of savanna systems which was partly simulated in literature using the grass and tree layers, DayCent is not able to well simulate forest production (Gathany and Burke, 2012). Also, to my knowledge, DC was never tested over tropical environments. Authors should better explain what they used as vegetation for model spin-up.

L335: Authors should consider replacing the term GWP with GHG balance. Despite the likely low effect of CH4, the model is not able to predict CH4 emissions, that therefore they cannot be considered in the whole balance. In this context, would be better to define the GWP as GHG balance since, in any case, the contribution of CH4 cannot be measured neither excluded.

L338: Figure 1 is included in M&M, please move below in Results.

L390-393: Authors can remove this part since calibration is widely recognized to improve model performances.

L394: ….and for aboveground biomass for all sites except Machanga. You mean Aludeka?

L399-401: please, when cited into the main text, report the supplementary figures in chronological order (why A9 before A4, etc…?). Also, why fig.4-5-6 in paragraph 3.5? It's quite hard to follow this flow….

Major weaknesses:

a) In Fig. 3 authors reported all together sites and management for comparing not vs calibrated model. To my opinion, this representation of model calibration is misleading. Firstly, looking at the performances for each site (Fig. A9), model calibration only little improve the model performances found using default values, with statistics confirming the improvement is quite low and lower for each site compared to when assessed overall. This confirm that averaging all sites make unclear to evaluate the model performances under different conditions. Also, it is not clear the ability of the model to reproduce different type of management after calibration process (Fig. A5 is poorly readable, and statistics should be reported. From a visual analysis, variability seem not well simulated). So, from the whole study, does not clearly emerge how the model is able to reproduce yield and AGB for each ISFM at each site. This do not allow to discuss why model does or does not work at each site and for each

management, which could be the limitations and weaknesses, which should be the best practice to use and its response at each site. Averaging all yield data does not clarify the efficiency of the model to be suitable as tool to assess the potential of specific ISFM management practices (as stated by authors in introduction) to cope with food insecurity or further issues. Authors should revise all this part to provide a more accurate response to what they stated in the introduction.

b) The GWP discussion is another major point of weakness. Results clearly showed as N2O is not well simulated neither at daily scale (Fig. A10) nor as cumulated (Fig. 7). Despite in discussions authors state that simulated N2O emissions were generally reasonably well predicted with this current DayCent calibration, looking at Fig. 7 emerged as at Aludeka and Embu the measured N emissions were more than double than those simulated. This clearly affect GWP analysis, especially considering the role of N in GWP analysis, thus making these results very uncertain. Authors should exclude GWP analysis from this study or should much better calibrate the N response to better fit with observations, otherwise GWP discussion risk to be highly speculative due to low level of confidence in N emission outcomes.

---

## Author Comment (AC1)

Reviewer 1:

The paper describes the capability of DayCent model to simulate yield and SOC development of the different ISFM practices in SSA and its improvement after cal-val. So as presented, the paper is quite long and verbose, resulting quite hard to follow. The figures do not follow a chronological order and are often hard to interpret (see fig. A5). While authors report in M&M a wide description of parameters selection and initialization values which is appropriate and detailed, results are not very clear, often reporting average data which do not highlight the model's ability to reproduce the different selected managements. Also, the mismatch in N2O simulations make hard accounting the GWP here reported. Based on these premises, I recommend a major revision before to be acceptable for publication.

Thank you for your critical feedback. As a response to your concerns, we will do further model runs, reconsider individual figures (e.g. A5) for their interpretability and present clearer results with respect to the model's ability to reproduce the different selected managements, also displaying results per site in the main text. We will shorten the text, where possible. We will also reconsider if it makes sense to exclude $N_2O$ from the GHG emissions for comparing treatments after a recalibration. See below our detailed responses to individual comments.

Comments:

L118: …CH oxidation4. Typo. Thanks for spotting this. It was corrected.

L241-243: As authors state, DayCent needs to initialize the SOM pools to equilibrium using the typical input of biomass of the native vegetation. However, simulating native vegetation in SSA is not plausible since it is characterized by tropical evergreen forest, dry savanna and humid savanna that, with the only exception of savanna systems which was partly simulated in literature using the grass and tree layers, DayCent is not able to well simulate forest production (Gathany and Burke, 2012). Also, to my knowledge, DC was never tested over tropical environments. Authors should better explain what they used as vegetation for model spin-up.

We agree that the spin-up is very uncertain for DayCent and for other similar models in general (and not just in SSA, but in general), and it was also raised as an issue by reviewer 2. Data on the history of land use is usually difficult to get in good quality (if any information is available at all), especially in SSA. This is why Mathers et al. (2023) have switched to using the spin-up and historical runs only for the distribution of total C among the different SOC pools (www.doi.org/10.1016/j.geoderma.2023.116647). However, even this comes with a lot of uncertainty regarding the real biophysical conditions and human interactions, so measured pools would in fact be best.

We therefore decided that we will eliminate the model spin-up completely – relying instead on a measured proxy of mineral-associated organic carbon (fraction of SOC that is MAOC; i.e., g MAOC g$^{-1}$ SOC). These (unpublished) data have been measured on soil samples collected in 2021 in the framework of a master thesis in our group (the mean across treatments was 0.91, 0.88, 0.85, 0.86 g MAOC g$^{-1}$ SOC for Aludeka, Embu, Machanga, and Sidada in 0-30 cm, respectively, with no significant treatment differences). We think this aligns with the DayCent model structure, because according to the DayCent manual, particulate organic carbon (POC) and MAOC are related (though not fully equivalent) to the slow pool and the passive pool, respectively. We will thus utilize the fraction of SOC that is MAOC to initialize the passive SOC pool of the model, while keeping the active pool at the

DayCent recommended mean 3% initially, and the slow pool as the rest. It is stated in the DayCent manual that the slow pool is larger than the measured POC fraction. Consequently, the passive pools must be smaller than the g MAOC g$^{-1}$ SOC fraction. Furthermore, we have only data from 2021, when the trials were already 19 and 16 years old. To account for these two points when using the g MAOC g$^{-1}$ SOC fraction as a proxy to initialize the passive pool, we will add two new parameters to the Bayesian calibration: 1) an intercept and 2) a slope for time since experiment start. The intercept accounts for the fact that the passive pool is smaller than the MAOC fraction, the slope for the fact that SOC has been on a loosing trajectory since the start of the experiments and that passive pool is usually lost at the slowest rate. Hence the fraction of g MAOC g$^{-1}$ SOC should have been lower at the start of the experiment. Therefore, the fractions have likely been shifted towards higher relative MAOC with time. We aim for a gaussian priors for these with a value of -0.3±0.1 for the intercept and -0.005±0.002 yr$^{-1}$ for the slope. This would translate into a passive pool estimate of around 40 to 50 % of the total SOC at the start of the experiment, a bit higher than usually in DayCent, according to the manual, but in alignment with the rather recent conversion of the sites to agriculture.

L335: Authors should consider replacing the term GWP with GHG balance. Despite the likely low effect of CH4, the model is not able to predict CH4 emissions, that therefore they cannot be considered in the whole balance. In this context, would be better to define the GWP as GHG balance since, in any case, the contribution of CH4 cannot be measured neither excluded.

Thanks. We will adhere to this suggestion. Depending on how well the $N_2O$ is predicted for the existing data after a needed recalibration, we might focus entirely on $CO_2$ and remove $N_2O$ as well.

L338: Figure 1 is included in M&M, please move below in Results.

Thanks, we moved it.

L390-393: Authors can remove this part since calibration is widely recognized to improve model performances.

We think it is important to keep it because the model performance improvement is from leave-one-site-out cross-validation. Hence the performance at each site was improved despite the fact that the calibration was done with only the other three sites. This is notable and indicates that the improvement of DayCent parameters suited the tropical conditions, and was not an overfitting to each site. We however see that the way we did not state clearly enough that most of the results are from the leave-one-site-out cross-validation (e.g., all Figures 4 to 7 are all from this leave-one-site-out cross-validation, despite combining all sites in one graph). We will make this clearer in the next version of the article.

L394: ….and for aboveground biomass for all sites except Machanga. You mean Aludeka?

No, this was correct, it is only that we had not displayed it for AGB. (see below). As specified above, we will use the graphs per site in the new version of the article

Uncalibrated

[Figure]

[Figure]

Calibrated

[Figure]

L399-401: please, when cited into the main text, report the supplementary figures in chronological order (why A9 before A4, etc...?). Also, why fig.4-5-6 in paragraph 3.5? It's quite hard to follow this flow....

Part of this is due to the automatic placement of figures in Latex, and it makes most sense to correct this in the final article. We will put special attention on the chronological order in the overhaul during review.

Major weaknesses:

a)  In Fig. 3 authors reported all together sites and management for comparing not vs calibrated model. To my opinion, this representation of model calibration is misleading. Firstly, looking at the performances for each site (Fig. A9), model calibration only little improve the model performances found using default values, with statistics confirming the improvement is quite low and lower for each site compared to when assessed overall. This confirm that averaging all sites make unclear to evaluate the model performances under different conditions. Also, it is not clear the ability of the model to reproduce different type of management after calibration process (Fig. A5 is poorly readable, and statistics should be reported. From a visual

analysis, variability seem not well simulated). So, from the whole study, does not clearly emerge how the model is able to reproduce yield and AGB for each ISFM at each site. This do not allow to discuss why model does or does not work at each site and for each management, which could be the limitations and weaknesses, which should be the best practice to use and its response at each site. Averaging all yield data does not clarify the efficiency of the model to be suitable as tool to assess the potential of specific ISFM management practices (as stated by authors in introduction) to cope with food insecurity or further issues. Authors should revise all this part to provide a more accurate response to what they stated in the introduction.

Based on your comment, we will report the results per site in the main manuscript (replacing Figures 3 and 5). We will also improve Fig. A5 with this in mind (adding evaluation statistics). Further, we will overhaul the Discussion Section after conducting additional simulations, to discuss how well yield and AGB for each ISFM management practice can be simulated at each site.

b) The GWP discussion is another major point of weakness. Results clearly showed as N2O is not well simulated neither at daily scale (Fig. A10) nor as cumulated (Fig. 7). Despite in discussions authors state that simulated N2O emissions were generally reasonably well predicted with this current DayCent calibration, looking at Fig. 7 emerged as at Aludeka and Embu the measured N emissions were more than double than those simulated. This clearly affect GWP analysis, especially considering the role of N in GWP analysis, thus making these results very uncertain. Authors should exclude GWP analysis from this study or should much better calibrate the N response to better fit with observations, otherwise GWP discussion risk to be highly speculative due to low level of confidence in N emission outcomes.

The reason why we stated that they were simulated reasonably was in consideration of the large uncertainty of measurements of cumulative $N_2O$ (confidence intervals of measurements overlap the 1:1 line) and that the modelling efficiencies were positive. We would thus think that the data we have cannot give a definite answer whether DayCent performs well or not. However, we agree that the simulation is not very good as stated above. Thus, we will remove $N_2O$ from the GHG balance if a model recalibration will not improve the simulations. For now, we adjusted the text as follows:

that are poorly represented in the tropics (Van Looy et al., 2017). However, the fact that cumulative $N_2O$ emissions were better captured than daily emissions, that there was no systematic under- or over-prediction of cumulative $N_2O$ emissions, and that simulated $N_2O$ emissions were in the uncertainty range of measured $N_2O$ emissions, does not provide evidence that this current DayCent calibration is not suitable to represent $N_2O$ emissions. This is important for the predictions of the GWP. Because the simulation of SOC change showed low bias, we can conclude that this part of the GWP is well represented. The contributions to GWP between 80% (Aludeka) and 20% of the GWP (other sites; Fig. 8), are less certain. The larger confidence intervals of the measured compared to the simulated cumulative $N_2O$ emissions suggest that the DayCent model cannot fully represent the variability. Although DayCent's simulation of $N_2O$ emissions is superior to using emission factors (dos Reis Martins et al.,

---

## Author Comment (AC2)

Reviewer 2:

The paper, entitled "A robust DayCent model calibration to assess the potential impact of integrated soil fertility management on maize yields, soil carbon stocks and greenhouse gas emissions in Kenya" emphasizes the importance of model calibration to enhance model accuracy. It utilizes a rich dataset from 4 sites in Kenya, an area that has been less represented/explored by many process-based models like DayCent, and thus, it provides a substantial amount valuable information. Furthermore, the paper centers its focuses on integrated soil fertility management (ISFM), maize yield, soil organic carbon, and greenhouse gas emission. Nevertheless, there are numerous concerns regarding the model calibration process (see Specific Comments section) and recommend a major revision to address these concerns before considering it for publication.

We thank you for your valuable feedback and will address the individual comments below.

General Comments:

- Line 106: it was not clear whether organic resources were applied once per year or once per season. Provide clarification.

This is specified in the next sentence. "Organic resources were applied only once a year, prior to planting for the long rainy season in January or February." However, we now also added it to the sentence you refer to.

- Section 2.3.3: Provide more detailed information on historical cropping and specify the simulation periods for reproducibility, preferably in a table format. Additionally, include information of the optimal duration of cropping systems following the transition from native condition to achieve the initial SOC levels. It would be helpful to provide a figure showing the time series of SOC stocks for the entire simulation including native condition and historic cropping systems for each site.

In response to the feedback you and reviewer 1 on the model initialization, we will now completely eliminate the spin-up and historical runs, instead relying on measured g MAOC g$^{-1}$ SOC as a proxy for the SOC in the passive pool. Thus, this whole section will be overhauled and the table will not be necessary. (see details in comment to reviewer 1 and your first "specific comment", below)

- In Section 2.5, provide the equation for the likelihood function used in the Bayesian calibration. Additionally, clarify whether the same likelihood function was employed for the GSA, and mention this in the text.

We will provide the likelihood function for the BC. For the GSA, we did not use a likelihood function, it was based on the simulated output. This is specified in the last sentence of Section 2.4 and we overhauled the sentence, to make this clearer: "The parameter sensitivity was independently determined for the mean maize grain yield and aboveground biomass, averaged over all seasons at all sites, as well as for the SOC and soil total N stocks at the end of the simulation period."

- Line 292-293, provide reference(s) for the statement, "Due to the large number of observations and the mostly balanced dataset, the off-diagonal elements were set to 0". Considering the higher autocorrelation in the time series for the modeled SOC stock, the statement may not hold true.

Based on your statement, we tested how the posterior would change if we include the covariance. It does in fact influence the results. Since, based on the reviewers' comments, we have to rerun the calibration, we will use the likelihood function with the proper variance-covariance matrix in the revised paper.

- In Figure 7, the caption mentioned "variance (measurements)". It is unclear whether the error bars represent variance, standard deviation, or 95% confidence interval. If variance is presented as error bars, this is unusual. Replace "variance" with "95% confidence interval" to main consistency consistent.

They are based on the measurement variance. We refined the statement to be clear. "Error bars represent 95% confidence intervals (measurements) and credibility intervals (simulations)."

- Figure 8 shows the difference relative to CT-N. It would be informative to show the relative differences in comparison to business-as-usual practices, as this would help identify and recommend management changes for better management practices.

Based on our field observations and discussions with local farmers and extension officers, the CT-N is in fact close to what smallholders do in the simulated regions in Kenya. For example, the average use of fertilizer in Kenya (which includes small- and large-scale farmers and all types of fertilizer) in the last two decades ranged between 30 and 50 kg ha$^{-1}$. (https://data.worldbank.org/indicator/AG.CON.FERT.ZS?locations=KE). We will look further into this issue at the national scale in a future publication, but this is beyond this article.

- In Table A1, include not only clay (%) but also sand (%) and silt (%) as required by DayCent for reproducibility.

Thank you for spotting this. We will add the sand (%), which thus will suffice DayCent input requirements (silt= 100%-sand-clay).

- In Figure A2, it is evident that measured SOC stock has been declining since the starting year. It would be helpful to discuss potential reasons for the decline and why model simulation is able to predict the decline.

This comment is likely referring to Figure 6, not A2. We have already discussed that soil erosion, which DayCent cannot simulate, could be the explanation. In our previous work we did regarding the SOC stocks at the simulated sites (https://soil.copernicus.org/articles/9/301/2023/), we discussed that the sites being relatively new in cultivation is another reason (Yet, this should be accounted for in the current manuscript, because DayCent includes the land-use history by allocating a relatively

high proportion of the initial total SOC to the slow pool, and should thus be able to simulate this effect). We will look further into this after model recalibration.

Specific Comments:

- • The manuscript employs a two-step process for model predictions: Step 1 involves running the model with one set of model parameters (i.e., native condition and historical simulation) up to the beginning of experiment (i.e., initial measurement of SOC). This is done with limited adjustment to better align the model's output with measured SOC. In Step 2, a model calibration is performed, updating various parameters to a different value, with some exhibiting significant changes of several magnitude, especially the decomposition rate of slow and passive pools. Extending the model simulation with the change in parameters may disrupt the equilibrium condition and induce a drift effect, where the model attempts to reach a new equilibrium condition due to parameter changes. This makes it challenging to determine whether the changes in SOC stocks are due to alteration in management practices or change in model parameters. The potential impacts of this should be thoroughly investigated. Additionally, in line 610, the authors claims that the newly calibrated model is applicable for "upscaling the model to larger areas in Kenya" without providing practical recommendations for simulations when two sets of model parameters are available. The associated risks of such recommendations should also be examined. To mitigate potential risk, I would recommend using a model calibration procedure that results in a single set of model parameters or joint posterior distribution.

Based on this comment and others, we decided that we will eliminate the model spin-up completely – relying instead on a measured proxy of mineral-associated organic carbon (fraction of SOC that is MAOC; i.e., g MAOC $g^{-1}$ SOC). These (unpublished) data have been measured on soil samples collected in 2021 in the framework of a master thesis in our group (the mean across treatments was 0.91, 0.88, 0.85, 0.86 g MAOC $g^{-1}$ SOC for Aludeka, Embu, Machanga, and Sidada in 0-30 cm, respectively, with no significant treatment differences). We think this aligns with the DayCent model structure, because according to the DayCent manual, particulate organic carbon (POC) and MAOC are related (though not fully equivalent) to the slow pool and the passive pool, respectively. We will thus utilize the fraction of SOC that is MAOC to initialize the passive SOC pool of the model, while keeping the active pool at the DayCent recommended mean 3% initially, and the slow pool as the rest. It is stated in the DayCent manual that the slow pool is larger than the measured POC fraction. Consequently, the passive pools must be smaller than the g MAOC $g^{-1}$ SOC fraction. Furthermore, we have only data from 2021, when the trials were already 19 and 16 years old. To account for these two points when using the g MAOC $g^{-1}$ SOC fraction as a proxy to initialize the passive pool, we will add two new parameters to the Bayesian calibration: 1) an intercept and 2) a slope for time since experiment start. The intercept accounts for the fact that the passive pool is smaller than the MAOC fraction, the slope for the fact that SOC has been on a loosing trajectory since the start of the experiments and that passive pool is usually lost at the slowest rate. Hence the fraction of g MAOC $g^{-1}$ SOC should have been lower at the start of the experiment. Therefore, the fractions have likely been shifted towards higher relative MAOC with time. We aim for a gaussian priors for these with a value of -0.3±0.1 for the intercept and -0.005±0.002 $yr^{-1}$ for the slope. This would translate into am fraction of around 40 to 50 % of the total SOC in the passive pools at start, a bit higher than usually in DayCent, according to the manual, but in alignment with the rather recent conversion of the sites to agriculture.

- The manuscript utilizes initial parameter value for SOM decomposition, as reported in Gurung et al. (2020), which were suitable for SOC in the top 30 cm. However, the modeled SOC stocks were compared against measured SOC stocks up to a depth of 20 cm, thus resulting in a non-equivalent comparison. This inconsistency is evident in Figure A7, where the reported model predictions consistently show higher values than the measured SOC.

- IPCC recommends modeling SOC to a depth of 30 cm for GHG accounting and reporting. Since SOC measurements to 30 cm were available, it would be more appropriate to calibrate the model to simulate SOC to 30 cm, aligning it with the IPCC's recommendation.

You are right with these two comments. As a result, we will redo the model calibration for the 0-30 cm soil depth. However, data from the 15-30 cm soil depth was only available from an intense soil sampling campaign in 2021 (https://soil.copernicus.org/articles/9/301/2023/). After a statistical test for the 15-30 cm soil depth on that dataset (see two graphs below), we found that the equivalent soil mass (ESM) based SOC stocks in the 15-30 cm layer (2.5-4.7 t soil ha$^{-1}$) were not different between the treatments (with only one single exception in Aludeka). We will therefore derive the 0-30 cm SOC stocks by adding the site-specific value of the 15-30 cm ESM based SOC stocks from 2021 to the SOC stocks for 0-15 cm, which previously we scaled to 0-20 using the equation of Jobbágy and Jackson (2000).

[Figure]

[Figure]

- The manuscript employs a "leave-one-site-out" cross-validation approach; however, the analysis and results of the cross-validation were not presented. I recommend including some detail about the cross-validation process and its results in the manuscripts.

Most the results displayed (e.g., Fig 3, 4, 5, 6, A3, A4, A7, A8, A9) show the results from the "leave-one-site-out" cross-validation approach. We see however, that this was not formulated clearly enough and that the fact that we represent only the joint posterior parameters of all sites is confusing. We will therefore specify this more clearly in the text (and also display the 4 different posteriors by leaving one site out in the appendix)

Technical Corrections:

- Line 324: move the explanation "O____$_y$ the mean of the y-th type of measurement" below equation-9.

Thanks, we have done so!

- Line 335: mass unit for CO2eq/ha/yea) is missing.

Thanks, we added it.

- In the caption for Figure 7, replace "95% confidence intervals" with "95% credible intervals" for BC.

Thanks! We have adjusted this, as specified above.

---

## Author Comment (AC3)

Community Comment 1:

General comments: Very nice paper. Generally well written. The M&M in particular are very thorough. I have not done much modeling, but I found that the M&M did a good job of explaining the model parameters and their calibration along with how sensitive they were. Apart from a bunch of small issues (see below), I found that the discussion around objective iii. was lacking a bit. What I was really looking forward to was more discussion around the trade-offs between yield and SOM / increases along with the global warming potential of the different ISFM treatments.

Thanks for this positive feedback. We aim to refine the discussion part on objective iii, after a model recalibration, based on the comments of reviewer 1 and 2.

Specific comments:

Lines 85-90: A map with the site locations would be helpful here as well.

We agree that a map would be helpful but decided against having one, since the manuscript already contains a lot of figures. We will add a map with the locations to the supplement.

Line 118 : should be "CH4 oxidation".

Thanks, this was corrected.

Lines 150-155: How many samples per chamber? How long was the deployment time? How did you calculate the change in mixing ratios over time (linear or non-linear?), how were gas samples analyzed? (on a GC? What kind?). You need a bit more detail here.

Thanks for making us aware of this, we added the missing information to the manuscript:

to mid-May in Sidada). They were conducted with the static chamber method (Hutchinson and Mosier, 1981). Two measuring frames were permanently installed in the plots for a whole rainy season (one within, one between maize rows). The chambers ($0.27 \times 0.375 \times 0.11$ m) were made of polyvinylchloride and equipped with a vent tube and a fan to homogenize gas inside them before gas sampling. Gas samples were extracted through a septum sealed sampling port using a 60mL Polypropylene syringe. Four gas samples were collected within the closure period at interval of 0, 15, 30 and 45min. The pooling method, as described in (Arias-Navarro et al., 2017) was used, where gas samples from the two subplot chambers (inside and between maize rows) were combined per time point and thus mixed in the same syringe. All the samples collected were measured using a SRI 8610C gas chromatography (456-GC, Scion Instruments, Livingston, United Kingdom) equipped with an electron capture detector for $N_2O$ analysis. The fluxes per surface area of the chamber were determined by using the linear slope the gas concentration over time (Pelster et al., 2017; Barthel et al., 2022). The measured $N_2O$ emissions were evaluated at two

Line 367: Is "langley" an SI unit? I had to do an internet search to find out what it is. Would it be possible to explain what this is? Or convert to SI units?

No, but it is the unit used in DayCent. We added the explanation "(1 langley is 41 840 J m$^{-2}$)" to the sentence.

Figure 2: shouldn't there be some label on the X and Y axes?

Yes, thanks! Should be "Density" and "value" as y and x axes. We will add this to the next version of the manuscript.

Line 395: perhaps I don't quite understand, but isn't the systemic underestimation at high yields (and AGB) a "bias"?

A systematic underestimation at high yields (and AGB) is shown by the nonunity slope (<1; overestimation at low values, underestimation at high values). A bias is considered the over- or underestimation across the full range of data. Due to the large amount of data (many overlaps in the average yields), visual inspection of bias etc. are misleading and thus the SB, NU, LC assessment of Gauch (2003) that we applied are a better approach to assess whether bias exists.

Line 446: wouldn't a negative reduction be an increase?

Yes, we reformulated: "The changes ranged from an increase of 0.2 t $CO_2$ equivalent $ha^{-1}$ $yr^{-1}$ to reductions of 1 t $CO_2$ equivalent $ha^{-1}$ $yr^{-1}$."

Line 447: I would say "led to" rather than "could lead to". Since there was a reduction noted.

Thanks, we changed the text accordingly.

Figure 7: It seems that you are unable to simulate the high emission days, which could be why the cumulative simulated emissions are typically lower than the 1:1 line. Also, in the Sidada site for simulate vs measured cumulative emissions, you have one data point that has a lot of leverage. I would consider seeing how the regression line looks without that point. And maybe investigate why that point is so different from the rest of the data at that site.

Thanks for this suggestion. We will consider these in the overhaul of the manuscript.

Lines 483-484: Mention here that DayCent overestimated SOC pre-calibration, but after the calibration the SOC concentrations (or stocks) were simulated much more accurately. This difference is clear when you look at the figure, but since the figure is in the appendix, it may not be readily apparent to the readers.

We changed the sentence to: "This is also the case for the present study; the results of the model simulations with the initial parameter sets looks good for the absolute SOC stocks, due to limited change, but not for to the changes in the SOC stocks (Fig. A7 vs 5). "

Line 513: why do you use $2^1$ and $2^3$ here? Why not just say 2 and 8? Or am I missing something?

We wanted to express it in terms of that can be related to first order kinetics. After your comment we decided to replace it by 2 and 8 as suggested.

Lines 526 to 529: Is there a reason why you switch between SOC and SOM? It seems like you are talking about the same thing.

Thanks for spotting this, it should all be SOM here. SOM is the model pool, SOC is related to carbon.

Line 534: "vary" not "very".

Thanks, we changed this.

Line 543: Are you saying that DayCent does not capture yield increases above 100-150 kg N per ha per season in general? Or just specifically in Kenya. I have not used DayCent, but I would be very surprised if it does not capture yield increases above 150 kg N per ha in temperate regions.

No, just in our study – we have not tested it for other sites/climates. We added "at the four sites" to the end of the sentence.

Line 549-553: I wouldn't worry too much about the poor match between simulated and measured daily fluxes. I would mention though that the timing of peak fluxes is related more to soil gas diffusivity and that soil hydraulics are more just a proxy of the diffusivity.

Thanks for this suggestion. We added this suggestion to the text.

Line 553: Sommer et al. 2016 does not quite say this. What they say is that "As such, the overall model fit was exceptionally good, even though the visual impression would suggest a significant overestimation of emissions by CropSyst". If you look at the figures in their study, the simulated line up very well with the measured emissions. It is just that there are a lot of peaks in the simulated that occur between samplings.

We reconsidered this part of the sentence and you are correct with regards to the text of Sommer et al (2016). We thus decided to remove the citation.

Line 569: I guess this is somewhat true, in that maize mono-cropping will still produce some GHG emissions. However what is the difference between the ISFM practices and the "typical" treatment (what is typical? No inputs? No N input and a small amount of FYM)? It seems like adding some inorganic N with 1.2 T C increased yields, without increasing yield scaled emissions compared with 0N 0C and compared with 0n 1.2T C. So even though it is not exactly "negative emission technology" it still seems to be an improvement.

Yes we agree. To highlight this we adjusted the sentence: "However, the strong differences in the yield-scaled GWP between treatments, such as a 72, 32, 63 and 14 % lower yield-scaled GWP in the FYM 1.2+N treatment compared to the control-N treatment, show that ISFM can still lead to an improvement compared to no- or low-input input treatments and that the yield-scaled GWP is highly relevant in practical terms." Based on the comments of reviewer, we will recalibrate and reconsider GWP and change it to only $CO_2$ emissions.

Line 570: why say "positive absolute" in stead of just "positive"?

We agree changed it to "positive" only.

Lines 578-580: While I agree that N fertilizer should only be applied to responsive soils, I'm not sure that is a conclusion of the date that you have here. If you look at yields, all the sites respond to N fertilizer (either mineral or organic). It is just that they seem to respond a bit differently, particularly in the N2O emissions, to the fertilizer applications. Besides, the 0N control also has much higher yield scaled GWP in Embu and Machanga, mainly related to loss of SOC, so I don't think the higher yield scaled emissions (compared with Sidada and Aludeka) with the +N treatments indicate that these shouldn't be fertilized. In fact, the decrease in yield-scaled GWP when adding N is greater at the sites in Central Kenya than they are at Sidada, which almost contradicts what you are saying here.

We agree and removed this sentence after reconsideration.

Line 610: Just mention which treatment had the lowest yield-scaled emissions (the mix of FYM +N) as the preferred INMS for Kenya.

We will consider this, but results may change after the recalibration we will do.

Table A1: can you add the sand content as well?

Done

Figure A3: what depth are you using to calculate the stocks? You mention 15 cm depth in some locations, but you also mention that DayCent uses 20 cm depth. And, I am having a hard time seeing how the Machanga site lost so much of its C. at 20 cm depth a soil with a C content of 0.3 and a BD of 1.51 would have about 10 t C per ha. And you are saying here that it lost about 10 t per ha (or essentially all of its soil C). Is my math off (wouldn't be the first time).

We added the depth to the Figure A3: (0-20 cm in this version, will be 0-30 in the next). To your question of Machanga – the site had initially about 20 t C ha$^{-1}$. We realized that the initial soil C and N in the Table A1 were still data from the original reference profile plot description, which consisted of a single measurement per horizon at each site, conducted before the trials were established. However, at the time of trial establishments, further soil C and N measurements were at each plots resulting in slightly different initial C and N contents (see below). We had actually used these more accurate measures of SOC to match the SOC stocks in DayCent. We now updated Table A1 for consistency. Thanks for spotting this.

| Soil characteristics | Embu | Machanga | Sidada | Aludeka |
|---|---|---|---|---|
| Latitude | -0.517 | -0.793 | 0.143 | 0.574 |
| Longitude | 37.459 | 37.664 | 34.422 | 34.191 |
| Initial soil C (%) | 3.1 | 0.8 | 2.6 | 0.7 |
| Initial N (%) | 0.3 | 0.05 | 0.21 | 0.06 |

Figure A6: the figure caption needs to be re-done. For example, the second sentence is missing a word somewhere (perhaps "was" before "insensitive"?). And secondly, are you sure about the 50/50 split application? You were calibrating to data where the split application was 40 kg N at planting and 80 kg after ~ 6 weeks (see line 107-108; also line 165).

Thanks for spotting this. We have rewritten the caption. We are sure, about the evenly split application – this was for the technical reason that DayCent did not allow to go for higher N applications than 200kg N per one application (and we wanted to test until 400).

**Figure A6.** Yield response curve of DayCent to varying levels of mineral N application (control + N treatment, without organic resources) using the calibrated DayCent parameters. Displayed are the simulated mean yields across all simulated seasons (32 in Sidada and Aludeka, 38 seasons in Embu and Machanga). The amount of mineral N applied per season in the simulations was evenly split between the actual application dates of mineral N in each season at each site.

Figure A10, can you increase the font size in the figure please?

We are not entirely sure if we will keep figure A10, after the overhaul of the manuscript. But in case we do, we will increase the font size as suggested.

---

## Author Response (AR1)

Reviewer 1:

The paper describes the capability of DayCent model to simulate yield and SOC development of the different ISFM practices in SSA and its improvement after cal-val. So as presented, the paper is quite long and verbose, resulting quite hard to follow. The figures do not follow a chronological order and are often hard to interpret (see fig. A5). While authors report in M&M a wide description of parameters selection and initialization values which is appropriate and detailed, results are not very clear, often reporting average data which do not highlight the model's ability to reproduce the different selected managements. Also, the mismatch in N2O simulations make hard accounting the GWP here reported. Based on these premises, I recommend a major revision before to be acceptable for publication.

Thank you for your critical feedback. As a response to your concerns, we have conducted additional model runs, reconsidered most of the figures (e.g., displaying per site) and we reflect more critically the results with respect to the model's ability to reproduce the different selected managements. We also improved the simulations of $N_2O$. We think the article improved considerably with the changes we made, and hope you agree with this. See below our detailed responses to the individual comments.

Comments:

L118: …CH oxidation4. Typo. Thanks for spotting this. It was corrected.

L241-243: As authors state, DayCent needs to initialize the SOM pools to equilibrium using the typical input of biomass of the native vegetation. However, simulating native vegetation in SSA is not plausible since it is characterized by tropical evergreen forest, dry savanna and humid savanna that, with the only exception of savanna systems which was partly simulated in literature using the grass and tree layers, DayCent is not able to well simulate forest production (Gathany and Burke, 2012). Also, to my knowledge, DC was never tested over tropical environments. Authors should better explain what they used as vegetation for model spin-up.

We agree that the spin-up is very uncertain for DayCent and for other similar models in general (and not just in SSA, but in general), and it was also raised as an issue by reviewer 2. Data on the history of land use is usually difficult to get in good quality (if any information is available at all), especially in SSA. This is why Mathers et al. (2023) have switched to using the spin-up and historical runs only for the distribution of total C among the different SOC pools (www.doi.org/10.1016/j.geoderma.2023.116647). However, even this comes with a lot of uncertainty regarding the real biophysical conditions and human interactions, so measured pools would in fact be best.

We therefore decided that we will eliminate the model spin-up completely – relying instead on a measured mineral-associated organic carbon pool (fraction of SOC that is MAOC; i.e., g MAOC g$^{-1}$ SOC). See new section 2.3.3 below.

**2.3.3  Soil organic matter pools initialization based on measured data**

[revised manuscript text omitted]

.

L335: Authors should consider replacing the term GWP with GHG balance. Despite the likely low effect of CH4, the model is not able to predict CH4 emissions, that therefore they cannot be considered in the whole balance. In this context, would be better to define the GWP as GHG balance since, in any case, the contribution of CH4 cannot be measured neither excluded.

Thanks. We have adjusted the name to GHG balance.

L338: Figure 1 is included in M&M, please move below in Results.

Thanks, we moved it.

L390-393: Authors can remove this part since calibration is widely recognized to improve model performances.

We gave this suggestion some thought but in the end concluded that it is important to keep it because the model performance improvement is from leave-one-site-out cross-validation. Hence, the performance at each of the four sites was improved despite the calibration being done with only the three other sites. This is notable and indicates that the improvement of DayCent parameters suited the tropical conditions and was not an overfitting for each individual site. However, we acknowledge that we did not clearly specify that most of the results are from the leave-one-site-out cross-validation (e.g., Figures 4 to 7 are all from this leave-one-site-out cross-validation, despite combining all sites in one graph). We made this clearer now in the revised manuscript.

L394: ….and for aboveground biomass for all sites except Machanga. You mean Aludeka?

This is actually correct, it is only that we had not displayed it for AGB. (see below). As specified above, we will use the graphs per site in the new version of the article.

Uncalibrated model results on top, calibrated ones at the bottom

[Figure]

L399-401: please, when cited into the main text, report the supplementary figures in chronological order (why A9 before A4, etc…?). Also, why fig.4-5-6 in paragraph 3.5? It's quite hard to follow this flow….

Thank you for this comment because this should indeed be in order. Hence, we put special attention on the chronological order of tables and figures during revision.

Major weaknesses:

a) In Fig. 3 authors reported all together sites and management for comparing not vs calibrated model. To my opinion, this representation of model calibration is misleading. Firstly, looking at the performances for each site (Fig. A9), model calibration only little improve the model performances found using default values, with statistics confirming the improvement is quite low and lower for each site compared to when assessed overall. This confirm that averaging all sites make unclear to evaluate the model performances under different conditions. Also, it is not clear the ability of the model to reproduce different type of management after calibration process (Fig. A5 is poorly readable, and statistics should be reported. From a visual analysis, variability seem not well simulated). So, from the whole study, does not clearly emerge how the model is able to reproduce yield and AGB for each ISFM at each site. This do not allow to discuss why model does or does not work at each site and for each management, which could be the limitations and weaknesses, which should be the best practice to use and its response at each site. Averaging all yield data does not clarify the efficiency of the model to be suitable as tool to assess the potential of specific ISFM management practices (as stated by authors in introduction) to cope with food insecurity or further issues. Authors should revise all this part to provide a more accurate response to what they stated in the introduction.

Based on your comment, we have put the site specific cross-evaluation results into the main text and report across site model statistics only in the figure captions. We now also describe in more detail, which ISFM techniques are reasonably represented by the model vs which ones are not, and for which sites and conditions the model is performing the least. We additionally added evaluation criteria to the new Fig. A4, showing that site means have a lower RSME than yearly simulated yields. Fig. A5 from the last version was removed, because it did not add any relevant information that was not presented in Figs. 3, 4, and A4, combined.

[Figure]

b) The GWP discussion is another major point of weakness. Results clearly showed as N2O is not well simulated neither at daily scale (Fig. A10) nor as cumulated (Fig. 7). Despite in discussions authors state that simulated N2O emissions were generally reasonably well predicted with this current DayCent calibration, looking at Fig. 7 emerged as at Aludeka and Embu the measured N emissions were more than double than those simulated. This clearly affect GWP analysis, especially considering the role of N in GWP analysis, thus making these results very uncertain. Authors should exclude GWP analysis from this study or should much better calibrate the N response to better fit with observations, otherwise GWP discussion risk to be highly speculative due to low level of confidence in N emission outcomes.

Thanks for this suggestion. After the recalibration in the revised paper (and when choosing more suitable $N_2O$ model parameters), we could improve the cumulative $N_2O$ predictions, removing most of the systematic model underprediction of $N_2O$ emissions.

. Third, for the parameters determining the minimum and maximum proportion of nitrified N lost as $N_2O$, we used a value that was in between the most recent values from Gurung et al. (2021),  because the default parameters led to too high, the Gurung et al. (2021) parameters to too low emissions.

[Figure]

We also adjusted the text as follows:

that are poorly represented in the tropics (Van Looy et al., 2017). However, the fact that cumulative $N_2O$ emissions were better captured than daily emissions, that there was no systematic under- or over-prediction of cumulative $N_2O$ emissions,   and that simulated $N_2O$ emissions were  in the uncertainty range of measured $N_2O$ emissions, does not provide any strong evidence against the suitability of DayCent to represent $N_2O$ emissions with this current  calibration. Because the simulation of SOC change  and cumulative $N_2O$ emissions showed only limited bias, the GWP seem to be at least a reasonable first estimate. The contributions of $N_2O$ emissions to GWP of up to 100% (Aludeka) and between 10 to 50% of the GWP (other sites; Fig. 9), are, however subject to high uncertainty, as already evident from the measurements. The larger confidence intervals of the measured compared to the simulated cumulative $N_2O$ emissions suggest that the DayCent model cannot fully represent the variability.  Thus, although DayCent's simulation of $N_2O$ emissions is superior to using emission factors (dos Reis Martins et al., 2022), simulating $N_2O$ emissions remains challenging and highly uncertain due to the complexity of the processes involved and the high temporal and spatial variability.

Reviewer 2:

The paper, entitled "A robust DayCent model calibration to assess the potential impact of integrated soil fertility management on maize yields, soil carbon stocks and greenhouse gas emissions in Kenya" emphasizes the importance of model calibration to enhance model accuracy. It utilizes a rich dataset from 4 sites in Kenya, an area that has been less represented/explored by many process-based models like DayCent, and thus, it provides a substantial amount valuable information. Furthermore, the paper centers its focuses on integrated soil fertility management (ISFM), maize yield, soil organic carbon, and greenhouse gas emission. Nevertheless, there are numerous concerns regarding the model calibration process (see Specific Comments section) and recommend a major revision to address these concerns before considering it for publication.

We thank you for your valuable feedback and will address the individual comments below.

General Comments:

- Line 106: it was not clear whether organic resources were applied once per year or once per season. Provide clarification.

This is specified in the next sentence. "Organic resources were applied only once a year, prior to planting for the long rainy season in January or February." However, we now also added this to the sentence you refer to.

nols (Table A2). Each organic resource was applied  at two rates, 1.2 and 4 t C ha$^{-1}$ yr$^{-1}$, while  mineral N fertilizer was applied at a fixed rate of 120 kg N ha$^{-1}$ (CaNH$_4$NO$_3$) in each of the two growing seasons Of this, 40 kg N ha$^{-1}$ were applied  at planting, and the remaining 80 kg N ha$^{-1}$ about six weeks  later. Organic resources were applied only once a year, prior to planting  in the long rainy season, i.e., in January or February. They were incorporated to a depth of 15 cm with hand hoes. Furthermore, a blanket application

- Section 2.3.3: Provide more detailed information on historical cropping and specify the simulation periods for reproducibility, preferably in a table format. Additionally, include information of the optimal duration of cropping systems following the transition from native condition to achieve the initial SOC levels. It would be helpful to provide a figure showing the time series of SOC stocks for the entire simulation including native condition and historic cropping systems for each site.

In response to the feedback from this reviewer and reviewer 1 on the model initialization, we have completely eliminated the spin-up and historical runs, instead we relied on measured mineral-associated organic carbon pool g (MAOC g$^{-1}$ SOC) as a proxy for the SOC in the passive pool. Thus, this whole section was rewritten and the table is no longer necessary. See details in response to reviewer 1 and first "specific comment" of this reviewer below.

- In Section 2.5, provide the equation for the likelihood function used in the Bayesian calibration. Additionally, clarify whether the same likelihood function was employed for the GSA, and mention this in the text.

We now provide the likelihood function for the BC.

autocorrelation of residuals. The likelihood was a function of the following form:

$$p(D|M,\theta_z) = \frac{1}{\sqrt{2\pi\Sigma}} \exp\left(-\frac{1}{2}(M(\theta_z) - D)^T \Sigma^{-1}(M(\theta_z) - D)\right)$$

For the GSA, we did not use a likelihood function, it was based on the simulated output. This is specified in the last sentence of Section 2.4 and we rewrote the sentence, to make this clearer:

 The parameter sensitivity was independently determined for the mean maize grain yield  and aboveground biomass, averaged over all seasons at all sites, as well as for the SOC and soil N stocks at the end of the simulation period.

- Line 292-293, provide reference(s) for the statement, "Due to the large number of observations and the mostly balanced dataset, the off-diagonal elements were set to 0". Considering the higher autocorrelation in the time series for the modeled SOC stock, the statement may not hold true.

Based on your statement, we tested how the posterior would change if we included the covariance. It does in fact influence the results and we updated the likelihood function to include the covariances now.

$$p(D|M,\theta_z) = \frac{1}{\sqrt{2\pi\Sigma}} \exp\left(-\frac{1}{2}(M(\theta_z) - D)^T \Sigma^{-1}(M(\theta_z) - D)\right) \qquad (3)$$

Here, $\Sigma$ is the variance covariance matrix, $M(\theta_z)$ is the vector of simulated values using the z-th parameter set $\theta_z$ and $D$ the vector of observed data. In the R software, this can be constructed by setting the residual (modelled value - measured) as the dependent variable of a zero intercept model with nested random effects (i.e., sampling date within site), and assigning the

inverse of the median standard deviation (of each type of measurement at each site) as weight. The logLik() function is then used to extract the log-likelihood, which is transformed to the likelihood by raising $e$ to the power of the log-likelihood.

- In Figure 7, the caption mentioned "variance (measurements)". It is unclear whether the error bars represent variance, standard deviation, or 95% confidence interval. If variance is presented as error bars, this is unusual. Replace "variance" with "95% confidence interval" to main consistency consistent.

Thanks for highlighting this unclear description. They are based on the measurement variance. We refined the statement to ensure clarity, as follows. "Error bars represent 95% confidence intervals (measurements) and credibility intervals (simulations)."

- Figure 8 shows the difference relative to CT-N. It would be informative to show the relative differences in comparison to business-as-usual practices, as this would help identify and recommend management changes for better management practices.

Based on our field observations and discussions with local farmers and extension officers, the CT-N is in fact the business-as-usual practice of smallholders in Kenya; many smallholders do not use chemical fertilizer because of being too costly (especially since the war in Ukraine) and use very minimal organic resources due to accessibility and labour constraints. Nevertheless, farmers are interested in the different ISFM treatments because they do observe soil degradation in their fields. Thus, they do want to go away from the business-as-usual scenario.

- In Table A1, include not only clay (%) but also sand (%) and silt (%) as required by DayCent for reproducibility.

Thank you for bringing this to our attention. We added the sand (%), which thus meets the DayCent input requirements (silt= 100%-sand-clay).

- In Figure A2, it is evident that measured SOC stock has been declining since the starting year. It would be helpful to discuss potential reasons for the decline and why model simulation is able to predict the decline.

This comment is likely referring to Figure 6, not A2. We have already discussed that soil erosion, which DayCent does not simulate, could be the explanation for declining SOC. We now added the explanation "In fact, the sites were under natural vegetation (i.e. forest) or fallow up to relatively shortly before the experiment establishment. Hence upon the start of cultivation, erosion and enhanced decomposition (due to disturbance) were accelerated and have likely not yet reached a new equilibrium with C inputs from the maize. Therefore, C loss is the dominant process occurring at the sites." More details can be found in our previous work, where we discussed other reasons why SOC stocks at the simulated sites are declining (https://soil.copernicus.org/articles/9/301/2023/).

Specific Comments:

- The manuscript employs a two-step process for model predictions: Step 1 involves running the model with one set of model parameters (i.e., native condition and historical simulation) up to the beginning of experiment (i.e., initial measurement of SOC). This is done with limited adjustment to better align the model's output with measured SOC. In Step 2, a model calibration is performed, updating various parameters to a different value, with some exhibiting significant changes of several magnitude, especially the decomposition rate of slow and passive pools. Extending the model simulation with the change in parameters may disrupt the equilibrium condition and induce a drift effect, where the model attempts to reach a new equilibrium condition due to parameter changes. This makes it challenging to determine whether the changes in SOC stocks are due to alteration in management practices or change in model parameters. The potential impacts of this should be thoroughly investigated. Additionally, in line 610, the authors claims that the newly calibrated model is applicable for "upscaling the model to larger areas in Kenya" without providing practical recommendations for simulations when two sets of model parameters are available. The associated risks of such recommendations should also be examined. To mitigate potential risk, I would recommend using a model calibration procedure that results in a single set of model parameters or joint posterior distribution.

Based on this comment and others, we decided to eliminate the model spin-up completely – relying instead on measured mineral-associated organic carbon (i.e., fraction of SOC that is MAOC; i.e., g MAOC $g^{-1}$ SOC). See section 2.3.3.

[revised manuscript text omitted]

- The manuscript utilizes initial parameter value for SOM decomposition, as reported in Gurung et al. (2020), which were suitable for SOC in the top 30 cm. However, the modeled SOC stocks were compared against measured SOC stocks up to a depth of 20 cm, thus resulting in a non-equivalent comparison. This inconsistency is evident in Figure A7, where the reported model predictions consistently show higher values than the measured SOC.

- • IPCC recommends modeling SOC to a depth of 30 cm for GHG accounting and reporting. Since SOC measurements to 30 cm were available, it would be more appropriate to calibrate the model to simulate SOC to 30 cm, aligning it with the IPCC's recommendation.

We agree with these two important comments. As a result, we have redone the model calibration, now using data for the 0-30 cm soil depth. See section 2.3 pasted below with associated figures:

soil bulk density per site was used to calculate SOC stocks of the top 15 cm of soil depth.  We used a DayCent parameterization that was developed to simulate SOC stocks of the IPCC-recommended 0-30 cm topsoil layer (Gurung et al., 2020) (further details in section 2.3.2). Thus, the 0-15 cm SOC stocks were adjusted to 0-30 cm depth. This was done by adding the site-specific SOC stocks from the 15-30 cm layer (specifically, the 15-30 cm equivalent-soil-mass-based ones (Wendt and Hauser, 2013; Lee et al., 2009)) to the treatment-specific SOC stocks

$$SOC_{20}(kg\ ha^{-1}) = \frac{1 - \beta^{20}}{1 - \beta^{15}} * SOC_{15}$$

 from 0-15 cm. Due to limited data availability for the 15-30 cm soil depth , this approach was considered the most conservative and robust; subsoil carbon usually changes very slowly, and a statistical test revealed no differences in the equivalent soil mass based SOC stocks of the 15-30  cm layer (2.5-4.7 t soil ha^-1) between treatments at the same site in 2021 (with only one single exception in Aludeka; Fig. A2).

[Figure]

[Figure]

- The manuscript employs a "leave-one-site-out" cross-validation approach; however, the analysis and results of the cross-validation were not presented. I recommend including some detail about the cross-validation process and its results in the manuscripts.

We acknowledge that we had not stated clearly enough that all plots of simulated compared to measured data were effectively from the "leave-one-site-out" cross-validation. This has been clarified, and we also show the different posteriors from leaving out each site. Based on the comments from reviewer 1, we also have moved the evaluation graphs by-site into the main text.

**2.3    Data used for the DayCent model** /**calibration** and evaluation

To provide an overall assessment of the performance of DayCent for its use in Kenya a leave-one-site-out cross-validation approach was applied. Specifically, this involved using a data sub-set from three of the four sites for  model calibration, with validation performed using the data from the fourth site.  This process was repeated four times, every time with another site serving as the validation site. Different data, were used for this: Maize grain yield and the aboveground biomass, both on a dry matter basis,

[Figure]

**Figure 2.** Prior compared to the posterior model parameter distribution resulting from the uncertainty-based Bayesian model calibration of DayCent using data from all sites combined (top) and the leave-one-site-out cross-validation (bottom). Dashed vertical lines represent the values of the  initially selected parameter set. The posterior distributions are based on all four study sites combined. For the description of the parameters see Table 1.

Technical Corrections:

- Line 324: move the explanation "O‾‾y the mean of the y-th type of measurement" below equation-9.

Thanks, we have done so!

- Line 335: mass unit for CO2eq/ha/yea) is missing.

Thanks, we added it.

- In the caption for Figure 7, replace "95% confidence intervals" with "95% credible intervals" for BC.

Thanks! We have adjusted this, as specified above.

Community Comment 1:

General comments: Very nice paper. Generally well written. The M&M in particular are very thorough. I have not done much modeling, but I found that the M&M did a good job of explaining the model parameters and their calibration along with how sensitive they were. Apart from a bunch of small issues (see below), I found that the discussion around objective iii. was lacking a bit. What I was really looking forward to was more discussion around the trade-offs between yield and SOM / increases along with the global warming potential of the different ISFM treatments.

Thanks for this positive feedback. We refined the discussion part on objective iii, after a model recalibration, based on the comments of reviewer 1 and 2. However, since there was still quite some uncertainty around simulated $N_2O$ emissions, we focused this discussion section on this uncertainty and its potential sources.

Specific comments:

Lines 85-90: A map with the site locations would be helpful here as well.

We agree that a map would be helpful but decided against having one, since the manuscript already contains a lot of figures. However, we added a map with the locations to the Supplementary Materials.

[Figure]

**Figure A1.**  Map displaying the  location of the  four study sites.

Line 118 : should be "CH4 oxidation".

Thanks, this was corrected.

Lines 150-155: How many samples per chamber? How long was the deployment time? How did you calculate the change in mixing ratios over time (linear or non-linear?), how were gas samples analyzed? (on a GC? What kind?). You need a bit more detail here.

Thanks for making us aware of this, we added the missing information to the manuscript, while at the same time trying to be brief.

and in 2021 (weekly measurements form mid-March to mid-May in Sidada).  The measurements applied the static chamber method (Hutchinson and Mosier, 1981)  with two measuring frames per plot permanently installed for a whole rainy season  (one within, one between maize rows). The sampling chambers (0.27 × 0.375 × 0.11 m)  had a vent tube and  fan for to homogenize the gas sample before extraction with a 60 mL polypropylene syringe through a septum-sealed sampling port. Four gas samples were collected at 0, 15, 30 and  45 min of chamber closure. Gas samples from within and between maize rows were combined per time point in the same syringe (Arias-Navarro et al., 2017). All analyses were conducted using a SRI 8610C gas chromatography (456-GC, Scion Instruments, Livingston, United Kingdom) equipped with an electron capture detector for N₂O analysis. Fluxes per surface area were determined using the linear slope of gas concentration over time (Pelster et al., 2017; Barthel et al., 2022).

Line 367: Is "langley" an SI unit? I had to do an internet search to find out what it is. Would it be possible to explain what this is? Or convert to SI units?

No, but it is the unit used in DayCent. We added the explanation "(1 langley is 41 840 J m$^{-2}$)" to the sentence.

Figure 2: shouldn't there be some label on the X and Y axes?

Yes, thanks! Should be "Density" and "value" as y and x axes. We added this to the new version of the manuscript.

Line 395: perhaps I don't quite understand, but isn't the systemic underestimation at high yields (and AGB) a "bias"?

A systematic underestimation at high yields (and AGB) is shown by the non-unity slope (<1; overestimation at low values, underestimation at high values). A bias is considered as the case of an over- or underestimation across the full range of data. Due to the large amount of data (many overlaps in the average yields), visual inspection of bias etc. are misleading and thus the SB, NU, LC assessment of Gauch (2003) are a better approach to assess whether bias exists.

Line 446: wouldn't a negative reduction be an increase?

Yes, we reformulated, as follows:

-N treatments were  simulated to have lower emissions (Fig.  9). Yet, including the +N treatments, the changes ranged from an increase of CO$_2$ equivalent ha$^{-1}$ yr$^{-1}$ to  a reduction of 2.5 t CO$_2$ equivalent ha$^{-1}$ yr$^{-1}$.  Embu was the site where the addition of mineral N (+N treatment)

Line 447: I would say "led to" rather than "could lead to". Since there was a reduction noted.

Thanks, we changed the text accordingly.

Figure 7: It seems that you are unable to simulate the high emission days, which could be why the cumulative simulated emissions are typically lower than the 1:1 line. Also, in the Sidada site for simulate vs measured cumulative emissions, you have one data point that has a lot of leverage. I would consider seeing how the regression line looks without that point. And maybe investigate why that point is so different from the rest of the data at that site.

Thanks for this suggestion. We have improved the simulation of N₂O in the revised version, by fitting more suitable model parameter values for N₂O production, effectively removing this bias that you and reviewer 1 mentioned (see figure below).

[Figure]

Lines 483-484: Mention here that DayCent overestimated SOC pre-calibration, but after the calibration the SOC concentrations (or stocks) were simulated much more accurately. This difference is clear when you look at the figure, but since the figure is in the appendix, it may not be readily apparent to the readers.

The whole section changed due to the new results, hence this part was removed.

Line 513: why do you use $2^1$ and $2^3$ here? Why not just say 2 and 8? Or am I missing something?

The whole section changed due to the new results, hence this part was removed.

Lines 526 to 529: Is there a reason why you switch between SOC and SOM? It seems like you are talking about the same thing.

Thanks for spotting this, it should all be SOM here. SOM is the model pool, SOC is related to carbon. We went through the manuscript to correct this everywhere.

Line 534: "vary" not "very".

Thanks, we changed this.

Line 543: Are you saying that DayCent does not capture yield increases above 100-150 kg N per ha per season in general? Or just specifically in Kenya. I have not used DayCent, but I would be very surprised if it does not capture yield increases above 150 kg N per ha in temperate regions.

No, just in our study – we have not tested it for other sites/climates. We added "at the four sites" at the end of the sentence.

SOC (e.g. Reichenbach et al., 2021; Mainka et al., 2022).  Finally, our model sensitivity test to mineral N  inputs suggests that the maize yield bias at high N is due to DayCent's inability to capture yield increases above 100-150 kg N per ha and season at the four sites (Fig. A5); the +N treatments of *Tithonia*, *Calliandra* and farmyard manure at 4 t C

Line 549-553: I wouldn't worry too much about the poor match between simulated and measured daily fluxes. I would mention though that the timing of peak fluxes is related more to soil gas diffusivity and that soil hydraulics are more just a proxy of the diffusivity.

Thanks for this suggestion. We added this suggestion to the text.

 moisture dynamics by the 'tipping bucket' soil water balance approach and that soil gas diffusivity is not explicitly simulated (Zhang and Yu, 2021; Wang et al., 2020)

Line 553: Sommer et al. 2016 does not quite say this. What they say is that "As such, the overall model fit was exceptionally good, even though the visual impression would suggest a significant overestimation of emissions by CropSyst". If you look at the figures in their study, the simulated line up very well with the measured emissions. It is just that there are a lot of peaks in the simulated that occur between samplings.

We reconsidered this part of the sentence and you are correct with regards to the text of Sommer et al (2016). We thus decided to remove the citation.

 One reason is the poor representation of soil

Line 569: I guess this is somewhat true, in that maize mono-cropping will still produce some GHG emissions. However what is the difference between the ISFM practices and the "typical" treatment (what is typical? No inputs? No N input and a small amount of FYM)? It seems like adding some inorganic N with 1.2 T C increased yields, without increasing yield scaled emissions compared with 0N 0C and compared with 0n 1.2T C. So even though it is not exactly "negative emission technology" it still seems to be an improvement.

Yes, we agree. To highlight this, we adjusted the sentence:

ner et al., 2020). However, the  large differences in the yield-scaled  GHG balance between treatments, such as  the 30 to 60% lower yield-scaled  GHG balance in the FYM 1.2+N treatment compared to the control-N treatment across the sites, indicate that ISFM has the potential to produce crops with relatively lower GHG emissions than no- or low-input input systems. Specifically, the ISFM treatments with low-emissions and high yields,

Line 570: why say "positive absolute" in stead of just "positive"?

We agree and changed it to "net positive".

Lines 578-580: While I agree that N fertilizer should only be applied to responsive soils, I'm not sure that is a conclusion of the date that you have here. If you look at yields, all the sites respond to N fertilizer (either mineral or organic). It is just that they seem to respond a bit differently, particularly in the N2O emissions, to the fertilizer applications. Besides, the 0N control also has much higher yield scaled GWP in Embu and Machanga, mainly related to loss of SOC, so I don't think the higher yield scaled emissions (compared with Sidada and Aludeka) with the +N treatments indicate that these shouldn't be fertilized. In fact, the decrease in yield-scaled GWP when adding N is greater at the sites in Central Kenya than they are at Sidada, which almost contradicts what you are saying here.

We fully agree and thus removed this sentence.

Line 610: Just mention which treatment had the lowest yield-scaled emissions (the mix of FYM +N) as the preferred INMS for Kenya.

After the model recalibration, we do not think that this comes out clear enough to put it into the conclusion. We hope this is agreeable to the editor and reviewers.

Table A1: can you add the sand content as well?

Done

Figure A3: what depth are you using to calculate the stocks? You mention 15 cm depth in some locations, but you also mention that DayCent uses 20 cm depth. And, I am having a hard time seeing how the Machanga site lost so much of its C. at 20 cm depth a soil with a C content of 0.3 and a BD of 1.51 would have about 10 t C per ha. And you are saying here that it lost about 10 t per ha (or essentially all of its soil C). Is my math off (wouldn't be the first time).

We added the soil depth to the Figure (now A7). To your question of Machanga – the site had initially about 20 t C ha$^{-1}$. We realized that the initial soil C and N in the Table A1 were still data from the original reference profile plot description, which consisted of a single measurement per horizon at each site, conducted before the trials were established. However, at the time of trial establishments, further soil C and N measurements were done in each plot, resulting in slightly different initial C and N contents (see below). We had actually used these more accurate measures of SOC to match the SOC stocks in DayCent. We now updated Table A1 for consistency. Thanks for bringing this to our attention!

| Soil characteristics | Embu | Machanga | Sidada | Aludeka |
|---:|---|---|---|---|
| Latitude | -0.517 | -0.793 | 0.143 | 0.574 |
| Longitude | 37.459 | 37.664 | 34.422 | 34.191 |
| Initial soil C (%) | 3.1 | 0.8 | 2.6 | 0.7 |
| Initial N (%) | 0.3 | 0.05 | 0.21 | 0.06 |

Figure A6: the figure caption needs to be re-done. For example, the second sentence is missing a word somewhere (perhaps "was" before "insensitive"?). And secondly, are you sure about the 50/50 split application? You were calibrating to data where the split application was 40 kg N at planting and 80 kg after ~ 6 weeks (see line 107-108; also line 165).

Thanks for bringing this to our attention. We have rewritten the caption. We are sure, about the evenly split application – this was for the technical reason that DayCent did not allow to go for higher N applications than 200kg N per one application (and we wanted to test until 400).

[Figure]

**Figure A5.**  Yield response curve  of DayCent to varying levels of mineral N application (control + N treatment, without organic resources) using the calibrated DayCent parameters. Displayed are the simulated mean yields across all simulated seasons (32 in Sidada and Aludeka, 38 seasons in Embu and Machanga). The amount of mineral N  applied per season in  the simulations was evenly split  between the  actual application dates of mineral N in each season at each site.

Figure A10, can you increase the font size in the figure please?

We have done so and removed the figures from Machanga, because we realized that due to erosion and runoff issues, the $N_2O$ and SOC dynamics at this site were not very well represented by DayCent.

---

## Referee Report (RR1)

**A robust DayCent model calibration to assess the potential impact of integrated soil fertility management on maize yields, soil carbon stocks and greenhouse gas emissions in Kenya**

The study uses a Bayesian calibration approach (sampling importance resampling) with leave-one-site-out cross-validation to calibrate the biogeogchemical model Daycent to yields, biomass and SOC at four sites in Kenya. The authors addressed adequately the suggestions of previous reviewers and the community comment and improved the quality of the manuscript. Overall, the manuscript is well-written, methods are sound and described sufficiently. I have some suggestions and comments (see below). I suggest to publish the manuscript after minor revisions.

I refer to the track changes version with my line numbers.

**Abstract**
L33: Daycent is well-suited to estimate the impact of ISFM
The impact of ISFM on what? -> Please add.

**Introduction**

L82: so a propagation of errors is possible in upscaling exercises
We can be sure the errors propagate in upscaling exercises even if you don't track them, you probably mean: So an **estimation** of uncertainties is possible in upscaling exercises

L103: ISFM can…. but at the same time **mitigate CO2 emissions due to the mineralization of SOC**
That's an ambiguous formulation, please rephrase to an unmistakable sentence.

L105: displaying the confidence in model parameters by Bayesian calibration
Not clear what you mean by that

**Methods**

L253: ,. taken calculated with the equation
   -   Typo, remove 'taken'

L495: in CO2 eq kg-1 maize grain yield
   -   in kg CO2 eq kg-1 maize grain yield

**Results**

Figure 2

My visual impression is that prior and posterior distributions are quite similar.
Why is the posterior less narrow in Figure 2 compared to the prior? Wouldn't one expect the calibration to constrain the parameters and give a narrower posterior compared to the prior?

Figure 2 caption: Not clear what you want to say by 'uncertainty-based Bayesian model calibration', but since this is not a term generally used or a method description, I would leave out the term 'uncertainty-based'.

Figure 7: 'the black solid line the simulation by the best parameter set for each site'
You did not calibrate by site, but the caption can be understood as if you did. Since the panels are per site anyway, I would recommend to omit 'for each site' here in the caption.

Figure 8: Credibility intervals for cumulative fluxes are quite narrow, and do not cover the 1:1 line. Are these really credibility intervals? Unlike the other figures, N2O was not calibrated. I think they are quite misleading here, since N2O was not included in the calibration so of course they remain narrow if you put narrow posterior distributions. Or is it variance that is displayed? Please add explanation in the caption.
For claiming that the posterior distributions are suitable for upscaling this must also be true for N2O, while my view for N2O a realistic uncertainty estimate is not shown.

Which ISFM method is simulated with highest accuracy etc?
If you target a robust fit for upscaling the effect of different ISFM methods, then it might be worth presenting the bias and rmse per treatment across site.

Figure 9: Please explain 9b in the caption (Mention 9 a b c in the caption.)

In several table & figure captions you explain the lowercase letters:
Same lowercase letters indicate the absence of a significant difference in XYZ ….
Easier to read would be a positive formulation: Different lowercase letters indicate a significant difference in XYZ between …

**Discussion & Conclusion**
These sections make sense to me and I have no further comments.

---

## Author Response (AR2)

Dear Moritz Laub,

you revised version of the manuscript: "A robust DayCent model calibration to assess the potential impact of integrated soil fertility management on maize yields, soil carbon stocks and greenhouse gas emissions in Kenya" has undergone a second round of reviews (3 reviewers). All reviewers are in agreement that the manuscript has improved considerable, with two suggesting some minor revisions including some clarifying questions and adjustments (revs 2 and 3) whereas the 3rd reviewer remains still critical (see the comments from rev 1 this text). This is particularly the case for the prior and posterior distribution which need to be addressed. This should be achievable relatively straight forward. Similarly the individual minor suggestions for the manuscript provided by all three reviewers can be incorporated right away. Following this, I am accepting the manuscript for publication in BG with subject to minor revisions.

with kind regards

Lutz Merbold

Dear Lutz Merbold,

Thank you for acknowledging the changes we made to improve the manuscript from the previous version. We have done our best to address the remaining concerns of the reviewers. We put specific focus on addressing the concerns that reviewer 2 had regarding the new method of initialization with measured SOC pools, the derivation of the coefficients of variation from the prior and the posterior, where reviewer 2 rightly pointed out an oversight from our side. Based on the feedback from reviewer 2 and reviewer 3, we also did calibration once more with wider priors (×1.5), which improved the results even further and led to the results showing a clear distinction between the prior and the posterior. We think that we have addressed all the important concerns of both reviewers with these changes. We hope that with these changes implemented, you will consider the manuscript to be acceptable for publication. Thank you very much for your efforts in handling the manuscript.

Kind regards on behalf of all coauthors,

Moritz Laub

Reviewer 1:

The paper describes the capability of DayCent model to simulate yield and SOC development of the different ISFM practices in SSA and its improvement after cal-val.

After the revisions made, the paper has strongly improved. All the raised issues were solved, the flow is now clear and figures were made more understundable for readers. Based on all these considerations, the manuscript can be considerd acceptable for publication in its present form.

Thank you for your positive assessment of the revisions that we made.

Reviewer 2:

The revision has addressed many of the issues raised in the first review. There has been a significant improvement in the content, flow, and structure, which has increased the readability of the manuscript. In addition, authors have incorporated two significant changes in the methodology section: (1) the selection of prior, and (2) the initialization of SOC pools. Reviews of this newly added section are provided in the subsequent paragraphs. Another area of concern is in the newly updated result section 3.2, which reported the posterior estimates from the inverse modeling using SIR algorithms. Here, the marginal distribution for the posterior is similar to the marginal distribution from the prior, indicating little or no influence of the dataset on the posterior suggesting nothing is learned from the data assimilation exercise. This is contrary to the title's claim of "robust DayCent model calibration …". For these reasons, I recommend a major revision before recommendation for publication. More details are provided below:

Thank you for your feedback. We agree that the original title was not fitting anymore. We were actually using Bayesian calibration to derive model parameters which we can consider to be robust and it is not the calibration that is robust. Therefore, we changed the title to "Modelling integrated soil fertility management for maize production in Kenya using a Bayesian calibration of the DayCent model.

**Modelling integrated soil fertility management  for maize  production in Kenya using a Bayesian calibration of the DayCent model.**

Apart from that, we have increased the range of the prior, based on your comment. This improved the simulation outcomes further and the change between prior and posterior is now obvious, making it clear that the data informed the model parameters.

[Figure]

SOC pool initialization: One of the major changes involved replacing the long historical simulation with measured SOC, which was adjusted backward in time and initialized at the start of the Experiment. With the update, the posterior parameter can be defined as $p(\theta|D,M,MAOM)$ and now conditions on data (D), model (M) and measured mineral associated organic matter (MAOM). Therefore, any future simulation leveraging this study and aiming to understanding the regional or national GHG balance within SSA, as represented by the four-experiment station, now also requires measured or estimated value of MAOM. Therefore, the simulation approach, which requires MAOM measurement, may limit broader use of the model in SSA region. Furthermore, the initialization of

SOC pools in the process-based ecosystem model may introduce significant bias in the model's estimates of SOC stock changes (Fallon and Smith, 2000, Zhou et. al., 2023). Both methods: (1) long historical simulations to equilibrium and (2) initializing model pools with measurement—have been extensively studied and used in literature. In my personal viewpoint, both methods are equally valid given adequate testing and reasoning. Both methods have strengths and weaknesses. The strength led to higher accuracy, while the weaknesses can introduce significant bias and contributing toward higher uncertainty.

We agree that both approaches have their pros and cons. If there is good knowledge available on the land-use history, a model spin-up may be the better choice. While the model now relies on measured or estimated MAOC, which may introduce its own level of uncertainty, global predictions of MAOC, including for soils in Africa, are available (see Georgiou et al., 2022; specifically Supplementary Figure 16 of their article) and the ranges presented on their maps for the SSA region (i.e., that about 60-90% of SOC is MOAC) agree with our measurements. These maps are produced based on observed MOAC values, extrapolated based on measured gradients of precipitation, temperature, and vegetation. In theory, one could argue that these factors include similar information as a model spin-up, which is based on historical data about land use and includes local weather. In the context of SSA, and in Kenya specifically, we consider that data on land-use history is subject to more uncertainty than the currently available maps of MOAC (e.g, the soilgrids have more than 2000 profiles in Kenya, while a documentation on historical land use is not readily available). This implies that several assumptions have to be made about land-use history, for example by using expert opinion combined with a rules based approach (see e.g., Kamoni et al., 2007). With maps, the only choice to make is which map to use, making this approach more reproducible and favorable in our case. Nonetheless, we agree that it comes with its own uncertainties, which we have addressed in the discussion. Based on the literature you suggested, and our own observations at scale, we recommend to reduce this uncertainty by working with a baseline scenario and an improved scenario instead.

610  Nevertheless, soil property maps are also subject to uncertainty. For example, differences between different SOC maps used in model initialization propagate into differences in the changes in SOC stocks (Zhou et al., 2023). It was shown that uncertainty of the simulated effect of a soil management practice on the difference of SOC stocks compared to a counterfactual is lower than the uncertainty of the simulated temporal development of SOC stocks (Zhou et al., 2023). Therefore, it may be best practice to work with a baseline and an improved scenario. Both spinup and SOC map initialization

615 have their shortcomings and in the end the model user must make an informed decision on which initialization method they consider subject to less uncertainty, based on which data is locally available.

Prior distribution: Another significant change in the updated manuscript is the introducing of a new prior distribution during the SIR step. As a result, the revised manuscript uses two sets of prior distribution: a uniform prior for the global sensitivity analysis, and a Gaussian prior for the SIR step. This is uncommon and generally not accepted in Bayesian inferences, as the prior is considered our initial beliefs about an uncertain parameter before observing any data. Therefore, introducing two beliefs for the same set of parameters in the model is unusual.

We would not call the distribution of the parameter values used in the global sensitivity analysis (GSA) a "prior" in the Bayesian sense and acknowledge that this has been poorly formulated in the previous version of the manuscript. For example, GSA is only constrained by a maximum and a minimum value – and thus any other distribution than uniform is not possible (a poor use of prior knowledge). For us, GSA was simply a preselection procedure to make the Bayesian calibration computationally feasible. Many studies commonly use background knowledge to select which parameters to calibrate and then use Gaussian priors in the calibration process (e.g., Menichetti et al., 2020; Ťupek et al., 2019). Therefore, if we consider the GSA solely as a preselection step, there

appears to be no impediment to subsequently conducting a Bayesian calibration using a Gaussian prior, if knowledge about the model parameters exists. The text of the methods section has been changed, accordingly.

distributions were The ranges used for the global sensitivity analysis , with the upper and lower parameter boundaries were centered around the initial parameter value obtained as described above (section 2.3.2). We based the global sensitivity analysis parameter ranges The upper and lower parameter boundaries were based on previous sensitivity analyses (e.g. Necpálová

300     et al., 2015; Gurung et al., 2020), plausible ranges reported in the DayCent manual and variations observed in different maize

Further, our GSA did not make use of the data. In contrast to Gurung et al. (2020), the sensitivity in our study was calculated based on the mean yields and AGB, and end-of-simulation SOC stocks across sites, and not on the mismatch between the simulated and observed values. Hence, we do not see that performing a Bayesian calibration afterward violates the assumption that that Gaussian prior is formulated before observing the simulations compared to the data. Our prior was mainly informed by previous studies (see comments to your next point, below).

The authors selected coefficients of variations ranging between 5% and 30% for DayCent parameters based on the level of range provided in its manual. However, they did not provide details on how these selection for coefficient of variation satisfy one of the requirements for SIR or similar method, which is that the prior range should cover the entire range of the posterior (Galman, 2014). Also, more detail should be provided on the choice for the coefficient of variation to convincing demonstrate that the empirical data suggest that the prior range is adequate enough for the theoretical understanding of these parameters.

Thanks for bringing this point to our attention. We agree that we had insufficiently described this in the methods. In selecting the coefficients of variation, we considered prior knowledge from pervious Bayesian calibration exercises performed on DayCent. Our aim was to choose a coefficient of variation per range level in a way that our prior covered the range that previous Bayesian calibrations on DayCent had as the posterior. Due to your comment, we have now increased the range of the prior to account for the uncertainty of applying DayCent in tropical conditions by increasing the coefficient of variation by a factor of 1.5. We now added a detailed explanation of all this to the text.

To ensure computational efficiency, we used informed Gaussian priors that were centered around the standard parameter values of DayCent, with different coefficients of variation of 0.05, 0.1, based on different observed ranges in previous studies. To make optimal use of existing knowledge about the parameters, the selected coefficients of variation per range were

365     initially based on previous studies that had performed Bayesian calibration of the DayCent model. The coefficients of variation were chosen in a way that the prior from our study covered the whole range of the posterior from previous studies and then multiplied by a factor of 1.5 to account for the additional uncertainty that arose from applying DayCent at tropical sites. The studies of Gurung et al. (2020) and Mathers et al. (2023) were the basis to derive the coefficient of variation for the parameters dec4, dec5(2), clteff(1,2,&4), ps1co2(1&2) & rsplig, and pmco2(1&2). The study of Yang et al. (2021) was the basis for the

370     parameters ppdf(1) and ppdf(2), and the study of (Necpálová et al., 2015, though not being Bayesian) was the basis for the parameters aneref(3) and fwloss(4). For himax and prdx(1), we looked into the default parameters of annual crops in DayCent, to assure that the whole range of values (0.30-0.55, and 1.1-3.5: 1.7-2.5, respectively) was covered by the prior. The final coefficients of variation were 0.08, 0.15, 0.25 and 0.3 0.23, 0.38 and 0.45 for parameters with very small, small, moderate, large and very large ranges (Table 1). For the newly introduced parameters, we used larger large coefficients of variation,

375     namely 0.38 for $SL_t$ and 1 for $IC_{t,MAOC}$ and 0.35 for SL, the reason for the latter being an initial test, in which $IC_{t,MAOC}$ was set to -0.3 instead of -0.1, which proved to be too low, but the uncertainty range with a standard deviation of 0.1 proved to be reasonable. Additionally, all parameters were constrained to remain within their physically sensible limits (i.e., not <0 for all and not >1 for those representing fractions).

Posterior distribution: The manuscript calibrated 13 model parameters after conducting a parameter screening using the GSA. In section 3.2 of the results, the posterior was presented in Figure 2. Throughout the text, a single parameter estimate was provided, but it was not specified which statistics (e.g., mean, mode, median, etc.) was presented. The posterior should be summarized with sufficient statistics, such as the mean, standard deviation, and 95% credible intervals. If the single parameter estimates the mean, mode, or median, there is a significant disagreement between the text and the figure for parameters clteff(1,2,&4) and pmco2(1&2). The reported posterior estimates of 19.1 and 0.82 fall well outside the curve region with higher density. I believe this could be a miscalculation or misinterpretation, and should be thoroughly investigated.

Thanks for pointing this out. The presented parameter set is neither mean, mode, nor the median, it is the parameter set from the posterior that had the highest likelihood, based on data from all four sites combined (i.e., not leaving any site out – the final step after cross-validation). While we had stated this in table 2, the description was very brief and did not appear in the main text. We see how this information could be easily overlooked by the reader. Therefore, we now added a better explanation to table 1. Further, we added the requested statistics to table 1:

[revised manuscript text omitted]

Some minor comments and corrections:

Line 20-23: The author claim that: "The model performance and the match between the cross-evaluation posterior credibility intervals for different sites indicated the robustness of the model

parameterization and the reliability of the DayCent model for spatial upscaling of simulation." However, the manuscript did not perform a large-scale simulation, and the claim for "spatial upscaling" should be removed or justified.

We agree that this was misleading, and thus refined the sentence to "for the conditions in Kenya".

> one parameter. Together with the model performance for the different sites in cross-validation, this indicated the robustness of the DayCent model parameterization and
* * *
20  its reliability for the conditions in Kenya. While DayCent poorly reproduced daily $N_2O$ emissions

Line 23: provide quantitative values (i.e., EF for daily N2O) instead of just mentioning negative value.

Thanks, we did so.

> 20  its reliability for the conditions in Kenya. While DayCent poorly reproduced daily $N_2O$ emissions (with EF ranging between -0.44 and -0.03 by site), cumulative seasonal $N_2O$ emissions were simulated more accurately (EF ranging between 0.06 and 0.69 by site). The simulated yield-scaled GHG

Line 70: The terms "validated" and "evaluated" were used interchangeably throughout the manuscript. For instance, in line 9, "cross-evaluation" is used but in line 164, "cross-validation" is used.

Thanks for pointing this out. We changed this to "evaluation of the model" throughout the text, with the method of evaluation being named "cross-validation". Any "cross-evaluation" was removed.

term "C sequestration" instead of "mineralization of SOC"

The ambiguity of this statement was also pointed out by other reviewers and we changed it accordingly:

> soil and optimizing crop yield (that is, sustainable intensification). ISFM can be a source of $N_2O$ to the atmosphere (Leitner
> 80  et al., 2020) but compared to standard practices, it reduces SOC losses or even increases SOC (Laub et al., 2023a), thereby mitigating $CO_2$ emissions.

Line 134: Should this

"Tithonia diversifolia (TD) green manure and Calliandra calothyrsus (CC) prunings, low quality stover of Zea mays (MS) and sawdust from Grevillea robusta trees (SD), locally available farmyard manure (FYM) and a control treatment"

be written as following.

"Tithonia diversifolia (TD) green manure, Calliandra calothyrsus (CC) prunings, low quality stover of Zea mays (MS), sawdust from Grevillea robusta trees (SD), locally available farmyard manure (FYM) and a control treatment"

Thanks! This was in fact not very clear and has been revised:

105 ments, with two crops per year, one in the long rainy season and one in the short rainy season. The experimental design was identical at all four sites and has been described in detail in earlier publications (Chivenge et al., 2009; Gentile et al., 2011; Laub et al., 2023a, b). Organic resource treatments consisted of high quality *Tithonia diversifolia* (TD) green manure, high quality *Calliandra calothyrsus* (CC) prunings, low quality stover of *Zea mays* (MS), low quality sawdust from *Grevillea robusta* trees (SD), locally available farmyard manure (FYM), and a control treatment (CT) without organic resource additions.

In Table 1. values for model parameters and coefficient of variation seems truncated given the table description (i.e., parameter values and coefficient of variations were missing)

The table description has been updated to specify that not all parameters considered in the GSA are shown in Table 1.

**Table 1.** DayCent model parameters and the coefficient of variation used in the calibration. Displayed are parameters considered for calibration due to total sensitivity index > 2.5% (top) and with a total sensitivity index > 1% (bottom). The remainder of parameters (<1%) are not included in this table and can be found in the supplementary (Table A3). The presented calibrated parameter values correspond to the single parameter set with the highest likelihood, which was derived by using the data from all four sites combined. The posterior was also derived by using the data from all four sites combined. Abbreviations: CV, coefficient of variation; SD, standard deviation, 95% CI, 95% credibility interval; Ly, langley.

Equation before line 185, if SOC stock estimates are for 0-30 cm as IPCC-recommended, it should be:

$$\llbracket SOC \rrbracket \_30 (kg \; \llbracket ha \rrbracket \^{(-1)} )= (1- \beta\^30)/(1-\beta\^15 )* \llbracket SOC \rrbracket \_15$$

Provide the value used for beta^15 and beta^30 used in the equation. The equation number is also missing.

The comment likely refers to the track change version of the article, in which the equation was still visible (in red, indicating that it had been removed in this version).

Because we have removed the equation and went for a different approach to calculate SOC stocks in the first 30 cm of soil, it is not longer necessary to provide beta. This new approach is described in lines 146ff (track-change version) or 139ff (clean version).

Line-261, it is a little confusing and not clear what the author wants to convey. Specifically, data availability and which model parameters and value used for initialization.

The sentences were overhauled:

 It was assumed that the organic resource inputs had the same properties across all sites (i.e., mean values of lignin contents and C/N  (ratios per organic resource were assumed; Table A2). This approach was used because measurements were not

205    available for all sites and years, and was justified as an analysis of variance of data from the years 2002, 2003, 2004, 2005 and

Line-266: It was not clear whether the author's discussion about aboveground biomass (AGB), yield (Y), and harvest index (HI) is based on the measured data or modeled values. In DayCent Y = HI*AGB (for grain crops). However, the parameter HIMAX (maximum harvest index) is adjusted due to stress to (HI <= HIMAX).

This was about the measured data. We added this to the sentence, as follows:

This was the mean value of measured grain C content across sites (standard deviation 1.8%) in the short rainy season 2018 and long rainy season 2019 (data not shown). Given the strong correlation between maize grain yield and aboveground biomass in

210    the measured data (r = 0.87), the aboveground biomass data was transformed to harvest index data for the model calibration

Line-395: Please clarify what multiply/divided by 3 and 10 means. Maybe it is self-explanatory when full view of Table-1 is available.

This sentence was reformulated to remove any ambiguity.

For parameters with large and very large ranges, the upper  boundaries were the initial parameter values multiplied by 3 and 10, respectively, the lower boundaries were the initial parameter values divided by 3 and 10, respectively. The parameter

Equaton-3: The Likelihood function provided in Equation-3 is applicable to only one type of measurement, such as Yield or SOC. Please provide details on how multiple likelihoods—for SOC, Yield, and Harvest Index were combined, if at all, for the final Bayesian calibration. If they were not combined, please provide an explanation.

In fact, we did combine all types of measurements in the same likelihood function. This was possible by supplying a weighting factor (the inverse of the standard deviation; SD) to the mixed effects model. The formula we used in R for the loglikelihood was:

logLik(lmer(resid~-1+(1|Site/date),weights = (1/SD),data=EC_HI_SOC))

We have further clarified this in the text.

325    the inverse of the median standard deviation (of each type of measurement at each site) as weight. By using the inverse of the standard deviation of each type of measurement as weight of the zero-intercept model, it is possible to include different types of measurements into the same likelihood function. This is similar to what is done in weighted analyses commonly performed in meta-analyses (Möhring and Piepho, 2009). The logLik() function is then used to extract the log-likelihood, which is transformed to the likelihood by raising $e$ to the power of the log-likelihood.

Line 571: The posterior credibility intervals in analog to confidence intervals in frequentist statistics and posterior prediction interval analog to prediction intervals. The coverage probability (i.e., 95% of observed) within the 95% Posterior prediction interval is only valid comparison but not with posterior credibility intervals (note that posterior credibility interval < posterior prediction interval).

We agree that these sections were a bit misleading and have removed these comparisons from the text.

465    were eliminated  at Sidada, reduced at Embu, but increased  at Machanga.

 While DayCent could

500     -1.9 compared to -4.8 before calibration). DayCent performed well in simulating the variability

Prove all the missing equations used in the analysis, (one such example is the equation for aggregated model output for the GSA) in the supplementary section.

Thanks for this suggestion. We went through the whole manuscript with a focus on this issue and added several equations to the supplementary section.

**A1 Pedotranfer functions to derive the hydraulic parameters**

The equations used to calculate the soil hydraulic properties were based on the pedotransfer functions of Hodnett and Tomasella (2002):

$$\theta_r = 0.22733 - 0.00164 \times Sa + 0.00235 \times CEC - 0.00831 \times pH + 1.8 \times 10^{-5} \times Cl^2 + 2.6 \times 10^{-5} \times Sa \times Cl \tag{A1}$$

$$\theta_s = 0.81799 + 9.9 \times 10^{-4} \times Cl - 0.3142 \times BD + 1.8 \times 10^{-4} \times CEC + 0.00451 \times pH - 5 \times 10^{-6} \times Sa \times Cl \tag{A2}$$

$$ln(\alpha) = -0.02294 - 0.03526 \times Si + 0.024 \times SOC - 7.6 \times 10^{-3} \times CEC - 0.11331 \times pH \tag{A3}$$

$$ln(n) = 0.62986 - 0.00833 \times Cl - 0.00529 \times SOC + 0.00593 \times pH + 7 \times 10^{-5} \times Cl^2 - 1.4 \times 10^{-4} \times Sa \times Si \tag{A4}$$

Here, $\theta_r$, $\theta_s$, $\alpha$, and $n$ are the soil water retention parameters of van Genuchten (1982), $Sa$, $Si$ and $Cl$ are Sand, Silt, and Clay content (in %), $BD$ is the bulk density (t m$^{-3}$) $CEC$ is the cation exchange capacity (cmol kg$^{-1}$), $pH$ is the soil pH measured in $H_2O$, and $SOC$ is the SOC content (g kg$^{-1}$).

The wilting point (WP) and field capacity (FC) values were then calculated as

$$WP = \theta_r + \frac{(\theta_s - \theta_r)}{(1 + (\alpha \times |-15000|)^n)^{1 - \frac{1}{n}}} \tag{A5}$$

$$FC = \theta_r + \frac{(\theta_s - \theta_r)}{(1 + (\alpha \times |-330|)^n)^{1 - \frac{1}{n}}} \tag{A6}$$

$K_S$ was calculated using the Saxton and Rawls (2006) equation, with values of the water retention curve, $\alpha$ and $n$ (van Genuchten, 1982), calculated with the equation from Hodnett and Tomasella (2002):

$$\lambda = \frac{\ln(FC) - \ln(WP)}{\ln(1500) - \ln(33)} \tag{A7}$$

$$K_S = \frac{1930 \times (\theta_s - FC)^{(3 - \lambda)}}{10 \times 60 \times 60} \tag{A8}$$

Here, $\lambda$ is the slope of logarithmic tension-moisture curve and $K_S$ is the saturated water conductivity (cm s$^{-1}$).

**A2 Equations for the global sensitivity analysis**

The means across all sites, which were used in the GSA were calculated as follows:

$$Mean = \frac{1}{n} \sum_{j=1}^{n} \frac{\sum_{i=1}^{N} Mod_{ij}}{N} \tag{A9}$$

Here $n$ is the number of sites (4), $N$ is the number of modelled values per site, and $Mod_{ij}$ are the individually modelled values. For aboveground biomass and grain yield, $N$ corresponded to the total number of modelled yields and biomass at all treatments and seasons. For SOC and soil N stock $N$ corresponded to the total number of treatments per site. The reason is that because changes in SOC and soil N stocks are expected to be stronger the longer a simulation lasts, only the stocks from the end of the simulation were used.

Reviewer 3:

The study uses a Bayesian calibration approach (sampling importance resampling) with leave -one-site-out cross-validation to calibrate the biogeochemical model Daycent to yields, biomass and SOC at four sites in Kenya. The authors addressed adequately the suggestions of previous reviewers and the community comment and improved the quality of the manuscript. Overall, the manuscript is well-written, methods are sound and described sufficiently, Results are well described and followed by a sensible Discussion. I have some suggestions and comments (see below), and suggest to publish the manuscript after these minor revisions.

Thank you for your overall positive assessment of our manuscript and for the constructive feedback that you provided. We have incorporated the necessary changes, based on your feedback. See the details below.

I refer to the track changes version with my line numbers.

Abstract L33: Daycent is well-suited to estimate the impact of ISFM The impact of ISFM on what? -> Please add

Thanks for spotting this unclear formulation. We added "on maize yields and SOC changes" to the sentence.

25  application of mineral N and of manure at a  rate of 1.2t C ha$^{-1}$ yr$^{-1}$. In conclusion, our results indicate that DayCent is well-suited  for estimating the impact of ISFM  on maize yield and SOC changes. They also indicate that the

Introduction

L82: so a propagation of errors is possible in upscaling exercises

We can be sure the errors propagate in upscaling exercises even if you don't track them, you probably mean: So an estimation of uncertainties is possible in upscaling exercises

You are right, this was not formulated well. Your suggestion was incorporated.

65  long-term experiments. Ideally, this calibration would include the uncertainty in the model parameters and model outputs (Clifford et al., 2014), so  an estimation of uncertainties is possible in upscaling exercises (Stella et al., 2019). This is especially relevant given a recent study showing considerable uncertainty in DayCent's SOM turnover rates,

L103: ISFM can…. but at the same time mitigate CO2 emissions due to the mineralization of SOC

That's an ambiguous formulation, please rephrase to an unmistakable sentence.

Thanks. We rephrased as follows:

soil and optimizing crop yield (that is, sustainable intensification). ISFM can be a source of N$_2$O to the atmosphere (Leitner et al., 2020) but compared to standard practices, it reduces SOC losses or even increases SOC (Laub et al., 2023a), thereby mitigating CO$_2$ emissions.

L105: displaying the confidence in model parameters by Bayesian calibration

Not clear what you mean by that

We reformulated this as follows:

85 Kenyan conditions using experimental data from four long-term experiments, displaying the  uncertainty of model parameters by Bayesian calibration, and (iii) to use the calibrated model to gain understanding of the GHG balance of the different ISFM treatments.

Methods

L253: ,. taken calculated with the equation

- Typo, remove 'taken'

Removed, thanks for spotting this.

method to estimate $K_{sS}$, $K_{sS}$ was calculated using the Saxton and Rawls (2006) equation, with values of the water retention curve, $\alpha$ and $n$ (van Genuchten, 1982),  calculated with the equation from Hodnett and Tomasella (2002). The equations
200 can be found in the supplementary material (A1).

L495: in CO2 eq kg-1 maize grain yield

- in kg CO2 eq kg-1 maize grain yield

We interpreted this comment as a hint to missing units and added also the unit of the annual GHG balance.

400 Here, $\Delta SOC$ is the change in SOC content (kg C ha$^{-1}$ yr$^{-1}$), $N_2O$ the cumulative $N_2O$ flux (kg $N_2O$ ha$^{-1}$ yr$^{-1}$). The $CH_4$ oxidation capacity was not considered, because it usually makes a very limited contribution to GHG balance in rainfed cropping systems (Lee et al., 2020) and we did not have data to evaluate the reliability of this simulated flux. In addition to the net annual GHG balance (in t $CO_2$eq ha$^{-1}$ yr$^{-1}$), we calculated the yield-scaled GHG balance (in $CO_2$eq kg$^{-1}$ maize grain yield) by dividing the cumulative GHG balance over the entire simulation period by cumulative simulated yields (dry matter base).

Results

Figure 2

My visual impression is that prior and posterior distributions are quite similar.

Why is the posterior less narrow in Figure 2 compared to the prior? Wouldn't one expect the calibration to constrain the parameters and give a narrower posterior compared to the prior?

We agree that they were very similar. Based on your comment and a comment from reviewer 2, we therefore increased the range of the prior to by increasing the coefficient of variation by a factor of 1.5. This led to the data clearly constraining the posterior. We have updated the results accordingly.

[Figure]

Following the global sensitivity analysis, 13 selected model parameters were calibrated using Gaussian priors which were centered around the initial parameter value, with standard deviations according to the uncertainty ranges (Table 1). The ranges

425  of the prior- and the posterior distributions, using It should be noted that the presented calibrated parameter values in Table 1 correspond to the single best parameter set for all four sites combined (i.e., the parameter set that had the highest likelihood in the case of no cross-validation).

Compared to the range of the prior parameter sets, the ranges of the posterior parameter sets calibrated with data from all four sites changed significantly for the parameters fwloss(4) and pmco2(1&2), had a similar mean value but a more narrow

430  distribution for the parameters IC$_{MAOC}$, prdx(1), and ps1co2(1&2)&rsplig, and changed slightly for the parameters dec4, were similar . Also the four different posterior distributions from dec5(2), ppdf(1), ppdf(2), and himax (Fig. 2). The posterior parameter sets of the leave-one-site-out cross-validations were largely similar to each other (in agreement with each other and with the posterior parameter sets calibrated with data from all four sites. The exception was the parameter pmco2(1&2), which was centered around 0.55 for the case that the Aludeka site was left out and around 0.70 for all other cases (Fig. 2).

435  However, several parameters slightly shifted from their initial values to the best parameter values across

The parameter that changed most strongly in the parameter sets calibrated with data from all four sites . The strongest differences between the initial and calibrated values existed for the potential maximum maize productivity per radiation (prdx(1 was the scaling factor for potential evapotranspiration (fwloss(4); from 2.25 to 1.85 g C m$^{-2}$ langley$^{-1}$) , the parameter representing the increase of SOM turnover after tillage (clteff(0.75 to 0.94) thereby not including the initial value in the

440  95% posterior credibility interval (0.81 to 0.99; Table 1). Also the CUE of metabolic litter was reduced (by an increase of pmco2(1,&2) from 0.54 to 0.91 g g$^{-1}$) but the initial value was still within the 95% posterior credibility interval (0.48 to 0.91 g g$^{-1}$). The turnover rates increased for both the slow SOM pool (dec5(2,&4) ; from 10 to 19.1) . An increase of the turnover rate of the ); from 0.10 to 0.13 g g$^{-1}$ yr$^{-1}$) and the passive SOM pool (dec4; from 0.0035 to 0.0056 0.0060 g g$^{-1}$ yr$^{-1}$)was partly , which was however counterbalanced by a decrease in the turnover rate of the slow SOM pool (dec5(reduction of the effect

445  of tillage on decomposition (clteff(1,2,&4); from 0.10 to 0.06 10 to 5) and all three of these parameters contained their initial values in the 95% posterior credibility intervals. The maximum harvest index slightly increased (himax; from 0.40 to 0.43 g g$^{-1}$yr) and so did the potential production of maize per unit of light interception (prdx(1); from 2.25 to 2.62 g C m$^{-2}$ langley$^{-1}$). Furthermore the loss of carbon from the metabolic litter pool upon decomposition was significantly increased (pmco2(Finally, the optimum temperature for maize growth decreased (ppdf(1&); from 30 to 28.6 °C), while the maximum temperature for

450  maize growth increased (ppdf(2); from 0.54 to 0.82 g g$^{-1}$). The 45 to 47.1 °C). Of the two parameters that translated measured MAOC into SOC in the passive SOM poolwere altered in opposite directions (IC$_{MAOC}$ , only IC$_{MAOC}$ was altered (from -0.1 to -0.21 -0.02 g g$^{-1}$; and SL$_t$, from -0.005 to -0.0024 ) but the initial value was still in the 95% posterior credibility intervals(-0.25 to 0.06 g g$^{-1}$yr$^{-1}$). Overall, the parameter correlations in the posterior parameter set across the four sites were minimal, and in no case stronger than 0.2 (low for soil carbon related parameters (around 0.4 at maximum), but stronger correlations existed

455  between plant productivity-related parameters (e.g., -0.7 between himax and prdx(1) and 0.58 between ppdf(1) and ppdf(2); Fig. A3).

Figure 2 caption: Not clear what you want to say by 'uncertainty-based Bayesian model calibration', but since this is not a term generally used or a method description, I would leave out the term 'uncertainty-based'.

We changed this formulation as follows:

**Figure 2.** Prior compared to the posterior model parameter distribution resulting from the uncertainty-based Bayesian model calibration of DayCent using data from all sites combined (top) and the leave-one-site-out cross-validation (bottom). The uncertainty ranges of the priors were based on the range of parameter values found in the literature and increased by a factor of 1.5, because DayCent was applied to tropical site, while historically, it was mostly calibrated based on temperate sites. Dashed vertical lines represent the values of the initially selected parameter set. The posterior distributions are based on all four study sites combined. For the description of the parameters see Table 1.

Figure 7: 'the black solid line the simulation by the best parameter set for each site' You did not calibrate by site, but the caption can be understood as if you did. Since the panels are per site anyway, I would recommend to omit 'for each site' here in the caption.

Thanks for spotting this ambiguity. We omitted "for each site" as suggested:

**Figure 7.** Measured (dots) versus simulated SOC stocks over time at the four study sites for the different organic resource and chemical nitrogen fertilizer treatments. Error bars represent 95% confidence intervals for measured data, the black solid line the simulation by the best parameter set. Grey bands represent the 95% credibility intervals of the model posterior simulations, calibrated by leave-one-site-out cross-validation. Note that due to intense soil erosion, data from Machanga was not used in the calibration process.

Figure 8: Credibility intervals for cumulative fluxes are quite narrow, and do not cover the 1:1 line. Are these really credibility intervals? Unlike the other figures, N2O was not calibrated. I think they are quite misleading here, since N2O was not included in the calibration so of course they remain narrow if you put narrow posterior distributions. Or is it variance that is displayed? Please add explanation in the caption.

You are right and we added this fact to the explanation:

**Figure 8.** Simulated compared to measured $N_2O$ emissions at the four study sites for the different organic resource and chemical nitrogen fertilizer treatments, based on the calibrated parameter set using leave-one-site-out cross-validation. Displayed are the measured versus modelled per treatment for the days where measurements were conducted (top) and for the mean of cumulative flux measurements per season using the trapeziod method (bottom). The 808 data points (top) correspond to the daily measurements from the experimental treatments over one to two seasons, depending on the site. Symbols represent the different organic resource and chemical nitrogen fertilizer treatments. Error bars represent 95% confidence intervals (measurements) and credibility intervals (simulations). Note that the credibility intervals are only informed by yield, SOC and harvest index data and therefore do not represent the full uncertainty of $N_2O$ emissions. Abbreviations: EF, Nash-Sutcliffe  model efficiency; RMSE, root mean squared error; SB, squared bias; NU, non-unity slope; LC, lack of correlation.

For claiming that the posterior distributions are suitable for upscaling this must also be true for N2O, while my view for N2O a realistic uncertainty estimate is not shown.

This is true, we added a sentence on this in the results and another one in the discussion:

per season, there was a better agreement between the simulated and measured values. All sites, except Machanga, showed positive model efficiencies (highest in Embu, 0.62; lowest in Sidada, 0.03), but generally underestimated the uncertainty around cumulative $N_2O$ emissions (Fig. 8). Additionally, the correlation between simulated and measured $N_2O$ emissions was

655   Nonetheless, the fact that the uncertainty around predicted cumulative $N_2O$ emissions was lower than the uncertainties of the measurements indicates that the posterior, which was only calibrated with yield, SOC, and harvest index data, underestimates the uncertainty around $N_2O$ emission predictions. Thus, although DayCent's simulations of $N_2O$ emissions are superior to using emission factor approaches (dos Reis Martins et al., 2022), simulating $N_2O$ emissions remains challenging and highly uncertain due to the complexity of the processes involved and their high temporal and spatial variability. Given the limited bias

Which ISFM method is simulated with highest accuracy etc?

If you target a robust fit for upscaling the effect of different ISFM methods, then it might be worth presenting the bias and rmse per treatment across site.

Thanks for this suggestion. We agree and have added this to the Supplementary Section.

[Figure]

**Figure A7.** Treatment-specific simulated compared to measured maize grain yields at the four study sites for the calibrated parameter set by leave-one-site-out cross-validation. Abbreviations: EF, Nash-Sutcliffe model efficiency; RMSE, root mean squared error; SB, squared bias; NU, non-unity slope; LC, lack of correlation.

[Figure]

**Figure A8.** Treatment-specific simulated compared to measured changes in SOC stocks (without the Machanga site) since the start of the experiment at the four study sites for the calibrated parameter set by leave-one-site-out cross-validation. Abbreviations: EF, Nash-Sutcliffe model efficiency; RMSE, root mean squared error; SB, squared bias; NU, non-unity slope; LC, lack of correlation.

We further added a few sentences addressing these new results in the results section:

+N treatments. Nonetheless, DayCent was able to acceptably simulate the variability of grain yields across sites by organic resource and mineral N treatment (model efficiencies between 0.30 and 0.54; with values for control -N (0.08) and sawdust -N (0.18) being the exception; Fig A7). Interestingly, DayCent poorly distinguished the mean yields and aboveground biomass of treatments with high compared to very high rates of N inputs (i.e., the differences between the different organic resources and the control within the +N treatment). An additional test of the model sensitivity of mean yields to different levels of mineral N

475

500  -1.9 compared to -4.8 before calibration). DayCent performed well in simulating the variability of the changes in SOC stocks across sites, evaluated by organic resource and mineral N treatment (also computed without Machanga). With the exception of the  treatments farmyard manure ±N, maize stover ±N, and sawdust -N model efficiencies were between 0.51 and 0.79 (with RSME between 3.2 and 4.3 t ha⁻¹; Fig A8) with the highest performance for the control +N (0.79) and control -N (0.69) treatments. The other treatments still had positive model

505 efficiencies (0.15 to 0.42), but the SOC losses of the farmyard manure treatments were overestimated (EF of 0.15 for -N, 0.29 for +N, RSME of 5.1 and 5.3).

And in the discussion section:

able for upscaling of model simulations. Specifically, the yields of the ISFM treatments applying farmyard manure, *Calliandra*, and *Tithonia* were simulated well, both with and without the addition of mineral N fertilizer (Fig A7). The changes in SOC stocks for the control, *Calliandra*, and *Tithonia* treatments were also simulated well across sites, while DayCent underestimates the SOC buildup from farmyard manure treatments (Fig A8). However, on should keep in mind that the season-to-season yield

570   variability is captured less accurately than the mean yields (lower RMSE) and that changes in SOC are better represented at

Figure 9: Please explain 9b in the caption (Mention 9 a b c in the caption.)

Thank you. We added this to the caption:

**Figure 9.** Cumulative simulated greenhouse gas (GHG) balance of $N_2O$ emissions and $CO_2$ emissions due to loss of SOC at the four study sites for different organic resource and chemical nitrogen fertilizer treatments , combined throughout the simulated period (16 years for Aludeka/Sidada; 19 years for Embu/Machanga). Displayed are the GHG balance a) per area of land and year, b) the difference of GHG balance per area of land and year to a no-input treatment, and c) the yield-scaled GHG balance. The GHG balance is expressed in $CO_2$ equivalent over a 100-year horizon.

In several table & figure captions you explain the lowercase letters:

Same lowercase letters indicate the absence of a significant difference in XYZ …. Easier to read would be a positive formulation: Different lowercase letters indicate a significant difference in XYZ between …

We agree that the positive wording you suggest sounds simpler, but it is ambiguous and strictly speaking not correct (see Piepho, 2018). To make it simpler, we adjusted it to the formulation that Piepho (2018) suggested: "Means not sharing any letter are significantly different".

**Table A2.** Mean measured chemical characteristics (and 95% confidence intervals) of organic resources applied at all sites. Measurements were available from Embu and Machanga from 2002 to 2004, all sites from 2005 to 2007 and in 2018. Significant differences in residue properties were found between the different organic resources, but not between sites and years.  Mean values in a row  not sharing any lowercase letter are significantly different from each other ($p < 0.05$). Abbreviations: n.c. = not classified * according to Palm et al. (2001). The table is adopted from Laub et al. (2023a) under the creative common license 4: http://creativecommons.org/licenses/by/4.0/.

**Figure A2.** Subsoil SOC stocks for the 2.5-4.7 kt ha$^{-1}$ equivalent soil mass layer, corresponding to an approximate soil depth of 15-30 cm. Displayed are the least square means estimated by the linear mixed model described in (Laub et al., 2023a) for planted plots by treatment (left) and site (right). Error bars display the 95% confidence intervals.  Mean values at  each site not sharing any lowercase letter are significantly different from each other (left figure) . In the right figure , mean values per site not sharing any lowercase letter are significantly different from each other (all $p < 0.05$). Abbreviations: CC, *Calliandra*; CT, control; FYM, farmyard manure; MS, maize stover; SD, sawdust; TD, *Tithonia Diversifolia*. 0, 1.2 and 4 correspond to C additions of 0, 1.2 and 4 t C ha$^{-1}$ yr$^{-1}$.

Discussion & Conclusion

These sections make sense to me and I have no further comments.

Thank you for your constructive feedback.

References for the revision:

Georgiou, K., Jackson, R.B., Vindušková, O., Abramoff, R.Z., Ahlström, A., Feng, W., Harden, J.W., Pellegrini, A.F.A., Polley, H.W., Soong, J.L., Riley, W.J., Torn, M.S., 2022. Global stocks and capacity of mineral-associated soil organic carbon. Nat Commun 13, 3797. https://doi.org/10.1038/s41467-022-31540-9

Gurung, R.B., Ogle, S.M., Breidt, F.J., Williams, S.A., Parton, W.J., 2020. Bayesian calibration of the DayCent ecosystem model to simulate soil organic carbon dynamics and reduce model uncertainty. Geoderma 376, 114529. https://doi.org/10.1016/j.geoderma.2020.114529

Kamoni, P.T., Gicheru, P.T., Wokabi, S.M., Easter, M., Milne, E., Coleman, K., Falloon, P., Paustian, K., 2007. Predicted soil organic carbon stocks and changes in Kenya between 1990 and 2030. Agriculture, Ecosystems & Environment, Soil carbon stocks at regional scales 122, 105–113. https://doi.org/10.1016/j.agee.2007.01.024

Menichetti, L., Kätterer, T., Bolinder, M.A., 2020. A Bayesian modeling framework for estimating equilibrium soil organic C sequestration in agroforestry systems. Agriculture, Ecosystems & Environment 303, 107118. https://doi.org/10.1016/j.agee.2020.107118

Piepho, H.-P., 2018. Letters in Mean Comparisons: What They Do and Don't Mean. Agronomy Journal 110, 431–434. https://doi.org/10.2134/agronj2017.10.0580

Ťupek, B., Launiainen, S., Peltoniemi, M., Sievänen, R., Perttunen, J., Kulmala, L., Penttilä, T., Lindroos, A.J., Hashimoto, S., Lehtonen, A., 2019. Evaluating CENTURY and Yasso soil carbon models for CO2 emissions and organic carbon stocks of boreal forest soil with Bayesian multi-model inference. European Journal of Soil Science 70, 847–858. https://doi.org/10.1111/ejss.12805